# EXACT RECOVERY GUARANTEES FOR PARAMETER-IZED NONLINEAR SYSTEM IDENTIFICATION PROBLEM UNDER ADVERSARIAL ATTACKS

## ABSTRACT

In this work, we study the system identification problem for parameterized non-linear systems using basis functions under adversarial attacks. Motivated by the LASSO-type estimators, we analyze the exact recovery property of a nonsmooth estimator, which is generated by solving an embedded $\ell_1$-loss minimization problem. First, we derive necessary and sufficient conditions for the well-specifiedness of the estimator and the uniqueness of global solutions to the underlying optimization problem. Next, we provide exact recovery guarantees for the estimator under two different scenarios of boundedness and Lipschitz continuity of the basis functions. The non-asymptotic exact recovery is guaranteed with high probability, even when there are more severely corrupted data than clean data. Finally, we numerically illustrate the validity of our theory. This is the first study on the sample complexity analysis of a nonsmooth estimator for the nonlinear system identification problem.

## 1 INTRODUCTION

Dynamical systems are the foundation for the areas of sequential decision-making, reinforcement learning, control theory, and recurrent neural networks. They are imperative for modeling the mechanics governing the system and for predicting the states of a system. However, it is cumbersome to exactly model these systems due to the growing complexity of contemporary systems. Thus, the learning of these system dynamics is essential for an accurate decision-making. The problem of estimating the dynamics of a system using past information collected from the system is called the *system identification* problem. This problem is ubiquitously studied in the control theory literature for systems under relatively small independent and identically distributed (i.i.d.) noise due to modeling, measurement, and sensor errors. Nevertheless, safety-critical applications, such as power systems, autonomous vehicles, and unmanned aerial vehicles, require the robust estimation of the system due to the possible presence of adversarial disturbance, such as natural disasters and data manipulation through cyberattacks and system hacking. Although machine learning techniques have been successful in addressing a wide range of problems, such as computer vision and language processing, their application in safety-critical systems has been extremely limited due to the lack of theoretical guarantees. This paper offers a strong result in this regard, which is concerned with studying dynamical systems via machine learning techniques.

As a motivating example, we consider the dynamical system corresponding to a power system (e.g., the U.S. electrical grid or a regional interconnect of the grid), where the states capture various physical parameters such as voltage magnitudes and frequencies in different parts of the system. With the objective of increasing sustainability, resiliency and efficiency of energy systems, modern power systems include a large volume of wind turbines, solar panels, and electric cars. The operation of power systems is further complicated by the fact that people have started to play an active role by observing electricity prices and taking strategic actions in response to the price signals. On the other hand, sensors have been widely installed across the grid to collect data to enable data-driven grid operation. This has raised a major concern since a small strategic data manipulation would cause power suppliers to over-supply or under-supply power electricity, which could lead to a system-wide blackout. This case can be modeled as a nonlinear dynamical system where the input of the system is subject to stealth attacks at various locations leading to injecting wrong values of electricity into the system. Given the presence of a large set of new devices in the system coupled by strategic human

behavior, the power operators do not have a complete model of the dynamical system. Therefore, they may need to learn the system and any possible adversarial attack simultaneously to be able to nullify the attack and restore the system's operation. When an attack to the input is not controlled, it will affect the transient behavior and make the signals unstable , which leads to a cascading failure across the grid.

The prior system identification literature has mainly focused on attacks on measurements, where the goal is to extract knowledge from noisy and corrupted measurements (such as the matrix sensing problem in machine learning). However, this paper considers an emerging and overlooked type of attack for safety-critical systems, where the attack on input data or actuators leads to injecting a wrong input signal into the system which affects the states of the system and makes them unstable. More relevant literature focused on the asymptotic properties of the least-squares estimator (LSE) Chen & Guo (2012); Ljung et al. (1999); Ljung & Wahlberg (1992); Bauer et al. (1999), and with the emergence of statistical learning theory, this area evolved into studying the necessary number of samples for a specific error threshold to be met Tsiamis et al. (2023). While early non-asymptotic analyses centered on linear-time invariant (LTI) systems with i.i.d. noise using mixing arguments Kuznetsov & Mohri (2017); Rostamizadeh & Mohri (2007), recent research employs martingale and small-ball techniques to provide sample complexity guarantees for LTI systems Simchowitz et al. (2018); Faradonbeh et al. (2018); Tsiamis & Pappas (2019). For nonlinear systems, recent studies investigated parameterized models Noël & Kerschen (2017); Nowak (2002); Foster et al. (2020); Sattar & Oymak (2022); Ziemann et al. (2022), showing convergence of recursive and gradient algorithms to true parameters with a rate of $T^{-1/2}$ using martingale techniques and mixing time arguments. Furthermore, efforts towards nonsmooth estimators for both linear and nonlinear systems Feng & Lavaei (2021); Feng et al. (2023); Yalcin et al. (2023), particularly in handling dependent and adversarial noise vectors, are limited. Robust regression techniques utilizing regularizers have been developed Xu et al. (2009); Bertsimas & Copenhaver (2018); Huang et al. (2016), yet non-asymptotic analysis on sample complexity remains sparse, especially for dynamical systems due to sample auto-correlation. A more detailed literature review is provided in Appendix.

This paper paves the way for the area of online optimal control in presence of adversaries and the first step is to learn the dynamics of the system, known as the system identification problem. More specifically, we study the system identification problem for parameterized nonlinear systems in the presence of adversarial attacks. We model the unknown nonlinear functions describing the system via a linear combination of some given basis functions, by taking advantage of their representation properties. Our goal is to learn the parameters of these basis functions that govern the updates of the dynamical system. Mathematically, we consider the following autonomous dynamical system:

$$x_0 = 0_n, \quad x_{t+1} = \bar{A} f(x_t) + \bar{d}_t, \quad \forall t \in \{0, \ldots, T-1\}, \tag{1}$$

where $f : \mathbb{R}^n \mapsto \mathbb{R}^m$ is a combination of $m$ known basis functions and $\bar{A} \in \mathbb{R}^{n \times m}$ is the unknown matrix of parameters. In addition, the system trajectory is attacked by the adversarial noise or disturbance $\bar{d}_t \in \mathbb{R}^n$, which is unknown to the system operator. At any time instance that the system is not attacked, we have $\bar{d}_t = 0$. In other words, the noise only stems from adversarial attacks. The goal of the system identification problem is to recover the ground truth matrix $\bar{A}$ using observations from the states of the system, i.e., $\{x_0, \ldots, x_T\}$. The adversarial noise $\bar{d}_t$'s are designed by an attacker to maximize the impact as much as possible and yet keep the attacks undetectable to the system operator. The underlying assumptions about the noise model will be given later.

One of the main challenges of this estimation problem is the time dependence of the collected samples. As opposed to the empirical risk minimization problem, there exists auto-correlation among the samples $\{x_0, \ldots, x_T\}$. As a result, the common assumption that the samples are i.i.d. instances of the data generation distribution is violated. The existence of the auto-correlation imposes significant challenges on the theoretical analysis, and we address it in this work by proposing a novel and non-trivial extension of the area of exact recovery guarantees to the system identification problem. Since the adversarial attacks $\bar{d}_t$ are unknown to the system operator, it is necessary to utilize estimators to the ground truth $\bar{A}$ that are robust to the noise $\bar{d}_t$ and converge to $\bar{A}$ when the sample size $T$ is large enough. Our work is inspired by Yalcin et al. (2023) that studies the above problem for linear systems. The linear case is noticeably simpler than the nonlinear system identification problem since each observation $x_t$ becomes a linear function of previous disturbances. In the nonlinear case, the relationship between the measurements and the disturbances are highly sophisticated, which requires significant technical developments compared to the linear case in Yalcin et al. (2023).

Motivated by the exact recovery property of nonsmooth loss functions (e.g., the $\ell_1$-norm and the nuclear norm), we consider the following estimator:

$$\hat{A} \in \arg \min_{A \in \mathbb{R}^{n \times m}} \sum_{t=0}^{T-1} \|x_{t+1} - Af(x_t)\|_2. \tag{2}$$

We note that the optimization problem on the right hand-side is convex in $A$ (while having a nonsmooth objective) and, therefore, it can be solved efficiently by various existing optimization solvers. The estimator equation 2 is closely related to the LASSO estimator in the sense that the loss function in equation 2 can be viewed as a generalization of the $\ell_1$-loss function. More specifically, in the case when $n = 1$, the estimator equation 2 reduces to

$$\hat{A} \in \arg \min_{A \in \mathbb{R}^{1 \times m}} \sum_{t=0}^{T-1} |x_{t+1} - Af(x_t)|,$$

which is the auto-correlated linear regression estimator with the $\ell_1$-loss function.

In this work, the goal is to prove the efficacy of the above estimator by obtaining mild conditions under which the ground truth $\bar{A}$ can be *exactly recovered* by the estimator equation 2. More specifically, we focus on the following questions:

i) What are the *necessary and sufficient* conditions such that $\bar{A}$ is an optimal solution to the optimization problem in equation 2 or the unique solution?

ii) What is the required number of samples such that the above necessary and sufficient conditions are satisfied with high probability under certain assumptions?

In this work, we provide answers to the above questions. In Section 2, we first analyze the necessary and sufficient conditions for the global optimality of $\bar{A}$ for the problem in equation 2. Then, in Section 3, we establish the necessary and sufficient conditions such that $\bar{A}$ is the unique solution. The results in these two sections provide an answer to question (i). Finally, in Sections 5 and 6, we derive lower bounds on the number of samples $T$ such that $\bar{A}$ is the unique solution with high probability in the case when the basis function $f$ is bounded or Lipschitz continuous, respectively. These results serve as an answer to question (ii). We provide numerical experiments that support the theoretical results throughout the paper in Section 7. This work provides the first non-asymptotic sample complexity analysis to the exact recovery of the nonlinear system identification problem.

**Notation.** For a positive integer $n$, we use $0_n$ and $I_n$ to denote the $n$-dimensional vector with all entries being 0 and the $n$-by-$n$ identity matrix. For a matrix $Z$, $\|Z\|_F$ denotes its Frobenius norm and $\mathbb{S}_F$ is the unit sphere of matrices with Frobenius norm $\|Z\|_F = 1$. For two matrices $Z_1$ and $Z_2$, we use $\langle Z_1, Z_2 \rangle = \mathrm{Tr}(Z_1^\top Z_2)$ to denote the inner-product. For a vector $z$, $\|z\|_2$ and $\|z\|_\infty$ denote its $\ell_2$- and $\ell_\infty$-norms, respectively. Moreover, $\mathbb{S}^{n-1}$ is the unit ball $\{z \in \mathbb{R}^n | \|z\|_2 = 1\}$. Given two functions $f$ and $g$, the notation $f(x) = \Theta[g(x)]$ means that there exist universal positive constants $c_1$ and $c_2$ such that $c_1 g(x) \leq f(x) \leq c_2 g(x)$. The relation $f(x) \lesssim g(x)$ holds if there exists a universal positive constant $c_3$ such that $f(x) \leq c_3 g(x)$ holds with high probability when $T$ is large. The relation $f(x) \gtrsim g(x)$ holds if $g(x) \lesssim f(x)$. $|S|$ shows the cardinality of a given set $S$. $\mathbb{P}(\cdot)$ and $\mathbb{E}(\cdot)$ denote the probability of an event and the expectation of a random variable. A Gaussian random vector $X$ with mean $\mu$ and covariance matrix $\Sigma$ is written as $X \sim \mathcal{N}(\mu, \Sigma)$.

## 2 GLOBAL OPTIMALITY OF GROUND TRUTH

In this section, we derive conditions under which the ground truth $\bar{A}$ is a global minimizer to the optimization problem in equation 2. By the system dynamics, the optimization problem is equivalent to

$$\min_{A \in \mathbb{R}^{n \times m}} \sum_{t=0}^{T-1} \|(\bar{A} - A)f(x_t) + \bar{d}_t\|_2, \tag{3}$$

where $x_0, \ldots, x_T$ are generated according to the unknown system under adversaries. We define the set of attack times as $\mathcal{K} := \{t \mid \bar{d}_t \neq 0\}$ and the normalized attacks as

$$\hat{d}_t := \bar{d}_t / \|\bar{d}_t\|_2, \quad \forall t \in \mathcal{K}.$$

The following theorem provides a necessary and sufficient condition for the global optimality of ground truth matrix $\bar{A}$ in problem equation 3.

**Theorem 1** (Necessary and sufficient condition for optimality). *The ground truth matrix $\bar{A}$ is a global solution to problem equation 3 if and only if*

$$\sum_{t \in \mathcal{K}} \hat{d}_t^\top Z f(x_t) \leq \sum_{t \in \mathcal{K}^c} \|Z f(x_t)\|_2, \quad \forall Z \in \mathbb{R}^{n \times m}, \tag{4}$$

*where $\mathcal{K}^c := \{0, \ldots, T-1\} \backslash \mathcal{K}$.*

Theorem 1 provides a necessary and sufficient condition for the well-specifiedness of optimization problem equation 3. Intuitively, we can view the left-hand side as the impact of noisy attacks and the right-hand side as the normal dynamics. If the impact from noise does not override the correct system dynamics, then the predictor is able to recover the ground truth system dynamics. The condition equation 4 is established by applying the generalized Farkas' lemma, which avoids the inner approximation of the $\ell_2$-ball by an $\ell_\infty$-ball in Yalcin et al. (2023). As a result, the sample complexity bounds to be obtained in this work are stronger than those in Yalcin et al. (2023) when specialized to the setting of linear systems; see Sections 5 and 6 for more details.

Using the condition in Theorem 1, we can derive the necessary conditions and sufficient conditions for the optimality of $\bar{A}$.

**Corollary 1** (Sufficient condition for optimality). *If it holds that*

$$\sum_{t \in \mathcal{K}} \|Z f(x_t)\|_2 \leq \sum_{t \in \mathcal{K}^c} \|Z f(x_t)\|_2, \quad \forall Z \in \mathbb{R}^{n \times m}, \tag{5}$$

*then the ground truth matrix $\bar{A}$ is a global solution to problem equation 3.*

**Corollary 2** (Necessary condition for optimality). *If the ground truth matrix $\bar{A}$ is a global solution to problem equation 3, then it holds that*

$$\left\| \sum_{t \in \mathcal{K}} f(x_t) \hat{d}_t^\top \right\|_F \leq \sum_{t \in \mathcal{K}^c} \|f(x_t)\|_2. \tag{6}$$

*In the case when $m = 1$, condition equation 6 is necessary and sufficient.*

The proof of Corollaries 1 and 2 is provided in the appendix. The above conditions are more general than many existing results in literature; see Examples 1 and 2 provided in Appendix B.

## 3 UNIQUENESS OF GLOBAL SOLUTIONS

In this section, we derive conditions under which the ground truth solution $\bar{A}$ is the unique solution to problem equation 3. We obtain the following necessary and sufficient condition on the uniqueness of global solutions, which is an extension of Theorem 1.

**Theorem 2** (Necessary and sufficient condition for uniqueness). *Suppose that condition equation 4 holds. The ground truth $\bar{A}$ is the unique global solution to problem equation 3 if and only if for every nonzero $Z \in \mathbb{R}^{n \times m}$, the following logical condition holds:*

$$\sum_{t \in \mathcal{K}} \hat{d}_t^\top Z f(x_t) = \sum_{t \in \mathcal{K}^c} \|Z f(x_t)\|_2 \implies \sum_{t \in \mathcal{K}} \left| \hat{d}_t^\top Z f(x_t) \right| < \sum_{t \in \mathcal{K}} \|Z f(x_t)\|_2, \tag{7}$$

*which means that whenever the left-hand side equality is satisfied for some nonzero $Z$, the right-hand side inequality must also hold.*

Based on the above theorem, the following corollary provides a sufficient condition for the uniqueness of $\bar{A}$, which is easier to verify in practice compared to equation 7. Note that the corollary also generalizes the sufficiency part of Corollary 2 to the multi-dimensional case.

**Corollary 3** (Sufficient condition for uniqueness). *Suppose that condition equation 4 holds. If it holds that*

$$\sum_{t \in \mathcal{K}} \hat{d}_t^\top Z f(x_t) < \sum_{t \in \mathcal{K}^c} \|Z f(x_t)\|_2, \quad \forall Z \in \mathbb{R}^{n \times m} \quad \text{s.t. } Z \neq 0, \tag{8}$$

*then the ground truth matrix $\bar{A}$ is the unique global solution to problem equation 3.*

*Proof.* The logical condition in equation 7 states that whenever the left-hand side equality is satisfied for some nonzero $Z$, the right-hand side inequality must also hold. Under the assumption in equation 8, there is no nonzero $Z$ satisfying the left hand-side equality. This implies that the logical condition in equation 7 automatically holds and, thus, Theorem 2 implies the uniqueness of $\bar{A}$. □

Similar to the optimality conditions in Section 2, Theorem 2 improves and generalizes the results for first-order systems, namely, Theorem 1 in Feng & Lavaei (2021). Please see the Example 3 provided in Appendix B.

## 4 ATTACK MODEL

To capture the severity of an attack strategy, we consider how frequently the system is under attack. We model this by an attack probability $p$ defined below.

**Definition 1** (Probabilistic attack model). *For each time instance $t$, the attack vector $\bar{d}_t$ is nonzero with probability $p \in (0, 1)$, which is also independent from other time instances.*

Note that the attack vectors $\bar{d}_t$'s are allowed to be correlated over time and Definition 1 is only about the times at which an attack happens. We do not assume that the probability distribution $p$ or the model generating the attack is known. Recall that we define $\mathcal{K} := \{t \mid \bar{d}_t \neq 0\}$. Then, with probability at least $1 - \exp[-\Theta(pT)]$, it holds that $|\mathcal{K}| = \Theta(pT)$. When $p$ is close to 0, the system is rarely under attack. A particular focus of this paper is on the case with $p$ close to 1, meaning that the system is under attack at almost all times. There are two related problems about attack analysis: (1) attack problem: the attacker aims to design the attack vectors $\bar{d}_t$ to maximize the damage on the system, (2) defense problem: the system operator aims to detect any suspicious attack and nullify it. To design a defense mechanism, there are two common strategies (which may also be combined): (i) inspect the behavior of the input of the system to detect possible attacks, (ii) analyze the collected state values to learn whether there has been any attack. In the next theorem, we show that if there are no assumptions on the attack vectors, the defense mechanism (ii) never works, meaning that there is no estimator that can reliably detect the attacks.

**Theorem 3.** *Consider the linear system $x_{t+1} = \bar{A}x_t + \bar{d}_t$, together with a linear subspace $D \subset \mathbb{R}^n$ whose dimension is less than $n$. Assume that the attack $\bar{d}_t$ is always chosen from this not-full-dimensional subspace. Then, there does not exist any estimator that can correctly identify the matrix $\bar{A}$ all the time from the states $x_0, ..., x_T$ no matter how large $T$ is.*

In other words, if the attacker is free to choose the attack values, the best strategy is to restrict them to a certain subspace so that no estimator works. However, such biased attacks are easier to detect using the defense strategy (i), meaning that the operator can study the statistical behavior of the input directly and realize that it cannot be a natural disturbance and would flag it as attack. Hence, the danger of choosing such *extreme* attacks by an attacker is that the system operator will have a higher chance to detect and nullify it through a properly designed defense mechanism. As a result, the attacker should strategically avoid extreme attacks and focus on those attacks that have a less chance of being detected.

As an example, consider the first-order system $x_{t+1} = x_t + \bar{d}_t$ where $\bar{d}_t \in \mathbb{R}$. Assume that this is a physical system which cannot accept any input with a magnitude larger than a given limit $\gamma$. Then, the most severe attack would be to choose $\bar{d}_t$ to be equal to $\gamma$ all the time, which makes the state grow as fast as possible. However, a proper defense mechanism can quickly detect this attack by realizing that the injected input is not a natural disturbance but rather a strategically designed input. Hence, the attacker should dial back and select $\bar{d}_t$ to be $+\bar{\gamma}$ and $-\bar{\gamma}$ with $\bar{\gamma} < \gamma$ to achieve two properties: (i) avoiding hitting the maximum limit, (ii) making the attack look like a disturbance by having a zero mean. This attack is much harder to detect but is still impactful since it affects the variance of $x_t$ and makes it oscillate. The idea behind this example was that the attacker should sometimes choose positive-value attacks and other times choose negative-value attacks. We can capture this idea through the notion of attack direction and define the following stealthy condition on the attack, which is aligned with the existing conditions in the literature Candès et al. (2011); Chen et al. (2021). To state the stealthy condition, we define the filtration $\mathcal{F}_t := \sigma\{x_0, x_1, \ldots, x_t\}$.

**Assumption 1** (Stealthy condition). *Conditional on the past information $\mathcal{F}_t$ and the event that $\bar{d}_t \neq 0_n$, the attack direction $\hat{d}_t = \bar{d}_t / \|\bar{d}_t\|_2$ is zero-mean.*

The above notion of stealthy attacks has been commonly used in real-world systems. To illustrate the idea, consider an energy system with two nodes: node 1 represents a neighborhood of homes, and node 2 is a supplier owned by a utility company. Every five minutes, node 1 reports to node 2 the amount of electrical power the neighborhood needs for the next five minutes. Assume the neighborhood requires a constant amount of 1 unit of power for the next five hours. However, an attacker has compromised the communication channel between node 1 and node 2, altering the requested amount from 1 unit to either $1 - e$ or $1 + e$ every half hour, where $e$ is a large number compared to 1 unit, and its value depends on $|x_t|$ at the current or previous time. Since the average of $1 - e$ and $1 + e$ is 1, this could serve as a stealthy attack. When node 1 actually needs one unit of power, but node 2 generates either $1 - e$ or $1 + e$, the mismatch violates the laws of physics, potentially triggering grid instability and leading to a blackout. In this scenario, the attacker infrequently injects an adversarial input into the system that can take arbitrary values. For the attack to remain stealthy, the mean of the attack should be zero (in this example, the mean of $1 - e$ and $1 + e$ is zero). Attacks of this type have appeared in different parts of the word leading to large-scale blackouts. The cyberattack problem on power systems fits perfectly into our mathematical models. Note that power system operators use hypothesis testing to detect anomalies in data, and if there is a nonzero mean, they would flag it.

In the next two sections, we provide lower bounds on the sample complexity $T$ such that the ground truth $\bar{A}$ is the unique solution to problem equation 3. The next section relies on Assumption 1. After that, we study unbounded basis functions, which needs Assumption 6 that is somewhat stronger than Assumption 1 but still practical. Note that the former assumption requires the attack directions to have a zero mean while the latter assumption requires the attack directions to be uniform. In both cases, the attack magnitudes are arbitrary (similar to the parameter $e$ in the above energy example) and it can attack arbitrary values to maximize the damage on the system.

Note that the existing results in the literature on finite-time system identifications for nonlinear systems have all assumed that $\bar{d}_t$'s are i.i.d. disturbances. Our work is the first one studying correlated disturbances and our results contribute not only to the area of attack detection but also to the area of non-attack corrollated disturbances. The assumptions made in this paper automatically hold under the assumptions made in the existing literature (such as i.i.d. Gaussian).

## 5 BOUNDED BASIS FUNCTION

In this section, we consider the case when the basis function $f$ is bounded.

**Assumption 2** (Bounded basis function). *The basis function $f : \mathbb{R}^n \mapsto \mathbb{R}^m$ satisfies*

$$\|f(x)\|_\infty \leq B, \quad \forall x \in \mathbb{R}^n,$$

*where $B > 0$ is a constant.*

Finally, to avoid the degenerate case, we assume that the norm of basis function is lower bounded under conditional expectation after an attack.

**Assumption 3** (Non-degenerate condition). *Conditional on the past information $\mathcal{F}_t$ and the event that $\bar{d}_t \neq 0_n$, the attack vector and the basis function satisfy*

$$\lambda_{min} \left[ \mathbb{E} \left[ f(x + \bar{d}_t) f(x + \bar{d}_t)^\top \mid \mathcal{F}_t, \bar{d}_t \neq 0_n \right] \right] \geq \lambda^2, \quad \forall x \in \mathbb{R}^n,$$

*where $\lambda_{min}(F)$ is the minimal eigenvalue of matrix $F$ and $\lambda > 0$ is a constant.*

Intuitively, the non-degenerate assumption allows the exploration of the trajectory in the state space. More specifically, it is necessary that the matrix

$$[f(x_t), \ t \in \mathcal{K}^c] \in \mathbb{R}^{m \times (T - |\mathcal{K}|)} \tag{9}$$

is rank-$m$ for the condition equation 8 to hold; see the proof of Theorem 5 for more details. The non-degenerate assumption guarantees that the basis function $f(x + \bar{d}_t)$ spans the whole state space in expectation and thus, the matrix equation 9 is full-rank with high probability when $T$ is large.

The following theorem proves that when the sample complexity is large enough, the estimator equation 2 exactly recovers the ground truth $\bar{A}$ with high probability.

**Theorem 4** (Exact recovery for bounded basis function). *Suppose that Assumptions 1-3 hold and define $\kappa := B/\lambda \geq 1$. For all $\delta \in (0, 1]$, if the sample complexity $T$ satisfies*

$$T \geq \Theta \left[ \frac{m^2 \kappa^4}{p(1-p)^2} \left[ mn \log \left( \frac{m\kappa}{p(1-p)} \right) + \log \left( \frac{1}{\delta} \right) \right] \right], \tag{10}$$

*then $\bar{A}$ is the unique global solution to problem equation 3 with probability at least $1 - \delta$.*

The above theorem provides a non-asymptotic bound on the sample complexity for the exact recovery with a specified probability $1 - \delta$. The lower bound grows with $m^3 n$, which implies that the required number of samples increases when the number of states $n$ and the number of basis functions $m$ is larger. In addition, the sample complexity is larger when $B$ is larger or $\lambda$ is smaller. This is also consistent with the intuition that $B$ reflects the size of the space spanned by the basis function and $\lambda$ measures the "speed" of exploring the spanned space.

For the dependence on attack probability $p$, we show in the next theorem that the dependence on $1/[p(1-p)]$ is inevitable under the probabilistic attack model. In addition, the theorem also establishes a lower bound on the sample complexity that depends on $m$ and $\log(1/\delta)$.

**Theorem 5.** *Suppose that the sample complexity satisfies*

$$T < \frac{m}{2p(1-p)}.$$

*Then, there exists a basis function $f : \mathbb{R}^n \mapsto \mathbb{R}^m$ and an attack model such that Assumptions 1-3 hold and the global solutions to problem equation 3 are not unique with probability at least $\max \left\{ 1 - 2 \exp \left( -m/3 \right), 2[p(1-p)]^{T/2} \right\}$. Furthermore, given a constant $\delta \in (0, 1]$, if*

$$T < \max \left\{ \frac{m}{2p(1-p)}, \frac{2}{-\log[p(1-p)]} \log \left( \frac{2}{\delta} \right) \right\},$$

*then the global solutions to problem equation 3 are not unique with probability at least $\max \left\{ 1 - 2 \exp \left( -m/3 \right), \delta \right\}$.*

**Remark 1.** *The main goal of the paper is to show that exact recovery is possible when more than half of the data are arbitrarily corrupted. We provide an upper bound on the required time horizon in Theorem 4. This result has a major implication for real-world systems. On the other hand, the lower bound in Theorem 5 is mainly a theoretical result. Unlike machine learning problems where the problem size is possibly on the order of tens of millions, the number of states for many real-world systems is much lower and less than several thousands. For that reason, our upper bound is already a practical number and improving the lower bound may have a marginal practical value, although tightening the lower bound is a relevant and interesting theoretical problem.*

## 6 LIPSCHITZ BASIS FUNCTION

In this section, we consider the case when the basis function $f(x)$ is Lipschitz continuous in $x$. More specifically, we make the following assumption.

**Assumption 4** (Lipschitz basis function). *The basis function $f : \mathbb{R}^n \mapsto \mathbb{R}^m$ satisfies*

$$f(0_n) = 0_m \quad and \quad \|f(x) - f(y)\|_2 \leq L\|x - y\|_2, \quad \forall x, y \in \mathbb{R}^n,$$

*where $L > 0$ is the Lipschitz constant.*

As a special case of Assumption 4, the basis function of a linear system is $f(x) = x$, which is Lipschitz continuous with Lipschitz constant 1.

**Remark 2.** *Note that the assumptions of boundedness or Lipschitz continuity are always satisfied for dynamical systems since the user has the choice to select appropriate basis functions to satisfy them. More concretely, the user can select any arbitrary set of basis functions to approximate the unknown function as a linear combination of the bases. This is different from classical machine learning problems where a model is trained to learn the function and there is not control on the Lipschitzness. On the other hand, if the user is not allowed to use unbounded basis functions or functions with a high Lipschitz constant, then the number of basis functions used to approximate the*

*unknown function may be higher. However, many real-world dynamical systems, from robotics to energy systems, are obtained from physical laws where the unknown dynamics is well behaved due to the smoothness of laws of physics, such as Newtonian laws and Kirchhoffs laws of electrical circuits. This is different from various machine learning problems for which the targeted optimal policy could be inevitably nonsmooth and highly complicated.*

In addition, we assume that the spectral norm of $\bar{A}$ is bounded.

**Assumption 5** (System stability). *The ground truth $\bar{A}$ satisfies*

$$\rho := \left\| \bar{A} \right\|_2 < \frac{1}{L}.$$

We note that Assumption 5 is related to the asymptotic stability of the dynamic system and is sufficient to avoid the finite-time explosion of the dynamics. We show in Theorem 7 that Assumption 5 may be necessary for exact recovery. Finally, we make the assumption that the attack is sub-Gaussian.

**Assumption 6** (Sub-Gaussian attacks). *Conditional on the filtration $\mathcal{F}_t$ and the event that $\bar{d}_t \neq 0_n$, the attack vector $\bar{d}_t$ is defined by the product $\ell_t \hat{d}_t$, where*

1. *$\hat{d}_t \in \mathbb{R}^n$ and $\ell_t \in \mathbb{R}$ are independent conditional on $\mathcal{F}_t$ and $\bar{d}_t \neq 0_n$;*

2. *$\hat{d}_t$ is a zero-mean unit vector, namely, $\mathbb{E}(\hat{d}_t \mid \mathcal{F}_t, \bar{d}_t \neq 0_n) = 0_n$ and $\|\hat{d}_t\|_2 = 1$;*

3. *$\ell_k$ is zero-mean and sub-Gaussian with parameter $\sigma$.*

As a special case, the sub-Gaussian assumption is guaranteed to hold if there is an upper bound on the magnitude of the attack. The bounded-attack case is common in practical applications since real-world systems do not accept inputs that are arbitrarily large. For example, physical devices have a clear limitation on the input size and the attacks cannot exceed that limit. In Assumption 6, $\hat{d}_t$ and $\ell_t$ play the roles of the direction and intensity (such as magnitude) of the attack, respectively. The parameters $\ell_t$'s could be correlated over time, while $\hat{d}_t$ and $\ell_t$ are assumed to be zero-mean to make the attack stealth.

Under the above assumptions, we can also guarantee the high-probability exact recovery when the sample size $T$ is sufficiently large.

**Theorem 6** (Exact recovery for Lipschitz basis function). *Suppose that Assumptions 3-6 hold and define $\kappa := \sigma L/\lambda \geq 1$. If the sample complexity $T$ satisfies*

$$T \geq \Theta \left[ \max \left\{ \frac{\kappa^{10}}{(1 - \rho L)^3 (1 - p)^2}, \frac{\kappa^4}{p(1 - p)} \right\} \times \left[ mn \log \left( \frac{1}{(1 - \rho L)\kappa p(1 - p)} \right) + \log \left( \frac{1}{\delta} \right) \right] \right], \tag{11}$$

*then $\bar{A}$ is the unique global solution to problem equation 3 with probability at least $1 - \delta$.*

Theorem 6 provides a non-asymptotic sample complexity bound for the case when the basis function is Lipschitz continuous. As a special case, when the basis function is $f(x) = x$ and the attack vector $\bar{d}_t$ obeys the Gaussian distribution $\mathcal{N}(0_n, \sigma^2 I_n)$ conditional on $\mathcal{F}_t$, we have $\kappa = 1$. Compared with Theorem 4, the dependence on attack probability $p$ is improved from $1/[p(1 - p)^2]$ to $1/[p(1 - p)]$, which is a result of the stability condition (Assumption 5). In addition, the dependence on the dimension $m$ is improved from $m^3$ to $m$. Intuitively, the improvement is achieved by improving the upper bound on the norm $\|f(x_t)\|_2$. In the bounded basis function case, the norm is bounded by $\sqrt{m}B$; while in the Lipschitz basis function case, the norm is bounded by $\sigma L$ with high probability, which is independent from the dimension $m$. Finally, the sample complexity bound grows with the parameter $\kappa = \sigma L/\lambda$ and the gap $1 - \rho L$, which is also consistent with the intuition.

On the other hand, we can construct counterexamples showing that when the stability condition (Assumption 5) is violated, the exact recovery fails with probability at least $p$.

**Theorem 7** (Failure of exact recovery for unstable systems). *There exists a system such that Assumptions 3, 4 and 6 are satisfied but for all $T \geq 1$, the ground truth $\bar{A}$ is not a global solution to problem equation 3 with probability at least $p[1 - (1 - p)^{T-1}]$.*

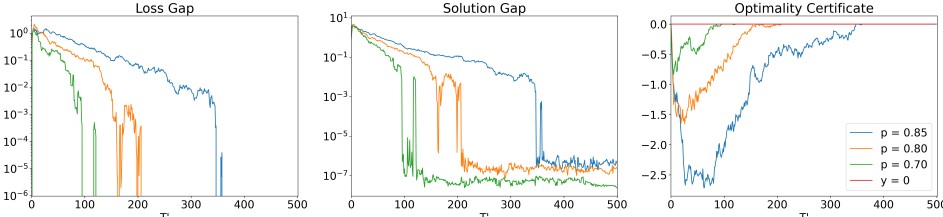

Figure 1: Loss gap, solution gap and optimality certificate of the Lipschitz basis function case with attack probability $p = 0.7, 0.8$ and $0.85$.

## 7  NUMERICAL EXPERIMENTS

We implement numerical experiments for the Lipschitz basis function cases to verify the exact recovery guarantees in Section 6. The descriptions of the basis functions and the results for the bounded basis function case are provided in Appendix. Specifically, we illustrate the convergence of estimator equation 2 with different values of the attack probability $p$, problem dimension $(n, m)$ and spectral norm $\rho$. We also numerically verify the necessary and sufficient condition in Section 3.

**Evaluation metrics.** Given a trajectory $\{x_0, \ldots, x_T\}$, we compute the estimators

$$\hat{A}^{T'} \in \arg \min_{A \in \mathbb{R}^{n \times m}} g_{T'}(A), \quad \forall T' \in \{1, \ldots, T\},$$

where we define the loss function $g_{T'}(A) := \sum_{t=0}^{T'-1} \|x_{t+1} - Af(x_t)\|_2$. In our experiments, we solve the convex optimization by the CVX solver Grant & Boyd (2014). Then, for each $T'$, we evaluate the recovery quality by the following three metrics:

- The **Loss Gap** is defined as $g_{T'}(\bar{A}) - g_{T'}(\hat{A}_{T'})$. The ground truth $\bar{A}$ is a global solution if and only if the loss gap is 0.

- The **Solution Gap** is defined as $\|\bar{A} - \hat{A}_{T'}\|_F$. The ground truth $\bar{A}$ is the unique solution only if the solution gap is 0.

- The **Optimality Certificate** is defined as

$$\min_{Z \in \mathbb{R}^{n \times m}} \sum_{t \in \mathcal{K}^c} \|Zf(x_t)\|_2 - \sum_{t \in \mathcal{K}} \tilde{d}_t^\top Zf(x_t) \quad \text{s.t. } \|Z\|_F \leq 1,$$

which is a convex optimization problem and can be solved by the CVX solver. The ground truth is a global solution if and only if the optimality certificate is equal to 0.

We evaluate the metrics in our experiments to illustrate the performance of the estimator equation 2 and the proposed optimality conditions. For each choice of parameters, we independently generate 10 trajectories using the dynamics equation 1 and compute the average of the three metrics.

**Results.** Since we need to solve estimator equation 2 many times (for different trajectories and steps $T'$), we consider relatively small-scale problems. In practice, the estimator equation 2 is only required for $T' = T$ and we only need to solve a single optimization problem. As a result, estimator equation 2 can be solved for large-scale real-world systems since it is convex and should be solved only once.

We first compare the performance of estimator equation 2 under different values of the attack probability $p$. We choose $T = 500$, $n = 3$ and $p \in \{0.7, 0.8, 0.85\}$. Additionally, we set the upper bound $\rho$ to be 1, which guarantees the stability condition (Assumption 5). The results are plotted in Figure 1. It can be observed that both the loss gap and the solution gap converge to 0 when the number of samples $T'$ is large, which implies that the estimator equation 2 exactly recovers the ground truth $\bar{A}$ when there exists a sufficient number of samples. Moreover, the optimality certificate converges to 0 at the same time as the solution gap, which verifies the validity of our necessary and sufficient condition in Sections 2 and 3. Furthermore, the required number of samples increases with probability $p$, which is consistent with the upper bound in Theorem 6.

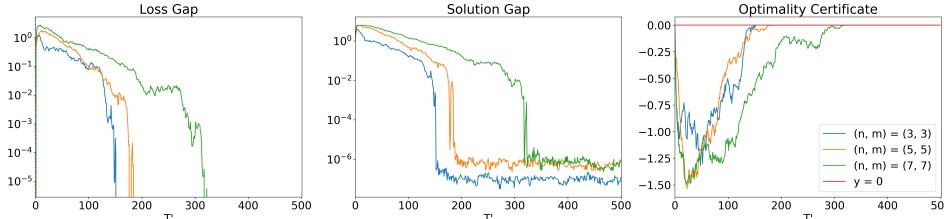

Figure 2: Loss gap, solution gap and optimality certificate of the Lipschitz basis function case with dimension $(n, m) = (3, 3), (5, 5)$ and $(7, 7)$.

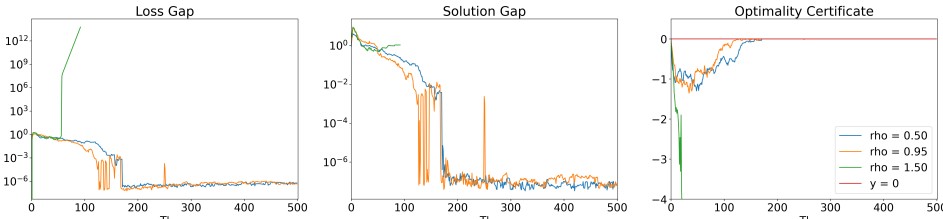

Figure 3: Loss gap, solution gap and optimality certificate of the Lipschitz basis function case with spectral norm $\rho = 0.5, 0.95$ and $1.5$.

Next, we show the performance of estimator equation 2 with different dimensions $(n, m)$. We choose $T = 500$, $p = 0.75$, $\rho = 1$ and $n \in \{3, 5, 7\}$. The results are plotted in Figure 2. We can see that when the problem dimension $(n, m)$ is larger, more samples are required to guarantee the exact recovery. This observation is also consistent with our bound in Theorem 4.

Finally, we illustrate the relation between the sample complexity and the spectral norm $\rho$. In this experiment, we choose $T = 100$, $p = 0.75$ and $n = 3$. To avoid the randomness in the spectral norm $\|\bar{A}\|_2$, we set singular values of $\bar{A}$ to be $\sigma_1 = \cdots = \sigma_n = \rho \in \{0.5, 0.95, 1.5\}$. For the case when $\rho = 1.5$, we terminate the simulation when $\|x_t\|_2 \geq 10^{14}$, which indicates that the trajectory diverges to infinity and this causes numerical issues for the CVX solver. The results are plotted in Figure 3. We can see that the required sample complexity slightly grows when $\rho$ increases from $0.5$ to $0.95$, which is consistent with Theorem 6. In addition, the system is not asymptotically stable when $\rho = 1.5$ and Assumption 5 is violated. The explosion of the system (namely, $\|x_t\|_2 \to \infty$) leads to numerical instabilities in computing the estimator equation 2. With that said, it is possible that estimator equation 2 still achieves the exact recovery with large values of $\rho$, when a stable numerical method is applied to compute the estimator equation 2. This does not contradict with our theory since Theorem 6 only serves as a sufficient condition for the exact recovery.

## 8 CONCLUSION

This paper is concerned with the parameterized nonlinear system identification problem with adversarial attacks. The nonsmooth estimator equation 2 is utilized to achieve the exact recovery of the underlying parameter $\bar{A}$. We first provide necessary and sufficient conditions for the well-specifiedness of estimator equation 2 and the uniqueness of optimal solutions to the embedded optimization problem equation 3. Moreover, we provide sample complexity bounds for the exact recovery of $\bar{A}$ in the cases of bounded basis functions and Lipschitz basis functions using the proposed sufficient conditions. For bounded basis functions, the sample complexity scales with $m^3 n$ in terms of the dimension of the problem and with $p^{-1}(1-p)^{-2}$ in terms of the attack probability up to a logarithm factor. As for Lipschitz basis functions, the sample complexity scales with $mn$ in terms of the dimension of the problem and with $\max\{(1-p)^{-2}, p^{-1}(1-p)^{-1}\}$ in terms of the attack probability up to a logarithm factor. Furthermore, if the sample complexity has a smaller order than $p^{-1}(1-p)^{-1}$, the high-probability exact recovery is not attainable. Hence, the term $p^{-1}(1-p)^{-1}$ in our bounds is inevitable. Lastly, numerical experiments are implemented to corroborate our theory.

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
