## A LITERATURE OVERVIEW

The literature on the system identification problem focused until recently on the asymptotic properties of the least squares estimator (LSE) Chen & Guo (2012); Ljung et al. (1999); Ljung & Wahlberg (1992); Bauer et al. (1999). With the growing popularity of statistical learning theory Vershynin (2018); Wainwright (2019), understanding the number of samples required for a certain error threshold for the system identification problem has gained significant importance. For an overview of the results and proof techniques, the reader is referred to the survey paper Tsiamis et al. (2023). The literature on the non-asymptotic analysis mainly focused on the linear-time invariant (LTI) system identification problem with i.i.d. noise. The earlier research used mixing arguments that are highly dependent on system stability Kuznetsov & Mohri (2017); Rostamizadeh & Mohri (2007). The most recent studies used martingale and small-ball techniques to provide sample complexity guarantees for least-squares estimators applied to LTI systems Simchowitz et al. (2018); Faradonbeh et al. (2018); Tsiamis & Pappas (2019). These works showed that the LSE converges to the true system parameters with the rate $T^{-1/2}$, where $T$ is the number of samples. This result was applied to the linear-quadratic regulator problem using adaptive control to obtain optimal regret bounds Dean et al. (2020); Abbasi-Yadkori & Szepesvári (2011); Dean et al. (2019).

The nonlinear system identification problem is vastly studied Noël & Kerschen (2017); Nowak (2002). Yet, the research on the non-asymptotic analysis of the nonlinear system identification is in its infancy and is mostly focused on parameterized nonlinear systems. Recursive and gradient algorithms designed for the least-squares loss function converge to the true system parameters with the rate $T^{-1/2}$ for nonlinear systems with a known link function $\phi$ of the form $\phi(\bar{A}x_t)$ using martingale techniques Foster et al. (2020) and mixing time arguments Sattar & Oymak (2022). Most recently, Ziemann et al. (2022) provided sample complexity guarantees for non-parametric learning of nonlinear system dynamics, which scales with $T^{-1/(2+q)}$. Here, $q$ scales with the size of the function class in which we search for the true dynamics. Existing studies on both linear and nonlinear system identification analyzed the problem under i.i.d. (sub)-Gaussian noise structures.

Despite the growing interest on non-asymptotic system identification, the literature on the system identification problem with nonsmooth estimators that can handle dependent and adversarial noise vectors is limited to linear systems. The studies Feng & Lavaei (2021) and Feng et al. (2023) considered a nonsmooth convex estimator in the form of the least absolute deviation estimator and analyzed the required conditions for the exact recovery of the system dynamics using the KKT conditions and the Null Space Property from the LASSO literature. Later, Yalcin et al. (2023) showed that exact recovery of system parameters is achievable with high probability even when more than half of the data is corrupted. This provides a further avenue of research for the adversarially robust system identification problem. Yalcin et al. (2023) was the first paper that employed a nonsmooth estimator for nonlinear system identification. Compared with Yalcin et al. (2023), the presence of nonlinear basis functions makes it impossible to directly analyze the optimization problem by writing the explicit expression of $x_t$; see the proof of Theorem 2 in Yalcin et al. (2023). Note that when the system is in the form of $x_{t+1} = Ax_t$, then $x_t$ can be written directly as $A^t x_0$ and we only need to analyze the eigenvalues of $A$. For a nonlinear system in the form of $x_{t+1} = f(x_t)$, writing $x_t$ in terms of $x_0$ needs the composition of $t$ functions, and this cannot be done analytically. There does not exist counterpart of linear-system eigenvalue analysis for nonlinear systems. This challenge is repeatedly acknowledged in many textbooks of nonlinear systems in the area of control theory, and for that reason several results known for linear systems do not have a counterpart in the nonlinear setting. Therefore, we took a different approach to estimate the terms that appear in the uniqueness condition equation 7 in Section 3 In addition, we do not need the stability assumption (Assumption 5) in the case of a bounded basis function (note that the stability assumption was the key in the linear case since it was directly related to the eigenvalues of $A$ and the behavior of $A^t$ when $t$ goes to infinity). As a result, the proof for the bounded case is novel and different from those in Yalcin et al. (2023). Finally, by utilizing the generalized Farkas' lemma, the necessary and sufficient conditions in Sections 2-3 are novel and stronger than the sufficient conditions in Yalcin et al. (2023).

On the other hand, robust regression techniques have been developed using regularizers in the objective function Xu et al. (2009); Bertsimas & Copenhaver (2018); Huang et al. (2016). In addition, the robust estimation literature provided multiple nonsmooth estimators, such as M-estimators, least absolute deviation, convex estimators, least median squares, and least trimmed squares Seber & Lee (2012). The convex estimator equation 2 was proposed in Bako & Ohlsson (2016); Bako (2017) in

the context of robust regression. They showed that the estimator can achieve the exact recovery when we have infinitely many samples. However, the study lacks a non-asymptotic analysis on the sample complexity. Additionally, the analysis techniques cannot be applied to the analysis of dynamical systems due to the autocorrelation among the samples.

The two recent papers Wu et al. (2022); Kumar et al. (2022) focused on the reinforcement learning (RL) problem, whose goal is to maximize the reward function. In contrast, in the system identification problem, the goal is to recover the underlying system dynamics and the application may not incur a naturally defined reward function. The two referenced papers assumed the perturbation to be bounded, which is a strict assumption and may not hold in practice. More importantly, controlling a system without learning its dynamics (e.g., by model-free RL techniques) is a dangerous approach since the policy during exploration could shift the state move out of safe regions and trigger instability; see the survey paper Moerland et al. (2023). Hence, for safety-critical systems, it is usually essential to first learn the system and then apply a control method, which could be classic optimal control or RL algorithms. Our paper is concerned with learning the model of the system where there is an attack to its dynamics. The existing RL methods, including Wu et al. (2022); Kumar et al. (2022), are concerned with a different problem. In addition, we note that although the area of robust model-based RL techniques is rich, our setting of unknown systems requires model-free RL techniques.

## B  COMPARING RESULTS TO EXISTING WORK

**Example 1** (First-order systems). *In the special case when $n = m = 1$ and the basis function is $f(x) = x$, condition equation 6 reduces to*

$$\left| \sum_{t \in \mathcal{K}} \hat{d}_t x_t \right| \leq \sum_{t \in \mathcal{K}^c} |x_t|,$$

*which is the same as Theorem 1 in Feng & Lavaei (2021).*

**Example 2** (Linear systems). *We consider the case when $m = n$ and the basis function is $f(x) = x$. We also assume the $\Delta$-spaced attack model; see the definition in Yalcin et al. (2023). By considering the attack period starting at the time step $t_1$, a sufficient condition to guarantee condition equation 4 is given by*

$$\hat{d}^\top Z \bar{A}^{\Delta-1} \bar{d}_{t_1} \leq \sum_{t=0}^{\Delta-2} \left\| Z \bar{A}^t \bar{d}_{t_1} \right\|_2, \quad \forall Z \in \mathbb{R}^{n \times n}, \tag{12}$$

*where we denote $\hat{d} := \hat{d}_{t_1}$ for simplicity. Let $\hat{D} \in \mathbb{R}^{n \times (n-1)}$ be the matrix of orthonormal bases of the orthogonal complementary space of $f$, namely, $\hat{D}^\top \hat{d} = 0$, $\hat{D}^\top \hat{D} = I_{n-1}$, and $\hat{D}\hat{D}^\top = I_n - \hat{d}\hat{d}^\top$. Then, we can calculate that*

$$\left\| Z \bar{A}^t \bar{d}_{t_1} \right\|_2^2 \geq \left( Z \bar{A}^t \bar{d}_{t_1} \right)^\top \hat{d}\hat{d}^\top \left( Z \bar{A}^t \bar{d}_{t_1} \right),$$

*where the equality holds when $\hat{D}^\top Z \bar{A}^t \bar{d}_{t_1} = 0$, i.e., $Z \bar{A}^t \bar{d}_{t_1}$ is parallel with $\hat{d}$. Therefore, for condition equation 12 to hold, it is equivalent to consider $Z$ with the form $Z = \hat{d} z^\top$ for some vector $z \in \mathbb{R}^n$. In this case, condition equation 12 reduces to*

$$z^\top \bar{A}^{\Delta-1} \bar{d}_{t_1} \leq \sum_{t=0}^{\Delta-2} \left| z^\top \bar{A}^t \bar{d}_{t_1} \right|, \quad \forall z \in \mathbb{R}^n. \tag{13}$$

*Condition equation 13 leads to a better sufficient condition than that in Yalcin et al. (2023). To illustrate the improvement, we consider the special case when the ground truth matrix is $\bar{A} = \lambda I_n$ for some $\lambda \in \mathbb{R}$. Then, condition equation 13 becomes*

$$|\lambda|^{\Delta-1} \leq \sum_{t=0}^{\Delta-2} |\lambda|^t = \frac{1 - |\lambda|^{\Delta-1}}{1 - |\lambda|} \quad \text{, which is further equivalent to } |\lambda| + |\lambda|^{1-\Delta} \leq 2,$$

*which is a stronger condition than that in Yalcin et al. (2023). When the attack period $\Delta$ is large, we approximately have $|\lambda| \leq 2 - 2^{1-\Delta}$, which is a better condition than that in Figure 1 of Yalcin et al. (2023).*

**Example 3** (First-order linear systems). *In the case when $m = n = 1$ and $f(x) = x$, our results state that the uniqueness of global solutions is equivalent to*

$$\left| \sum_{t \in \mathcal{K}} \hat{d}_t x_t \right| < \sum_{t \in \mathcal{K}^c} |x_t|. \tag{14}$$

*As a comparison, the sufficient condition in Theorem 1 in Feng & Lavaei (2021) is*

$$\sum_{t \in \mathcal{K}} |x_t| < \sum_{t \in \mathcal{K}^c} |x_t|.$$

*Since $|\hat{d}_t| = 1$ for all $t \in \mathcal{K}$, our results equation 14, as well as Theorem 2, are more general and stronger than that in Feng & Lavaei (2021).*

## C FUTURE WORKS

One potential future direction is to study the case when there exists dense but small noises in the observations of $x_t$. Our analysis can be naturally extended to this case if an upper bound on the noise scale is assumed. In this work, we mainly focus on large but sparse attacks to exhibit the relation between the sample complexity and the attacks. To provide an intuitive explanation, first assume that the small and dense noise $\xi_t$ is zero. The Lasso-type estimator equation 2 can be written as a constrained optimization problem, where each equation

$$x_{t+1} - Af(x_t) - d_t = 0$$

appears as a constraint. We have derived conditions under which the optimal solution is the correct parameters of the system. Adding $\xi_t$ is essentially equivalent to a perturbation to the constraints of an optimization problem. It is easy to measure how much the optimal solution changes when there is a right-hand side uncertainty. The bound is easy to derive and depends on a given upper bound on the magnitude of $\xi_t$. This relies on classic results in optimization. Moreover, it is possible to improve the sample complexity by injecting small noise into the system dynamics. Intuitively, the injected noise accelerates the "exploration" of $f(x_t)$ in the basis space. This claim can be rigorously proved by utilizing the same techniques as in the paper; see Section V of Yalcin et al. (2023) for an example of the linear system identification problem.

The extension to more general parameterized dynamical systems is another important future direction. The theoretical challenge of the generalization lies in the fact that more complex models, such as generative language models, do not use linear parameterization equation 2 The optimality conditions for deep neural networks are still vague without additional assumptions. This work serves as a first step towards understanding non-linearly parameterized dynamical systems.

## D PROOFS

### D.1 PROOF OF THEOREM 1

*Proof of Theorem 1.* Since problem equation 3 is convex in $A$, the ground truth matrix $\bar{A}$ is a global optimum if and only if

$$0 \in \sum_{t \in \mathcal{K}^c} f(x_t) \otimes \partial \|\mathbf{0}_n\|_2 + \sum_{t \in \mathcal{K}} f(x_t) \otimes \hat{d}_t. \tag{15}$$

Using the form of the subgradient of the $\ell_2$-norm, condition equation 15 holds if and only if there exist vectors

$$g_t \in \mathbb{R}^n, \quad \forall t \in \mathcal{K}^c$$

such that

$$\sum_{t \in \mathcal{K}^c} f(x_t) g_t^\top + \sum_{t \in \mathcal{K}} f(x_t) \hat{d}_t^\top = \mathbf{0}_{n \times n}, \quad \|g_t\|_2 \le 1, \quad \forall t \in \mathcal{K}^c. \tag{16}$$

Define the matrices

$$B := [f(x_t) \quad \forall t \in \mathcal{K}^c] \in \mathbb{R}^{m \times (T - |\mathcal{K}|)}, \quad V := [f(x_t) \quad \forall t \in \mathcal{K}] \in \mathbb{R}^{m \times |\mathcal{K}|},$$

$$G := [g_t \quad \forall t \in \mathcal{K}^c] \in \mathbb{R}^{n \times (T - |\mathcal{K}|)}, \quad F := [\hat{d}_t \quad \forall t \in \mathcal{K}] \in \mathbb{R}^{n \times |\mathcal{K}|}.$$

Condition equation 16 can be written as a combination of second-order cone constraints and linear constraints:

$$\exists G \in \mathbb{R}^{n \times (T - |\mathcal{K}|)}, s, r \in \mathbb{R} \quad \text{s.t. } BG^\top + VF^\top = \mathbf{0}_{m \times n}, \quad \|G_{:,t}\|_2 \leq s, \ \forall t,$$
$$s + r = 1, \quad s, r \geq 0, \tag{17}$$

where $G_{:,t}$ is the $t$-th column of $G$ for all $t \in \{1, \ldots, T - |\mathcal{K}|\}$. We define the closed convex cone

$$\mathcal{S} := \left\{ z \in \mathbb{R}^{(T - |\mathcal{K}|)n + 2} \middle| \sqrt{\sum_{i=1}^{n} z_{(T - |\mathcal{K}|)i + t}^2} \leq z_{(T - |\mathcal{K}|)n + 1}, \ \forall t \in \{0, \ldots, T - |\mathcal{K}| - 1\}, \right.$$

$$\left. z_{(T - |\mathcal{K}|)n + 1}, z_{(T - |\mathcal{K}|)n + 2} \geq 0 \right\},$$

and we define the matrix and vector

$$\mathcal{A} := \begin{bmatrix} I_n \otimes B & 0 & 0 \\ 0 & 1 & 1 \end{bmatrix} \in \mathbb{R}^{(mn+1) \times [(T - |\mathcal{K}|)n + 2]}, \quad b := \begin{bmatrix} -(VF^\top)_{:,1} \\ -(VF^\top)_{:,2} \\ \vdots \\ -(VF^\top)_{:,n} \\ 1 \end{bmatrix} \in \mathbb{R}^{mn+1},$$

where $(VF^\top)_{:,i}$ is the $i$-th column of $VF^\top$. Then, condition equation 17 can be equivalently written as

$$\exists z \in \mathbb{R}^{(T - |\mathcal{K}|)n + 2} \quad \text{s.t. } \mathcal{A}z = b, \quad z \in \mathcal{S}. \tag{18}$$

Since the cone $\mathcal{S}$ is closed and convex, we can apply the *generalized Farka's lemma* to conclude that condition equation 18 is equivalent to

$$\forall y \in \mathbb{R}^{mn+1}, \quad \left( \mathcal{A}^\top y \in \mathcal{S}^* \implies b^\top y \geq 0 \right), \tag{19}$$

where $\mathcal{S}^*$ is the dual cone of $\mathcal{S}$. It can be verified that the dual cone is

$$\mathcal{S}^* = \left\{ z \in \mathbb{R}^{(T - |\mathcal{K}|)n + 2} \middle| \sum_{t=0}^{T - |\mathcal{K}| - 1} \sqrt{\sum_{i=1}^{n} z_{(T - |\mathcal{K}|)i + t}^2} \leq z_{(T - |\mathcal{K}|)n + 1}, \right.$$

$$\left. z_{(T - |\mathcal{K}|)n + 1}, z_{(T - |\mathcal{K}|)n + 2} \geq 0 \right\}.$$

We can equivalently write condition equation 19 as

$$\forall Z \in \mathbb{R}^{n \times m}, \ p \in \mathbb{R}, \quad \left( \|ZB\|_{2,1} \leq p, \quad p \geq 0 \implies \langle VF^\top, Z^\top \rangle \leq p \right),$$

By eliminating variable $p$, we get

$$\langle VF^\top, Z^\top \rangle \leq \|ZB\|_{2,1}, \quad \forall Z \in \mathbb{R}^{n \times m},$$

where the $\ell_{2,1}$-norm is defined as

$$\|M\|_{2,1} := \sum_{j=1}^{n} \sqrt{\sum_{i=1}^{m} M_{ij}^2}, \quad \forall M \in \mathbb{R}^{n \times m}.$$

The above condition is equivalent to condition equation 4, and this completes the proof. $\qquad \square$

## D.2 PROOF OF COROLLARY 1

*Proof of Corollary 1.* The sufficient condition follows from the fact that $\|\hat{d}_t\|_2 = 1$ and

$$\hat{d}_t^\top Z f(x_t) \le \|Z f(x_t)\|_2, \quad \forall t \in \mathcal{K}.$$

This completes the proof. □

## D.3 PROOF OF COROLLARY 2

*Proof of Corollary 2.* We choose

$$Z := \frac{\sum_{t \in \mathcal{K}} \hat{d}_t f(x_t)^\top}{\left\|\sum_{t \in \mathcal{K}} \hat{d}_t f(x_t)^\top\right\|_F}.$$

Then, condition equation 4 implies

$$\left\|\sum_{t \in \mathcal{K}} f(x_t)\hat{d}_t^\top\right\|_F = \sum_{t \in \mathcal{K}} \hat{d}_t^\top Z f(x_t) \le \sum_{t \in \mathcal{K}^c} \|Z f(x_t)\|_2 \le \sum_{t \in \mathcal{K}^c} \|f(x_t)\|_2,$$

where the last step is because $\|Z\|_2 \le \|Z\|_F = 1$. Now, suppose that the basis dimension is $m = 1$. In this case, we have

$$\sum_{t \in \mathcal{K}} \hat{d}_t^\top Z f(x_t) = \left(\sum_{t \in \mathcal{K}} f(x_t)\hat{d}_t\right)^\top Z^\top \le \left\|\sum_{t \in \mathcal{K}} f(x_t)\hat{d}_t\right\|_F \|Z\|_2,$$

$$\sum_{t \in \mathcal{K}^c} \|Z f(x_t)\|_2 = \sum_{t \in \mathcal{K}^c} |f(x_t)|\|Z\|_2 = \sum_{t \in \mathcal{K}^c} \|f(x_t)\|_2 \|Z\|_2.$$

Combining the above two inequalities shows that condition equation 6 is also a sufficient condition. □

## D.4 PROOF OF THEOREM 2

We establish the sufficient and the necessary parts of Theorem 2 by the following two lemmas.

**Lemma 1** (Sufficient condition for uniqueness). *Suppose that condition equation 4 holds. If for every nonzero $Z \in \mathbb{R}^{n \times m}$ such that*

$$\sum_{t \in \mathcal{K}} \hat{d}_t^\top Z f(x_t) = \sum_{t \in \mathcal{K}^c} \|Z f(x_t)\|_2,$$

*it holds that*

$$\sum_{t \in \mathcal{K}} \left|\hat{d}_t^\top Z f(x_t)\right| < \sum_{t \in \mathcal{K}} \|Z f(x_t)\|_2.$$

*Then, the ground truth matrix $\bar{A}$ is the unique global solution to problem equation 3.*

*Proof.* The ground truth $\bar{A}$ is the unique solution if and only if for every matrix $A \in \mathbb{R}^{n \times m}$ such that $A \ne \bar{A}$, the loss function of $A$ is larger than that of $\bar{A}$, namely,

$$\sum_{t \in \mathcal{K}} \|\bar{d}_t\|_2 < \sum_{t \in \mathcal{K}^c} \|(\bar{A} - A)f(x_t)\|_2 + \sum_{t \in \mathcal{K}} \|(\bar{A} - A)f(x_t) + \bar{d}_t\|_2. \tag{20}$$

Denote

$$Z := A - \bar{A} \in \mathbb{R}^{n \times m}.$$

The inequality equation 20 becomes

$$\sum_{t \in \mathcal{K}^c} \| - Z f(x_t)\|_2 + \sum_{t \in \mathcal{K}} \left(\| - Z f(x_t) + \bar{d}_t\|_2 - \|\bar{d}_t\|_2\right) > 0. \tag{21}$$

Since problem equation 3 is convex in $A$, it is sufficient to guarantee that $\bar{A}$ is a strict local minimum. Therefore, the uniqueness of global solutions can be formulated as

$$\text{condition equation 21 holds}, \quad \forall Z \in \mathbb{R}^{n \times m} \quad \text{s.t. } 0 < \|Z\|_F \le \epsilon, \tag{22}$$

where $\epsilon > 0$ is a sufficiently small constant. In the following, we fix the direction $Z$ and discuss two different cases.

**Case I.**   We first consider the case when condition equation 4 holds strictly, namely,

$$\sum_{t \in \mathcal{K}^c} \|Zf(x_t)\|_2 - \sum_{t \in \mathcal{K}} \hat{d}_t^\top Zf(x_t) > 0.$$

Since the $\ell_2$-norm is a convex function, it holds that

$$\| - Zf(x_t) + \bar{d}_t\|_2 - \|\bar{d}_t\|_2 \geq \left\langle \partial \|\bar{d}_t\|_2, -Zf(x_t) \right\rangle = -\hat{d}_t^\top Zf(x_t).$$

Therefore, we get

$$\sum_{t \in \mathcal{K}^c} \| - Zf(x_t)\|_2 + \sum_{t \in \mathcal{K}} \left( \| - Zf(x_t) + \bar{d}_t\|_2 - \|\bar{d}_t\|_2 \right)$$

$$\geq \sum_{t \in \mathcal{K}^c} \| - Zf(x_t)\|_2 + \sum_{t \in \mathcal{K}} -\hat{d}_t^\top Zf(x_t) > 0,$$

which exactly leads to inequality equation 21.

**Case II.**   Next, we consider the case when

$$\sum_{t \in \mathcal{K}} \hat{d}_t^\top Zf(x_t) = \sum_{t \in \mathcal{K}^c} \|Zf(x_t)\|_2, \quad \sum_{t \in \mathcal{K}} \left| \hat{d}_t^\top Zf(x_t) \right| < \sum_{t \in \mathcal{K}} \|Zf(x_t)\|_2. \tag{23}$$

Since $\epsilon$ is a sufficiently small constant, we know

$$\bar{d}_t^\alpha := -\alpha Zf(x_t) + \bar{d}_t \neq 0, \quad \forall \alpha \in [0, 1],$$

and the $\ell_2$-norm is second-order continuously differentiable in an open set that contains the line. Therefore, the *mean value theorem* implies that there exists $\alpha \in [0, 1]$ such that for each $t \in \mathcal{K}$, it holds

$$\| - Zf(x_t) + \bar{d}_t\|_2 - \|\bar{d}_t\|_2 = \left\langle \hat{d}_t, -Zf(x_t) \right\rangle \tag{24}$$

$$+ \frac{1}{2} \left[ -Zf(x_t) \right]^\top \left( \frac{I}{\|\bar{d}_t^\alpha\|_2} - \frac{\bar{d}_t^\alpha \left( \bar{d}_t^\alpha \right)^\top}{\|\bar{d}_t^\alpha\|_2^3} \right) \left[ -Zf(x_t) \right].$$

We can calculate that

$$\left[ -Zf(x_t) \right]^\top \left( \frac{I}{\|\bar{d}_t^\alpha\|_2} - \frac{\bar{d}_t^\alpha \left( \bar{d}_t^\alpha \right)^\top}{\|\bar{d}_t^\alpha\|_2^3} \right) \left[ -Zf(x_t) \right] \tag{25}$$

$$= \frac{\|Zf(x_t)\|_2^2}{\|\bar{d}_t^\alpha\|_2} - \frac{\left\langle \bar{d}_t^\alpha, Zf(x_t) \right\rangle^2}{\|\bar{d}_t^\alpha\|_2^3} \geq 0,$$

where the equality holds if and only if $Zf(x_t)$ is parallel with $\bar{d}_t^\alpha$. By the definition of $\bar{d}_t^\alpha$, the equality holds if and only if $Zf(x_t)$ is parallel with $\bar{d}_t$, which is further equivalent to

$$\left| \left\langle \hat{d}_t, Zf(x_t) \right\rangle \right| = \|Zf(x_t)\|_2.$$

Substituting equation 24 and equation 25 into equation 21, we have

$$\sum_{t \in \mathcal{K}^c} \| - Zf(x_t)\|_2 + \sum_{t \in \mathcal{K}} \left( \| - Zf(x_t) + \bar{d}_t\|_2 - \|\bar{d}_t\|_2 \right)$$

$$\geq \sum_{t \in \mathcal{K}^c} \|Zf(x_t)\|_2 - \sum_{t \in \mathcal{K}} \left\langle \hat{d}_t, Zf(x_t) \right\rangle = 0,$$

where the equality holds if and only if

$$\left| \left\langle \hat{d}_t, Zf(x_t) \right\rangle \right| = \|Zf(x_t)\|_2, \quad \forall t \in \mathcal{K}.$$

Considering the second condition in equation 23, the above equality condition is violated by some $t \in \mathcal{K}$. Therefore, we have proven that condition equation 21 holds strictly.

Combining the two cases, we complete the proof.    □

Next, we prove that the condition in Lemma 1 is also necessary for the uniqueness.

**Lemma 2** (Necessary condition for uniqueness). *Suppose that condition equation 4 holds. If the ground truth matrix $\bar{A}$ is the unique global solution to problem equation 3, then for every nonzero $Z \in \mathbb{R}^{n \times m}$, we have*

$$\sum_{t \in \mathcal{K}} \hat{d}_t^\top Z f(x_t) < \sum_{t \in \mathcal{K}^c} \|Z f(x_t)\|_2 \quad or \quad \sum_{t \in \mathcal{K}} \left| \hat{d}_t^\top Z f(x_t) \right| < \sum_{t \in \mathcal{K}} \|Z f(x_t)\|_2. \tag{26}$$

*Proof.* Assume conversely that there exists a nonzero $Z \in \mathbb{R}^{n \times m}$ such that

$$\sum_{t \in \mathcal{K}} \hat{d}_t^\top Z f(x_t) = \sum_{t \in \mathcal{K}^c} \|Z f(x_t)\|_2, \quad \sum_{t \in \mathcal{K}} \left| \hat{d}_t^\top Z f(x_t) \right| = \sum_{t \in \mathcal{K}} \|Z f(x_t)\|_2. \tag{27}$$

Without loss of generality, we assume that

$$0 < \|Z\|_2 \le \epsilon$$

for a sufficiently small $\epsilon$. In this case, the second condition in equation 27 implies that

$$\left| \hat{d}_t^\top Z f(x_t) \right| = \|Z f(x_t)\|_2, \text{ and } \quad Z f(x_t) \text{ is parallel with } \bar{d}_t, \quad \forall t \in \mathcal{K}.$$

Therefore, when $\epsilon$ is sufficiently small, equations equation 25 and equation 23 lead to

$$\| - Z f(x_t) + \bar{d}_t \|_2 - \|\bar{d}_t\|_2 = - \left\langle \hat{d}_t, Z f(x_t) \right\rangle, \quad \forall t \in \mathcal{K}.$$

We now show that condition equation 21 fails:

$$\sum_{t \in \mathcal{K}^c} \| - Z f(x_t) \|_2 + \sum_{t \in \mathcal{K}} \left( \| - Z f(x_t) + \bar{d}_t \|_2 - \|\bar{d}_t\|_2 \right)$$

$$= \sum_{t \in \mathcal{K}} \left\langle \hat{d}_t, Z f(x_t) \right\rangle - \sum_{t \in \mathcal{K}} \left\langle \hat{d}_t, Z f(x_t) \right\rangle = 0.$$

This contradicts with the assumption that $\bar{A}$ is the unique solution to problem equation 3. $\qquad\square$

Combining Lemmas 1 and 2, we have the following necessary and sufficient condition for the uniqueness of the ground truth solution $\bar{A}$.

### D.5 PROOF OF THEOREM 4

*Proof of Theorem 4.* Since both sides of inequality equation 8 are affine in $Z$, it suffices to prove that

$$\mathbb{P} \left[ \hat{d}_1(Z) - \hat{d}_2(Z) < 0, \ \forall Z \in \mathbb{S}_F \right] \ge 1 - \delta, \tag{28}$$

where $\mathbb{S}_F$ is the Frobenius-norm unit sphere in $\mathbb{R}^{n \times m}$ and

$$\hat{d}_1(Z) := \sum_{t \in \mathcal{K}} \langle Z^\top, f(x_t) \hat{d}_t^\top \rangle, \quad \hat{d}_2(Z) := \sum_{t \in \mathcal{K}^c} \|Z f(x_t)\|_2.$$

The proof is divided into two steps.

**Step 1.** First, we fix the vector $Z \in \mathbb{S}_F$ and prove that

$$\mathbb{P} \left[ \hat{d}_1(Z) - \hat{d}_2(Z) < -\theta \right] \ge 1 - \delta,$$

holds for some constant $\theta > 0$. Using Markov's inequality, it is sufficient to prove that for some $\nu > 0$, it holds that

$$\mathbb{E} \left[ \exp \left( \nu \left[ \hat{d}_1(Z) - \hat{d}_2(Z) \right] \right) \right] \le \exp(-\nu\theta)\delta. \tag{29}$$

We focus on the case when $\mathcal{K}$ is not empty, which happens with high probability. The proof of this step is also divided into two sub-steps.

**Step 1-1.** We first analyze the term $\hat{d}_1(Z)$. Let $T'$ be the last attack time instance, i.e.,

$$T' := \max\{t \mid t \in \mathcal{K}\}.$$

Then, we have

$$\mathbb{E}\left[\exp\left[\nu\hat{d}_1(Z)\right]\right] = \mathbb{E}\left[\exp\left(\nu\sum_{t\in\mathcal{K}\setminus\{T'\}}\left\langle Z^\top, f(x_t)\hat{d}_t^\top\right\rangle\right) \times \mathbb{E}\left[\exp\left[\nu\left\langle Z^\top, f(x_{T'})\hat{d}_{T'}^\top\right\rangle\right] \mid \mathcal{F}_{T'}\right]\right].$$

$$(30)$$

According to Assumption 1, the direction $\hat{d}_{T'}$ is a unit vector. Since

$$\left|[Zf(x_{T'})]^\top\hat{d}_{T'}\right| \leq \|Zf(x_{T'})\|_2 \leq \|Z\|_2\|f(x_{T'})\|_2$$
$$\leq \|Z\|_F\sqrt{m}\|f(x_{T'})\|_\infty \leq \sqrt{m}B,$$

the random variable $[Zf(x_{T'})]^\top\hat{d}_{T'}$ is sub-Gaussian with parameter $mB^2$. Therefore, the property of sub-Gaussian random variables implies that

$$\mathbb{E}\left[\exp\left[\nu\left\langle Z^\top, f(x_{T'})\hat{d}_{T'}^\top\right\rangle\right] \mid \mathcal{F}_{T'}\right] \leq \exp\left(\frac{\nu^2\cdot mB^2}{2}\right).$$

Substituting into equation 30, we get

$$\mathbb{E}\left[\exp\left[\nu\hat{d}_1(Z)\right]\right] \leq \mathbb{E}\left[\exp\left(\nu\sum_{t\in\mathcal{K}\setminus\{T'\}}\left\langle Z^\top, f(x_t)\hat{d}_t^\top\right\rangle\right)\right]\cdot\exp\left(\frac{\nu^2\cdot mB^2}{2}\right).$$

Continuing this process for all $t \in \mathcal{K}$, it follows that

$$\mathbb{E}\left[\exp\left[\nu\hat{d}_1(Z)\right]\right] \leq \exp\left(\frac{\nu^2\cdot mB^2|\mathcal{K}|}{2}\right).$$

$$(31)$$

**Step 1-2.** Now, we consider the second term in equation 29, namely, $-\hat{d}_2(Z)$. Define

$$\mathcal{K}' := \{t \mid 1 \leq t \leq T,\ t \in \mathcal{K}^c,\ t-1 \in \mathcal{K}\}.$$

With probability at least $1 - \exp[-\Theta[p(1-p)T]]$, we have

$$|\mathcal{K}'| = \Theta[p(1-p)T].$$

Therefore, $\mathcal{K}'$ is non-empty with high-probability. Since $\|Zf(x_t)\|_2 \geq 0$ for all $t \in \mathcal{K}^c$, we have

$$\mathbb{E}\left[\exp\left[-\nu\hat{d}_2(Z)\right]\right] \leq \mathbb{E}\left[\exp\left(-\nu\sum_{t\in\mathcal{K}'}\|Zf(x_t)\|_2\right)\right] \qquad (32)$$

$$= \mathbb{E}\left[\exp\left(-\nu\sum_{t\in\mathcal{K}'\setminus\{T'\}}\|Zf(x_t)\|_2\right) \times \mathbb{E}\left[\exp\left(-\nu\|Zf(x_{T'})\|_2\right) \mid \mathcal{F}_{T'}\right]\right],$$

where $T'$ is the last time instance in $\mathcal{K}'$, namely,

$$T' := \max\{t \mid t \in \mathcal{K}'\}.$$

By Bernstein's inequality Wainwright (2019), we can estimate that

$$\mathbb{E}\left[\exp\left(-\nu\|Zf(x_{T'})\|_2\right) \mid \mathcal{F}_{T'}\right]$$
$$\leq \exp\left[-\nu\mathbb{E}\left(\|Zf(x_{T'})\|_2 \mid \mathcal{F}_{T'}\right) + \frac{\nu^2}{2}\mathbb{E}\left(\|Zf(x_{T'})\|_2^2 \mid \mathcal{F}_{T'}\right)\right]$$
$$\leq \exp\left[-\frac{\nu}{\sqrt{m}B}\mathbb{E}\left(\|Zf(x_{T'})\|_2^2 \mid \mathcal{F}_{T'}\right) + \frac{\nu^2}{2}\mathbb{E}\left(\|Zf(x_{T'})\|_2^2 \mid \mathcal{F}_{T'}\right)\right],$$

where the last inequality is from

$$\|Zf(x_{T'})\|_2 \leq \sqrt{m}B.$$

Assumption 3 implies that

$$\mathbb{E}\left(\|Zf(x_{T'})\|_2^2 \mid \mathcal{F}_{T'}\right) = \left\langle ZZ^\top, \mathbb{E}\left[f(x_{T'})f(x_{T'})^\top \mid \mathcal{F}_{T'}\right]\right\rangle \geq \lambda^2 \|Z\|_F^2 = \lambda^2.$$

If we choose $\nu$ such that

$$0 < \nu < \frac{2}{\sqrt{m}B}, \tag{33}$$

we have

$$\mathbb{E}\left[\exp\left(-\nu\|Zf(x_{T'})\|_2\right) \mid \mathcal{F}_{T'}\right] \leq \exp\left[\left(\frac{\nu^2}{2} - \frac{\nu}{\sqrt{m}B}\right)\lambda^2\right].$$

Substituting into inequality equation 32, it follows that

$$\mathbb{E}\left[\exp\left[-\nu\hat{d}_2(Z)\right]\right]$$

$$\leq \mathbb{E}\left[\exp\left(-\nu\sum_{t\in\mathcal{K}'\setminus\{T'\}}\|Zf(x_t)\|_2\right) \times \exp\left[\left(\frac{\nu^2}{2} - \frac{\nu}{\sqrt{m}B}\right)\lambda^2\right]\right].$$

Continuing this process for all $t \in \mathcal{K}'$, we have

$$\mathbb{E}\left[\exp\left[-\nu\hat{d}_2(Z)\right]\right] \leq \exp\left[\left(\frac{\nu^2}{2} - \frac{\nu}{\sqrt{m}B}\right)\lambda^2|\mathcal{K}'|\right]. \tag{34}$$

Combining the inequalities equation 31 and equation 34, we have

$$\mathbb{E}\left[\exp\left(\nu\left[\hat{d}_1(Z) - \hat{d}_2(Z)\right]\right)\right] \leq \exp\left[\frac{m\nu^2 B^2}{2}|\mathcal{K}| + \left(\frac{\nu^2}{2} - \frac{\nu}{\sqrt{m}B}\right)\lambda^2|\mathcal{K}'|\right].$$

We choose

$$\theta := \frac{\lambda^2 p(1-p)T}{4\sqrt{m}B}.$$

In order to satisfy condition equation 29, it is equivalent to have

$$\frac{m\nu^2 B^2}{2}|\mathcal{K}| + \left(\frac{\nu^2}{2} - \frac{\nu}{\sqrt{m}B}\right)\lambda^2|\mathcal{K}'| + \frac{\lambda^2\nu p(1-p)T}{4\sqrt{m}B} \leq \log(\delta). \tag{35}$$

Now, we consider the fact that $\mathcal{K}$ is generated by the probabilistic attack model. Using the Bernoulli bound, it holds with probability at least $1 - \exp[-\Theta[p(1-p)T]]$ that

$$|\mathcal{K}| \leq 2pT, \quad |\mathcal{K}'| \geq \frac{p(1-p)T}{2}. \tag{36}$$

Thus, with the same probability, we have the estimation

$$\frac{m\nu^2 B^2}{2}|\mathcal{K}| + \left(\frac{\nu^2}{2} - \frac{\nu}{\sqrt{m}B}\right)\lambda^2|\mathcal{K}'| + \frac{\lambda^2\nu p(1-p)T}{4\sqrt{m}B}$$

$$\leq \frac{m\nu^2 B^2}{2} \cdot 2pT + \left(\frac{\nu^2}{2} - \frac{\nu}{2\sqrt{m}B}\right)\lambda^2 \cdot \frac{p(1-p)T}{2}.$$

Choosing

$$\nu := \frac{\lambda^2(1-p)}{2\sqrt{m}B[4mB^2 + \lambda^2(1-p)]},$$

we get

$$\frac{m\nu^2 B^2}{2}|\mathcal{K}| + \left(\frac{\nu^2}{2} - \frac{\nu}{\sqrt{m}B}\right)\lambda^2|\mathcal{K}'| + \frac{\lambda^2\nu p(1-p)T}{4\sqrt{m}B} \leq -\frac{p(1-p)^2}{16m\kappa^2(4m\kappa^2 + 1 - p)} \cdot T,$$

where we define $\kappa := B/\lambda \geq 1$. Note that our choice of $\nu$ satisfies the condition equation 33. Therefore, in order for inequality equation 35 to hold, the sample complexity should satisfy

$$T \geq \frac{16m\kappa^2(4m\kappa^2 + 1 - p)}{p(1-p)^2} \log\left(\frac{1}{\delta}\right).$$

By considering the Bernoulli bound equation 36, the sample complexity bound becomes

$$T \geq \Theta\left[\max\left\{\frac{m\kappa^2(m\kappa^2 + 1 - p)}{p(1-p)^2}, \frac{1}{p(1-p)}\right\} \log\left(\frac{1}{\delta}\right)\right] \tag{37}$$

$$= \Theta\left[\frac{m^2\kappa^4}{p(1-p)^2} \log\left(\frac{1}{\delta}\right)\right].$$

**Step 2.** Next, we establish the bound equation 28 by discretization techniques. More specifically, suppose that $\epsilon > 0$ is a constant and $\{Z^1, \ldots, Z^N\} \subset \mathbb{S}_F$ is an $\epsilon$-net of the sphere $\mathbb{S}_F$ under the Frobenius norm, where we can bound

$$\log(N) \leq mn \cdot \log\left(1 + \frac{2}{\epsilon}\right).$$

Then, for every $Z \in \mathbb{S}_F$, we can find a point in the $\epsilon$-net, denoted as $Z'$, such that

$$\|Z - Z'\|_F \leq \epsilon.$$

Now, we upper bound the difference $f(Z) - f(Z')$, where we define the function

$$f(Z) := \hat{d}_1(Z) - \hat{d}_2(Z), \quad \forall Z \in \mathbb{R}^{n \times m}.$$

We can calculate that

$$f(Z) - f(Z') = \sum_{t \in \mathcal{K}} \hat{d}_t(Z - Z')f(x_t) - \sum_{t \in \mathcal{K}^c} \left(\|Zf(x_t)\|_2 - \|Z'f(x_t)\|_2\right)$$

$$\leq \sum_{t \in \mathcal{K}} \hat{d}_t(Z - Z')f(x_t) + \sum_{t \in \mathcal{K}^c} \|(Z - Z')f(x_t)\|_2$$

$$\leq \sum_{t \in \mathcal{K}} \|Z - Z'\|_F \|f(x_t)\hat{d}_t^\top\|_F + \sum_{t \in \mathcal{K}^c} \|Z - Z'\|_2 \|f(x_t)\|_2$$

$$\leq \sum_{t \in \mathcal{K}} \|Z - Z'\|_F \|f(x_t)\|_2 + \sum_{t \in \mathcal{K}^c} \|Z - Z'\|_F \|f(x_t)\|_2$$

$$\leq T \cdot \epsilon\sqrt{m}B = \sqrt{m}TB \cdot \epsilon.$$

We choose

$$\epsilon := \frac{\theta}{\sqrt{m}TB} = \Theta\left[\frac{p(1-p)}{m\kappa^2}\right].$$

Therefore, under the event that

$$f(Z^i) < -\theta, \quad \forall i = 1, \ldots, N, \tag{38}$$

we have

$$f(Z) < -\theta + \sqrt{m}TB \cdot \epsilon = 0, \quad \forall Z \in \mathbb{S}_F.$$

Hence, it suffices to estimate the probability that event equation 38 happens. To bound the failing probability, we replace $\delta$ with $\delta/N$ in equation 37 and it follows that

$$\mathbb{P}\left[f(Z^i) < -\theta\right] \geq 1 - \frac{\delta}{N}, \quad \forall i = 1, \ldots, N.$$

Applying the union bound over all $i \in \{1, \ldots, N\}$, the event equation 38 happens with probability at least $1 - \delta$, namely,

$$\mathbb{P}\left[f(Z^i) < -\theta, \forall i = 1, \ldots, N\right] \geq 1 - \delta.$$

With this choice of $\delta$, the sample complexity should be at least

$$T \geq \Theta\left[\frac{m^2\kappa^4}{p(1-p)^2} \log\left(\frac{N}{\delta}\right)\right]$$

$$= \Theta\left[\frac{m^2\kappa^4}{p(1-p)^2} \left[mn \log\left(\frac{m\kappa}{p(1-p)}\right) + \log\left(\frac{1}{\delta}\right)\right]\right].$$

This completes the proof. $\qquad\qquad\square$

### D.6 PROOF OF THEOREM 5

*Proof of Theorem 5.* We only need to show that condition equation 7 fails with probability at least $1 - \exp(-m/3)$. We choose the matrix

$$\bar{A} := \begin{bmatrix} 1 & 0_{1 \times (m-1)} \\ 0_{n-1} & 0_{(n-1) \times (m-1)} \end{bmatrix} \in \mathbb{R}^{n \times m}.$$

As a result, the last $n - 1$ elements of $\bar{A}f(x)$ are zero for every state $x \in \mathbb{R}^n$. Moreover, we will choose the basis function $f$ such that its values will only depend on the first element of state $x \in \mathbb{R}^n$. With these definitions, the dynamics of $x_t$ reduces to the dynamics of its first element $(x_t)_1$. Hence, we can assume without loss of generality that $n = 1$ in the remainder of the proof.

We define the basis function $f : \mathbb{R} \mapsto \mathbb{R}^m$ as

$$\tilde{f}(x) := \begin{bmatrix} \frac{x}{\max\{|x|, 1\}} & \sin(x) & \sin(2x) & \cdots & \sin[(m-1)x] \end{bmatrix}, \quad \forall x \in \mathbb{R}.$$

Under the above definitions, it is straightforward to show that the following properties hold and we omit the proof:

$$f(0) = 0_m, \quad f\left[\bar{A}f(x)\right] = f(x), \quad \forall x \in \mathbb{R}. \tag{39}$$

Finally, the attack vector is defined as

$$\bar{d}_t | \mathcal{F}_t \sim \text{Uniform} \left\{[-(|x_t| + 2\pi), -(|x_t| + \pi)] \cup [|x_t| + \pi, |x_t| + 2\pi]\right\}, \quad \forall t \in \mathcal{K}.$$

The remainder of the proof is divided into three steps.

**Step 1.** In the first step, we prove that Assumptions 1-3 hold. By the definition of $f(x)$, we have

$$\|f(x)\|_\infty = \max \left\{ \frac{|x|}{\max\{|x|, 1\}}, |\sin(x)|, \dots, |\sin[(m-1)x]| \right\} \leq 1, \quad \forall x \in \mathbb{R},$$

which implies that Assumption 2 holds with $B = 1$. Moreover, the stealthy condition (Assumption 1) is a result of the symmetric distribution of $\bar{d}_t | \mathcal{F}_t$.

Finally, we prove that Assumption 3 holds. For the notational simplicity, in this step, we omit the subscript $t$, the conditioning on the filtration $\mathcal{F}_t$ and the event $t \in \mathcal{K}$. The model of attack $d$ implies that

$$|x + d| \geq |d| - |x| \geq \pi > 1.$$

Therefore, we have

$$f(x + d) = \begin{bmatrix} \frac{x+d}{|x+d|} & \sin[(x+d)] & \cdots & \sin[(m-1)(x+d)] \end{bmatrix}.$$

For any vector $\nu \in \mathbb{R}^m$, we want to estimate

$$\nu^\top \mathbb{E}\left[f(x+d)f(x+d)^\top\right]\nu = \mathbb{E}\left[\nu_1 \frac{x+d}{|x+d|} + \sum_{i=1}^{m-1} \nu_{i+1} \sin[i(x+d)]\right]^2.$$

First, we can calculate that

$$\mathbb{E}\left(\nu_1 \frac{x+d}{|x+d|}\right)^2 = \nu_1^2, \quad \mathbb{E}\left[\nu_{i+1} \sin[i(x+d)]\right]^2 = \nu_{i+1}^2 \cdot \frac{1}{2}, \quad \forall i \in \{1, \dots, m-1\}. \tag{40}$$

Then, for every $i \in \{1, \dots, m-1\}$, we have

$$\mathbb{E}\left[\nu_1 \frac{x+d}{|x+d|} \cdot \nu_{i+1} \sin[i(x+d)]\right] \tag{41}$$

$$= \nu_1 \nu_{i+1} \left[\int_{-|x|-2\pi}^{-|x|-\pi} \frac{x+d}{|x+d|} \sin[i(x+d)] \, \mathrm{d}d + \int_{|x|+\pi}^{|x|+2\pi} \frac{x+d}{|x+d|} \sin[i(x+d)] \, \mathrm{d}d\right]$$

$$= \nu_1 \nu_{i+1} \left[\int_{-|x|-2\pi}^{-|x|-\pi} -\sin[i(x+d)] \, \mathrm{d}d + \int_{|x|+\pi}^{|x|+2\pi} \sin[i(x+d)] \, \mathrm{d}d\right] = 0.$$

For every $i, j \in \{1, \ldots, m-1\}$ such that $i \neq j$, it holds that

$$\mathbb{E}\left[\nu_{i+1} \sin[i(x+d)] \cdot \nu_{j+1} \sin[j(x+d)]\right] \tag{42}$$

$$=\nu_{i+1}\nu_{j+1}\left[\int_{-|x|-2\pi}^{-|x|-\pi} \sin[i(x+d)] \sin[j(x+d)] \, \mathrm{d}d\right.$$

$$\left.+ \int_{|x|+\pi}^{|x|+2\pi} \sin[i(x+d)] \sin[j(x+d)] \, \mathrm{d}d\right] = 0.$$

Combining equations equation 40-equation 42, it follows that

$$\nu^\top \mathbb{E}\left[f(x+d)f(x+d)^\top\right] \nu = \nu_1^2 + \frac{1}{2}\sum_{i=1}^{m-1} \nu_{i+1}^2 \geq \frac{1}{2}\|\nu\|_2^2,$$

which implies that Assumption 3 holds with $\lambda^2 = 1/2$.

**Step 2.** In this step, we prove that the linear space spanned by the set of vectors

$$\mathcal{F}^c := \{f(x_t) \mid t \in \mathcal{K}^c\}$$

has dimension at most $m-1$ with probability at least $1-\delta$. By the second property in equation 39, the subspace spanned by $\mathcal{F}^c$ is equivalent to that spanned by

$$\mathcal{F}' := \{f(x_t) \mid t \in \mathcal{K}'\},$$

where we define

$$\mathcal{K}' := \{t \mid t-1 \in \mathcal{K}, \ t \in \mathcal{K}^c\}.$$

Therefore, the dimension of the subspace is at most $|\mathcal{K}'|$.

To estimate the cardinality of $\mathcal{K}'$, we divide $\mathcal{K}'$ into the following two disjoint sets:

$$\mathcal{K}_1' := \{2t+1 \mid 2t \in \mathcal{K}, \ 2t+1 \in \mathcal{K}^c\}, \quad \mathcal{K}_2' := \{2t \mid 2t-1 \in \mathcal{K}, \ 2t \in \mathcal{K}^c\}.$$

The size of $\mathcal{K}_1'$ is the summation of $\lceil T/2 \rceil$ independent Bernoulli random variables with parameter $p(1-p)$. Therefore, the Chernoff bound implies

$$\mathbb{P}\left[|\mathcal{K}_1'| \leq 2p(1-p) \cdot \left\lceil\frac{T}{2}\right\rceil\right] \geq 1 - \exp\left[-\frac{p(1-p)}{3} \cdot \left\lceil\frac{T}{2}\right\rceil\right]. \tag{43}$$

Similarly, the size of $\mathcal{K}_2'$ is the summation of $\lfloor T/2 \rfloor$ independent Bernoulli random variables with parameter $p(1-p)$. Therefore, the Chernoff bound implies

$$\mathbb{P}\left[|\mathcal{K}_2'| \leq 2p(1-p) \cdot \left\lfloor\frac{T}{2}\right\rfloor\right] \geq 1 - \exp\left[-\frac{p(1-p)}{3} \cdot \left\lfloor\frac{T}{2}\right\rfloor\right]. \tag{44}$$

Combining the bounds equation 43 and equation 44 and applying the union bound, it holds that

$$\mathbb{P}\left[|\mathcal{K}'| \leq 2p(1-p)T\right] \geq 1 - \exp\left[-\frac{p(1-p)}{3} \cdot \left\lceil\frac{T}{2}\right\rceil\right] - \exp\left[-\frac{p(1-p)}{3} \cdot \left\lfloor\frac{T}{2}\right\rfloor\right]$$

$$\geq 1 - 2\exp\left[-\frac{p(1-p)T}{3}\right],$$

where the last inequality is because $\lfloor T/2 \rfloor \leq \lceil T/2 \rceil \leq T$. Since

$$T < \frac{m}{2p(1-p)},$$

we know

$$\mathbb{P}\left[|\mathcal{K}'| < m\right] \geq 1 - 2\exp\left(-m/3\right). \tag{45}$$

In addition, when $\mathcal{K}$ is the empty set $\emptyset$ or the full set $\{0, \ldots, T-1\}$, the set $\mathcal{K}'$ is an empty set, which implies that $|\mathcal{K}'|$ is smaller than $m$. This event happens with probability

$$p^\top + (1-p)^\top \geq 2[p(1-p)]^{T/2}.$$

Combining with inequality equation 45, we get

$$\mathbb{P}\left[|\mathcal{K}'| < m\right] \geq \max\left\{1 - 2\exp\left(-m/3\right), 2[p(1-p)]^{T/2}\right\}.$$

**Step 3.** Finally, we prove that if the dimension of the subspace spanned by $\mathcal{F}^c$ is smaller than $m$, the condition equation 7 cannot hold. Since the dimension of the subspace is at most $m - 1$, there exists $Z \in \mathbb{R}^m$ such that

$$Z f(x_t) = 0, \quad \forall t \in \mathcal{K}^c.$$

With this choice of $Z$, the condition on the left hand-side of equation 7 holds while the strict inequality on the right hand-side fails. Therefore, we know that $\bar{A}$ is not the unique global solution to equation 3. $\qquad \square$

### D.7 PROOF OF THEOREM 6

*Proof of Theorem 6.* The proof is similar to that of Theorem 4. Since both sides of inequality equation 8 are affine in $Z$, it suffices to prove that

$$\mathbb{P}\left[\hat{d}_1(Z) - \hat{d}_2(Z) < 0, \ \forall Z \in \mathbb{S}_F\right] \geq 1 - \delta,$$

where $\mathbb{S}_F$ is the Frobenius-norm unit sphere in $\mathbb{R}^{n \times m}$ and

$$\hat{d}_1(Z) := \sum_{t \in \mathcal{K}} \left\langle Z^\top, f(x_t)\hat{d}_t^\top \right\rangle, \quad \hat{d}_2(Z) := \sum_{t \in \mathcal{K}^c} \|Z f(x_t)\|_2.$$

The proof is divided into two steps.

**Step 1.** First, we fix the vector $Z \in \mathbb{S}_F$ and prove that

$$\mathbb{P}\left[\hat{d}_1(Z) - \hat{d}_2(Z) < -\theta\right] \geq 1 - \delta,$$

holds for some constant $\theta > 0$. The proof of this step is divided into two steps.

**Step 1-1.** We first analyze the term $\hat{d}_1(Z)$. For each $k \in \mathcal{K}$, we define the following attack vectors:

$$\bar{d}_t^k := \begin{cases} \bar{d}_t & \text{if } t \leq k, \\ 0_n & \text{otherwise,} \end{cases} \quad \forall t \in \{0, \ldots, T-1\}.$$

Then, we define the trajectory generated by the above attack vectors:

$$x_0^k = 0_m, \quad x_{t+1}^k = \bar{A}f(x_t^k) + \bar{d}_t^k, \quad \forall t \in \{0, \ldots, T-1\}.$$

Let

$$\mathcal{K} = \{k_1, \ldots, k_{|\mathcal{K}|}\},$$

where the elements are sorted as $k_1 < k_2 < \cdots < k_{|\mathcal{K}|}$. Under the above definition, we know $x_t^{k_{|\mathcal{K}|}} = x_t$ for all $t$. We define

$$g_t^{k_j} := \begin{cases} f(x_t^{k_j}) - f(x_t^{k_{j-1}}) & \text{if } j > 1, \\ f(x_t^{k_1}) & \text{if } j = 1, \end{cases} \quad \forall j \in \{1, \ldots, |\mathcal{K}|\}.$$

We note that $g_t^{k_j}$ is measurable on $\mathcal{F}_{k_j}$. Using these introduced notations, we can write $\hat{d}_1(Z)$ as

$$\hat{d}_1(Z) = \sum_{j=1}^{|\mathcal{K}|} \left\langle Z^\top, f(x_{k_j})\hat{d}_{k_j}^\top \right\rangle = \sum_{j=1}^{|\mathcal{K}|} \left\langle Z^\top, \sum_{\ell=1}^{j-1} g_{k_j}^{k_\ell} \hat{d}_{k_j}^\top \right\rangle = \sum_{\ell=1}^{|\mathcal{K}|} \sum_{j=\ell+1}^{|\mathcal{K}|} \hat{d}_{k_j}^\top Z g_{k_j}^{k_\ell}.$$

Then, Assumption 6 implies that $\bar{d}_t$ is sub-Gaussian with parameter $\sigma$ conditional on $\mathcal{F}_t$. Now, we estimate the expectation

$$\mathbb{E}\left[\exp\left[\nu \hat{d}_1(Z)\right]\right],$$

where $\nu \in \mathbb{R}$ is an arbitrary constant. First, for each $\ell \in \{1, \ldots, |\mathcal{K}| - 1\}$, we estimate the following probability:

$$\mathbb{P}\left(\left|\sum_{j=\ell+1}^{|\mathcal{K}|} \hat{d}_{k_j}^\top Z g_{k_j}^{k_\ell}\right| \geq \epsilon \ \middle| \ \mathcal{F}_{k_\ell}\right).$$

Since $\hat{d}_{k_j}$ is a unit vector and $\|Z\|_F = 1$, we know

$$\left\| \hat{d}_{k_j}^\top Z \right\|_2 \leq \|\hat{d}_{k_j}^\top\|_2 \|Z\|_2 \leq \|\hat{d}_{k_j}^\top\|_2 \|Z\|_F = 1. \tag{46}$$

Moreover, we can estimate that

$$\begin{aligned}
\left\| g_{k_j}^{k_\ell} \right\|_2 &= \left\| f(x_{k_j}^{k_\ell}) - f(x_{k_j}^{k_{\ell-1}}) \right\|_2 \leq L \left\| x_{k_j}^{k_\ell} - x_{k_j}^{k_{\ell-1}} \right\|_2 \\
&= L \left\| \bar{A} \left[ f\left( x_{k_j-1}^{k_\ell} \right) - f\left( x_{k_j-1}^{k_{\ell-1}} \right) \right] \right\|_2 \leq \rho L \left\| f\left( x_{k_j-1}^{k_\ell} \right) - f\left( x_{k_j-1}^{k_{\ell-1}} \right) \right\|_2 \\
&\leq L(\rho L) \left\| x_{k_j-1}^{k_\ell} - x_{k_j-1}^{k_{\ell-1}} \right\|_2 \leq \cdots \leq L(\rho L)^{k_j-k_\ell-1} \left\| x_{k_\ell+1}^{k_\ell} - x_{k_\ell+1}^{k_{\ell-1}} \right\|_2 \\
&= L(\rho L)^{k_j-k_\ell-1} \|\bar{d}_{k_\ell}\|_2,
\end{aligned} \tag{47}$$

where the first inequality holds because $f$ has Lipschitz constant $L$, the second inequality is from $\|\bar{A}\|_2 \leq \rho$ and the last equality holds because

$$x_{k_\ell+1}^{k_\ell} = \bar{A} f\left( x_{k_\ell}^{k_\ell} \right) + \bar{d}_{k_\ell}, \quad x_{k_\ell+1}^{k_{\ell-1}} = \bar{A} f\left( x_{k_\ell}^{k_{\ell-1}} \right) = \bar{A} f\left( x_{k_\ell}^{k_\ell} \right).$$

By the sub-Gaussian assumption (Assumption 6), it holds that

$$\mathbb{P}\left( \|\bar{d}_{k_\ell}\|_2 \geq \eta \,\Big|\, \mathcal{F}_{k_\ell} \right) \leq 2 \exp\left( -\frac{\eta^2}{2\sigma^2} \right), \quad \forall \eta \geq 0. \tag{48}$$

Combining inequalities equation 46-equation 48, we get

$$\begin{aligned}
\mathbb{P}\left( \left| \sum_{j=\ell+1}^{|\mathcal{K}|} \hat{d}_{k_j}^\top Z^\top g_{k_j}^{k_\ell} \right| \geq \epsilon \,\Big|\, \mathcal{F}_{k_\ell} \right) &\leq \mathbb{P}\left( \sum_{j=\ell+1}^{|\mathcal{K}|} \left\| g_{k_j}^{k_\ell} \right\|_2 \geq \epsilon \,\Big|\, \mathcal{F}_{k_\ell} \right) \\
&\leq \mathbb{P}\left( \sum_{j=\ell+1}^{|\mathcal{K}|} L(\rho L)^{k_j-k_\ell-1} \|\bar{d}_{k_\ell}\|_2 \geq \epsilon \,\Big|\, \mathcal{F}_{k_\ell} \right) \\
&\leq \mathbb{P}\left( \frac{L(\rho L)^{\Delta_j}}{1-\rho L} \|\bar{d}_{k_\ell}\|_2 \geq \epsilon \,\Big|\, \mathcal{F}_{k_\ell} \right) \leq 2\exp\left[ -\frac{(1-\rho L)^2 \epsilon^2}{2\sigma^2 L^2 (\rho L)^{2\Delta_j}} \right],
\end{aligned} \tag{49}$$

where $\Delta_j := k_j - k_{j-1} - 1$ and the second last inequality is from

$$\sum_{j=\ell+1}^{|\mathcal{K}|} L(\rho L)^{k_j-k_\ell-1} < \sum_{i=\Delta_j}^{\infty} L(\rho L)^i = \frac{L(\rho L)^{\Delta_j}}{1-\rho L}.$$

Since

$$\mathbb{E}\left( \sum_{j=\ell+1}^{|\mathcal{K}|} \hat{d}_{k_j}^\top Z g_{k_j}^{k_\ell} \,\Big|\, \mathcal{F}_{k_\ell} \right) = 0,$$

inequality equation 49 implies that the random variable $\sum_{j=\ell+1}^{|\mathcal{K}|} \hat{d}_{k_j}^\top Z^\top g_{k_j}^{k_\ell}$ is zero-mean and sub-Gaussian with parameter $\sigma L/(1-\rho L)$ conditional on $\mathcal{F}_{k_\ell}$. By the property of sub-Gaussian random variables, we have

$$\mathbb{E}\left[ \exp\left( \nu \sum_{j=\ell+1}^{|\mathcal{K}|} \hat{d}_{k_j}^\top Z g_{k_j}^{k_\ell} \right) \,\Big|\, \mathcal{F}_{k_\ell} \right] \leq \exp\left[ \frac{\nu^2 \sigma^2 L^2 (\rho L)^{2\Delta_j}}{2(1-\rho L)^2} \right], \quad \forall \nu \geq 0.$$

Finally, utilizing the tower property of conditional expectation, we have

$$\mathbb{E}\left[\exp\left[\nu\hat{d}_1(Z)\right]\right] = \mathbb{E}\left[\exp\left(\nu\sum_{\ell=1}^{|\mathcal{K}|-2}\sum_{j=\ell+1}^{|\mathcal{K}|}\hat{d}_{k_j}^\top Z g_{k_j}^{k_\ell}\right)\right.$$ (50)

$$\times \mathbb{E}\left[\exp\left(\nu\sum_{j=|\mathcal{K}|}^{|\mathcal{K}|}\hat{d}_{k_j}^\top Z g_{k_j}^{k_\ell}\right)\ \Bigg|\ \mathcal{F}_{k_{|\mathcal{K}|-1}}\right]\Bigg]$$

$$\leq \mathbb{E}\left[\exp\left(\nu\sum_{\ell=1}^{|\mathcal{K}|-2}\sum_{j=\ell+1}^{|\mathcal{K}|}\hat{d}_{k_j}^\top Z g_{k_j}^{k_\ell}\right) \times \exp\left[\frac{\nu^2\sigma^2 L^2(\rho L)^{2\Delta_j}}{2(1-\rho L)^2}\right]\right]$$

$$\leq \cdots \leq \exp\left[\frac{\nu^2\sigma^2 L^2}{2(1-\rho L)^2}\sum_{j\in\mathcal{K}}(\rho L)^{2\Delta_j}\right], \quad \forall \nu \geq 0.$$

Since the random variable $(\rho L)^{\Delta_j}$ is bounded in $[0,1]$ and thus, it is sub-Gaussian with parameter $1/2$. Therefore, with constant number of samples, the mean of $(\rho L)^{2\Delta_j}$ will concentrate around its expectation, which is approximately

$$\sum_{\Delta=0}^\infty p(1-p)^{2\Delta}(\rho L)^{2\Delta} = \frac{p}{1-(1-p)^2(\rho L)^2} \leq \frac{p}{1-\rho L}.$$

Then, the bound in equation 50 becomes

$$\mathbb{E}\left[\exp\left[\nu\hat{d}_1(Z)\right]\right] \lesssim \exp\left[\frac{\nu^2\sigma^2 L^2 p|\mathcal{K}|}{2(1-\rho L)^3}\right], \quad \forall \nu \geq 0.$$ (51)

Applying Chernoff's bound to equation 51, we get

$$\mathbb{P}\left[\hat{d}_1(Z) \leq \epsilon\right] \geq 1 - \exp\left[-\frac{(1-\rho L)^3}{2\sigma^2 L^2 p|\mathcal{K}|}\cdot\epsilon^2\right], \quad \forall \epsilon \geq 0.$$ (52)

**Step 1-2.** Next, we analyze the term $\hat{d}_2(Z)$. Define the set

$$\mathcal{K}' := \{t \mid 1 \leq t \leq T,\ t \in \mathcal{K}^c,\ t-1 \in \mathcal{K}\}.$$

With probability at least $1 - \exp[-\Theta[p(1-p)T]]$, we have

$$|\mathcal{K}'| = \Theta[p(1-p)T].$$

Therefore, $\mathcal{K}'$ is non-empty with high-probability. Since $\|Zf(x_t)\|_2 \geq 0$ for all $t \in \mathcal{K}^c$, we know

$$\hat{d}_2(Z) \geq \sum_{k\in\mathcal{K}'}\|Zf(x_t)\|_2.$$

To establish a high-probability lower bound of $\|Zf(x_t)\|_2$, we prove the following lemma.

**Lemma 3.** *For each $t \in \mathcal{K}'$, it holds that*

$$\mathbb{P}\left[\|Zf(x_t)\|_2 \geq \frac{\lambda}{2}\ \Bigg|\ \mathcal{F}_t\right] \geq \frac{c\lambda^4}{\sigma^4 L^4},$$

*where $c := 1/1058$ is an absolute constant.*

For each $t \in \mathcal{K}'$, let $\mathbf{1}_t$ be the indicator of the event that $\|Zf(x_t)\|_2$ is larger than the $\frac{c\lambda^4}{\sigma^4 L^4}$-quantile conditional on $\mathcal{F}_t$. Then, it holds that

$$\mathbb{P}(\mathbf{1}_t = 1 \mid \mathcal{F}_t) = 1 - \mathbb{P}(\mathbf{1}_t = 0 \mid \mathcal{F}_t) = \frac{c\lambda^4}{\sigma^4 L^4}.$$

Therefore, we know

$$\left\{\mathbf{1}_t - \frac{c\lambda^4}{\sigma^4 L^4},\ t \in \mathcal{K}'\right\}$$

is a martingale with respect to filtration set $\{\mathcal{F}_t, \ t \in \mathcal{K}'\}$. Applying Azuma's inequality, it holds with probability at least $1 - \exp[-\Theta(\frac{\lambda^4 |\mathcal{K}'|}{\sigma^4 L^4})]$ that

$$\sum_{t \in \mathcal{K}'} \mathbf{1}_t \geq \frac{c\lambda^4 |\mathcal{K}'|}{2\sigma^4 L^4},$$

which means that for at least $\frac{c\lambda^4 |\mathcal{K}'|}{2\sigma^4 L^4}$ elements in $\mathcal{K}'$, the event that $\|Zf(x_t)\|_2$ is larger than the $\frac{c\lambda^4}{\sigma^4 L^4}$-quantile conditional on $\mathcal{F}_t$ happens. Using the lower bound on the quantile in Lemma 3, we know

$$\sum_{t \in \mathcal{K}'} \|Zf(x_t)\|_2 \geq \frac{c\lambda^4 |\mathcal{K}'|}{2\sigma^4 L^4} \cdot \frac{\lambda}{2} + \left( |\mathcal{K}'| - \frac{c\lambda^4 |\mathcal{K}'|}{2\sigma^4 L^4} \right) \cdot 0 = \frac{c\lambda^5 |\mathcal{K}'|}{4\sigma^4 L^4} \tag{53}$$

holds with the same probability.

Combining inequalities equation 52 and equation 53, we get

$$\mathbb{P}\left[ f(Z) \leq \epsilon - \frac{c\lambda^5 |\mathcal{K}'|}{4\sigma^4 L^4} \right] \geq 1 - \exp\left[ -\frac{(1-\rho L)^3}{2\sigma^2 L^2 p |\mathcal{K}|} \cdot \epsilon^2 \right] - \exp\left[ -\Theta\left( \frac{\lambda^4 |\mathcal{K}'|}{\sigma^4 L^4} \right) \right],$$

where we define $f(Z) := \hat{d}_1(Z) - \hat{d}_2(Z)$. Choosing

$$\epsilon := \frac{c\lambda^5 |\mathcal{K}'|}{8\sigma^4 L^4},$$

it follows that

$$\mathbb{P}\left[ f(Z) \leq -\frac{c\lambda^5 |\mathcal{K}'|}{8\sigma^4 L^4} \right] \tag{54}$$

$$\geq 1 - \exp\left[ -\Theta\left( \frac{(1-\rho L)^3 \lambda^{10} |\mathcal{K}'|^2}{\sigma^{10} L^{10} p |\mathcal{K}|} \right) \right] - \exp\left[ -\Theta\left( \frac{\lambda^4 |\mathcal{K}'|}{\sigma^4 L^4} \right) \right].$$

By the definition of the probabilistic attack model, it holds with probability at least $1 - \exp[-\Theta[p(1-p)T]]$ that

$$|\mathcal{K}| \leq 2pT, \quad |\mathcal{K}'| \geq \frac{p(1-p)T}{2}. \tag{55}$$

Therefore, the probability bound in equation 54 becomes

$$\mathbb{P}\left[ f(Z) \leq -\frac{c\lambda^5 p(1-p)T}{16\sigma^4 L^4} \right] \geq 1 - \exp\left[ -\Theta\left( \frac{(1-\rho L)^3 \lambda^{10} (1-p)^2 T}{\sigma^{10} L^{10}} \right) \right]$$

$$- \exp\left[ -\Theta\left( \frac{\lambda^4 p(1-p)T}{\sigma^4 L^4} \right) \right] - \exp[-\Theta[p(1-p)T]].$$

Now, if the sample complexity satisfies

$$T \geq \Theta\left[ \max\left\{ \frac{\kappa^{10}}{(1-\rho L)^3 (1-p)^2}, \frac{\kappa^4}{p(1-p)} \right\} \log\left( \frac{1}{\delta} \right) \right], \tag{56}$$

we know

$$\mathbb{P}\left[ f(Z) \leq -\theta \right] \geq 1 - \delta, \tag{57}$$

where we define

$$\kappa := \frac{\sigma L}{\lambda}, \quad \theta := \frac{c\lambda^5 p(1-p)T}{16\sigma^4 L^4}.$$

**Step 2.** In the second step, we apply discretization techniques to prove that condition equation 57 holds for all $Z \in \mathbb{S}_F$. For a sufficiently small constant $\epsilon > 0$, let

$$\{Z^1, \ldots, Z^N\}$$

be an $\epsilon$-cover of the unit ball $\mathbb{S}_F$. Namely, for all $Z \in \mathbb{S}_F$, we can find $r \in \{1, 2, \ldots, N\}$ such that $\|Z - Z^r\|_F \leq \epsilon$. It is proved in Wainwright (2019) that the number of points $N$ can be bounded by

$$\log(N) \leq mn \log\left(1 + \frac{2}{\epsilon}\right).$$

Now, we estimate the Lipschitz constant of $f(Z)$ and construct a high-probability upper bound for the Lipschitz constant. For all $Z, Z' \in \mathbb{R}^{n \times m}$, we can calculate that

$$
\begin{aligned}
f(Z) - f(Z') &= \sum_{t \in \mathcal{K}} \left\langle (Z - Z')^\top, f(x_t)\hat{d}_t^\top \right\rangle - \sum_{t \in \mathcal{K}^c} \left( \|Zf(x_t)\|_2 - \|Z'f(x_t)\|_2 \right) \\
&\leq \|Z - Z'\|_F \sum_{t \in \mathcal{K}} \left\| f(x_t)\hat{d}_t^\top \right\|_F + \|Z - Z'\|_2 \sum_{t \in \mathcal{K}^c} \|f(x_t)\|_2 \\
&\leq \|Z - Z'\|_F \sum_{t=0}^{T-1} \|f(x_t)\|_2 .
\end{aligned}
\tag{58}
$$

Using the decomposition in **Step 1-1**, we have

$$f(x_t) = \sum_{\ell=1}^{j} g_t^{k_\ell},$$

where $k_j$ is the maximal element in $\mathcal{K}$ such that $k_j < t$. Therefore, we can calculate that

$$\sum_{t=0}^{T-1} \|f(x_t)\|_2 \leq \sum_{j=1}^{|\mathcal{K}|} \sum_{t=k_j+1}^{T-1} \left\| g_t^{k_j} \right\|_2 .
\tag{59}$$

For each $j \in \{1, \ldots, |\mathcal{K}|\}$, we can prove in the same way as equation 47 that

$$\left\| g_t^{k_j} \right\|_2 \leq L(\rho L)^{k_j - t - 1} \|\bar{d}_{k_j}\|_2, \quad \forall t > k_j.$$

Substituting into inequality equation 59, it follows that

$$\sum_{t=0}^{T-1} \|f(x_t)\|_2 \leq \sum_{j=1}^{|\mathcal{K}|} \sum_{t=k_j+1}^{T-1} L(\rho L)^{k_j - t - 1} \|\bar{d}_{k_j}\|_2 \leq \frac{L}{1 - \rho L} \sum_{j=1}^{|\mathcal{K}|} \|\bar{d}_{k_j}\|_2 .$$

Using Assumption 6 and the same technique as in equation 50, we know

$$\mathbb{P}\left( \sum_{j=1}^{|\mathcal{K}|} \|\bar{d}_{k_j}\|_2 \leq \eta \right) \geq 1 - 2\exp\left( -\frac{\eta^2}{2\sigma^2|\mathcal{K}|} \right) \geq 1 - 2\exp\left( -\frac{\eta^2}{4\sigma^2 pT} \right),$$

where the second inequality is from the high probability bound in equation 55. Hence, it holds that

$$\mathbb{P}\left( \sum_{t=0}^{T-1} \|f(x_t)\|_2 \leq \eta \right) \geq 1 - 2\exp\left( -\frac{\eta^2(1 - \rho L)^2}{4\sigma^2 L^2 pT} \right),
\tag{60}$$

Choosing

$$\eta := \frac{\theta}{2\epsilon},$$

the bound in equation 60 becomes

$$
\begin{aligned}
\mathbb{P}\left( \sum_{t=0}^{T-1} \|f(x_t)\|_2 \leq \frac{\theta}{2\epsilon} \right) &\geq 1 - 2\exp\left( -\frac{(1 - \rho L)^2}{4\sigma^2 L^2 pT\epsilon^2} \cdot \theta^2 \right) \\
&= 1 - 2\exp\left[ -\Theta\left[ \frac{(1 - \rho L)^2}{4\sigma^2 L^2 pT\epsilon^2} \cdot \left( \frac{\lambda^5 p(1-p)T}{\sigma^4 L^4} \right)^2 \right] \right] \\
&= 1 - 2\exp\left[ -\Theta\left[ \frac{(1 - \rho L)^2 \kappa^{10} p(1-p)^2 T}{\epsilon^2} \right] \right].
\end{aligned}
\tag{61}
$$

We set

$$\epsilon := \Theta \left[ \sqrt{(1 - \rho L)^2 \kappa^{10} p (1 - p)^2} \right].$$

Then, it follows that

$$\exp \left[ -\Theta \left[ \frac{(1 - \rho L)^2 \kappa^{10} p (1 - p)^2 T}{\epsilon^2} \right] \right] = \exp \left[ -\Theta(T) \right] \leq \frac{\delta}{4},$$

where the last inequality is from the choice of $T$ in equation 56. Substituting back into equation 61, we get

$$\mathbb{P} \left( \sum_{t=0}^{T-1} \|f(x_t)\|_2 \leq \frac{\theta}{2\epsilon} \right) \geq 1 - \frac{\delta}{2}. \tag{62}$$

Under the event in equation 62, for all $Z \in \mathbb{S}_F$, there exists an element $Z^r$ in the $\epsilon$-net such that

$$f(Z) \leq f(Z^r) + \epsilon \cdot \sum_{t=0}^{T-1} \|f(x_t)\|_2 \leq f(Z^r) + \frac{\theta}{2}.$$

If we replace $\delta$ with $\delta/(2N)$ in equation 57 and choose $Z = Z^r$ for all $r \in \{1, \ldots, N\}$, the union bound implies that

$$\mathbb{P} \left[ f(Z^r) \leq -\theta, \ r = 1, \ldots, N \right] \geq 1 - \frac{\delta}{2}. \tag{63}$$

Under the above condition, we have

$$f(Z) \leq f(Z^r) + \frac{\theta}{2} \leq -\frac{\theta}{2} < 0.$$

To satisfy condition equation 63, the sample complexity bound equation 56 becomes

$$T \geq \Theta \left[ \max \left\{ \frac{\kappa^{10}}{(1 - \rho L)^3 (1 - p)^2}, \frac{\kappa^4}{p(1 - p)} \right\} \log \left( \frac{2N}{\delta} \right) \right]$$

$$= \Theta \left[ \max \left\{ \frac{\kappa^{10}}{(1 - \rho L)^3 (1 - p)^2}, \frac{\kappa^4}{p(1 - p)} \right\} \right.$$

$$\left. \times \left[ mn \log \left( \frac{1}{(1 - \rho L) \kappa p (1 - p)} \right) + \log \left( \frac{1}{\delta} \right) \right] \right],$$

which is the desired sample complexity bound in the theorem.

**Lower bound of $\kappa$.** Before we close the proof, we provide a lower bound of $\kappa = \sigma L / \lambda$. Equivalently, we provide an upper bound on $\lambda^2$, which is at most the minimal eigenvalue of

$$\mathbb{E} \left[ f(x + \bar{d}_t) f(x + \bar{d}_t)^\top \mid \mathcal{F}_t, \bar{d}_t \neq 0_n \right].$$

Let $\nu \in \mathbb{R}^m$ be a vector satisfying

$$\|\nu\|_2 = 1, \quad \nu^\top f(x) = 0.$$

Then, we know

$$\nu^\top f(x + \bar{d}_t) f(x + \bar{d}_t)^\top \nu = \nu^\top \left[ f(x + \bar{d}_t) - f(x) \right] \left[ f(x + \bar{d}_t) - f(x) \right]^\top \nu \tag{64}$$

$$= \left[ \left[ f(x + \bar{d}_t) - f(x) \right]^\top \nu \right]^2 \leq \left\| f(x + \bar{d}_t) - f(x) \right\|_2^2$$

$$\leq L^2 \|\bar{d}_t\|_2^2,$$

where the last inequality is from the Lipschitz continuity of $f$. Using the sub-Gaussian assumption, it follows that

$$\mathbb{E} \left[ \|\bar{d}_t\|_2^2 \mid \mathcal{F}_t, \ \bar{d}_t \neq 0_n \right] \leq \sigma^2, \tag{65}$$

where we utilize the fact that the standard deviation of sub-Gaussian random variables with parameter $\sigma$ is at most $\sigma$. Combining inequalities equation 64 and equation 65, it follows that

$$\nu^\top \mathbb{E}\left[f(x + \bar{d}_t)f(x + \bar{d}_t)^\top \mid \mathcal{F}_t, \bar{d}_t \neq 0_n\right]\nu \leq \sigma^2 L^2.$$

Therefore, it holds that

$$\lambda^2 \leq \lambda_{min}\left[\mathbb{E}\left[f(x + \bar{d}_t)f(x + \bar{d}_t)^\top \mid \mathcal{F}_t, \bar{d}_t \neq 0_n\right]\right] \leq \sigma^2 L^2, \quad \forall x \in \mathbb{R}^n,$$

which further leads to

$$\kappa = \frac{\sigma L}{\lambda} \geq 1.$$

This completes the proof. $\qquad\square$

### D.8 PROOF OF LEMMA 3

*Proof of Lemma 3.* Let

$$\delta := \frac{c\lambda^4}{\sigma^4 L^4}, \quad \theta_t := \left\|Z^\top f\left[\bar{A}f(x_{t-1})\right]\right\|_2.$$

We finish the proof by discussing two cases.

**Case 1.** We first consider the case when

$$\theta_t \geq \frac{\lambda}{2} + \sqrt{2\sigma^2 L^2 \log\left(\frac{2}{1-\delta}\right)}.$$

Using the Lipschitz continuity of $f$, we have

$$\begin{aligned}
\|Zf(x_t)\|_2 &= \left\|\left[Zf(x_t) - Z^\top f\left[\bar{A}f(x_{t-1})\right]\right] + Zf\left[\bar{A}f(x_{t-1})\right]\right\|_2 \qquad (66)\\
&\geq \left\|Zf\left[\bar{A}f(x_{t-1})\right]\right\|_2 - \left\|Zf(x_t) - Zf\left[\bar{A}f(x_{t-1})\right]\right\|_2 \\
&\geq \theta_t - \|Z\|_2\left\|f(x_t) - f\left[\bar{A}f(x_{t-1})\right]\right\|_2 \\
&\geq \theta_t - \|Z\|_F \cdot L\left\|\bar{d}_t\right\|_2 \geq \theta_t - L\left\|\bar{d}_t\right\|_2.
\end{aligned}$$

By Assumption 6, we know $\left\|\bar{d}_t\right\|_2 = |\ell_t|$ and it follows that

$$\mathbb{P}\left(\left\|\bar{d}_t\right\|_2 \geq \epsilon \mid \mathcal{F}_t\right) \leq 2\exp\left(-\frac{\epsilon^2}{2\sigma^2}\right), \quad \forall \epsilon \geq 0.$$

Therefore, we get the estimation

$$\begin{aligned}
\mathbb{P}\left(\|Zf(x_t)\|_2 \leq \frac{\lambda}{2} \,\Big|\, \mathcal{F}_t\right) &\leq \mathbb{P}\left(\theta_t - L\left\|\bar{d}_t\right\|_2 \leq \frac{\lambda}{2} \,\Big|\, \mathcal{F}_t\right) \\
&= \mathbb{P}\left(\left\|\bar{d}_t\right\|_2 \geq \frac{\theta_t - \lambda/2}{L} \,\Big|\, \mathcal{F}_t\right) \\
&\leq \mathbb{P}\left(\left\|\bar{d}_t\right\|_2 \geq \sqrt{2\sigma^2 \log\left(\frac{2}{1-\delta}\right)} \,\Big|\, \mathcal{F}_t\right) \leq 1 - \delta.
\end{aligned}$$

Therefore, we have proved that

$$\mathbb{P}\left(\|Zf(x_t)\|_2 \geq \frac{\lambda}{2} \,\Big|\, \mathcal{F}_t\right) \geq \delta.$$

**Case 2.** Then, we focus on the case when

$$\theta_t \leq \frac{\lambda}{2} + \sqrt{2\sigma^2 L^2 \log\left(\frac{2}{1-\delta}\right)}. \qquad (67)$$

Assume conversely that

$$\mathbb{P}\left(\|Zf(x_t)\|_2 \geq \frac{\lambda}{2} \,\middle|\, \mathcal{F}_t\right) < \delta. \tag{68}$$

Similar to inequality equation 66, the Lipschitz continuity of $f$ implies

$$\|Zf(x_t)\|_2 \leq \theta_t + L \left\|\bar{d}_t\right\|_2.$$

Therefore, by applying Assumption 6, we get the tail bound

$$\mathbb{P}\left(\|Zf(x_t)\|_2 \geq \theta \mid \mathcal{F}_t\right) \leq \mathbb{P}\left(\theta_t + L \left\|\bar{d}_t\right\|_2 \geq \theta \mid \mathcal{F}_t\right)$$

$$=\mathbb{P}\left(\left\|\bar{d}_t\right\|_2 \geq \frac{\theta - \theta_t}{L} \,\middle|\, \mathcal{F}_t\right) \leq 2\exp\left[-\frac{(\theta - \theta_t)^2}{2\sigma^2 L^2}\right], \quad \forall \theta \geq \theta_t.$$

Define $(x)_+ := \max\{x, 0\}$. The above bound leads to

$$\mathbb{P}\left(\|Zf(x_t)\|_2 \geq \theta \mid \mathcal{F}_t\right) \leq 2\exp\left[-\frac{(\theta - \theta_t)_+^2}{2\sigma^2 L^2}\right], \quad \forall \theta \in \mathbb{R}. \tag{69}$$

Using the definition of expectation, we can calculate that

$$\mathbb{E}\left[\|Zf(x_t)\|_2^2 \mid \mathcal{F}_t\right] = \int_0^\infty 2\theta \cdot \mathbb{P}\left[\|Zf(x_t)\|_2 \geq \theta \mid \mathcal{F}_t\right] \, d\theta$$

$$\leq \frac{\lambda^2}{4} + \int_{\lambda/2}^\infty 2\theta \cdot \mathbb{P}\left[\|Zf(x_t)\|_2 \geq \theta \mid \mathcal{F}_t\right] \, d\theta.$$

By condition equation 68, we get

$$\mathbb{P}\left[\|Zf(x_t)\|_2 \geq \theta \mid \mathcal{F}_t\right] \leq \mathbb{P}\left[\|Zf(x_t)\|_2 \geq \frac{\lambda}{2} \,\middle|\, \mathcal{F}_t\right] \leq \delta, \quad \forall \theta \geq \frac{\lambda}{2}.$$

Combining with inequality equation 69, it follows that

$$\mathbb{E}\left[\|Zf(x_t)\|_2^2 \mid \mathcal{F}_t\right] \leq \frac{\lambda^2}{4} + \int_{\lambda/2}^\infty 2\theta \cdot \min\left\{\delta, 2\exp\left[-\frac{(\theta - \theta_t)_+^2}{2\sigma^2 L^2}\right]\right\} \, d\theta \tag{70}$$

$$= \frac{\lambda^2}{4} + \delta\left(\theta_1^2 - \frac{\lambda^2}{4}\right) + \int_{\theta_1}^\infty 4\theta \exp\left[-\frac{(\theta - \theta_t)^2}{2\sigma^2 L^2}\right] \, d\theta,$$

where we define

$$\theta_1 := \max\left\{\frac{\lambda}{2}, \theta_t + \sqrt{2\sigma^2 L^2 \log\left(\frac{2}{\delta}\right)}\right\} \geq \theta_t.$$

Using condition equation 67, we know

$$\theta_1^2 \leq \left(\frac{\lambda}{2} + \sqrt{2\sigma^2 L^2 \log\left(\frac{2}{1-\delta}\right)} + \sqrt{2\sigma^2 L^2 \log\left(\frac{2}{\delta}\right)}\right)^2 \tag{71}$$

$$\leq \left(\frac{\lambda}{2} + 2\sqrt{2\sigma^2 L^2 \log\left(\frac{2}{\delta}\right)}\right)^2 \leq \frac{\lambda^2}{2} + 16\sigma^2 L^2 \log\left(\frac{2}{\delta}\right),$$

where the last inequality is from Cauchy's inequality. Moreover, we can estimate that

$$\int_{\theta_1}^\infty 4\theta \exp\left[-\frac{(\theta - \theta_t)^2}{2\sigma^2 L^2}\right] \, d\theta \leq \int_{\theta_2}^\infty 4\theta \exp\left[-\frac{(\theta - \theta_t)^2}{2\sigma^2 L^2}\right] \, d\theta \tag{72}$$

$$= \int_{\theta_2}^\infty 4\theta_t \exp\left[-\frac{(\theta - \theta_t)^2}{2\sigma^2 L^2}\right] \, d\theta + \int_{\theta_2}^\infty 4(\theta - \theta_t) \exp\left[-\frac{(\theta - \theta_t)^2}{2\sigma^2 L^2}\right] \, d\theta$$

$$= \int_{\theta_2}^\infty 4\theta_t \exp\left[-\frac{(\theta - \theta_t)^2}{2\sigma^2 L^2}\right] \, d\theta + 2\delta\sigma^2 L^2,$$

where we denote $\theta_2 := \theta_t + \sqrt{2\sigma^2 L^2 \log\left(\frac{2}{\delta}\right)} \leq \theta_1$. Utilizing the following bound on the cumulative density function of the standard Gaussian distribution:

$$\int_\eta^\infty e^{-\frac{x^2}{2}} \, dx \leq \eta^{-1} e^{-\frac{\eta^2}{2}}, \quad \forall \eta > 0,$$

we have

$$\int_{\theta_2}^\infty 4\theta_t \exp\left[-\frac{(\theta - \theta_t)^2}{2\sigma^2 L^2}\right] \, d\theta \leq 4\theta_t \sigma L \cdot \frac{1}{\sqrt{2\log\left(\frac{2}{\delta}\right)}} \cdot \frac{\delta}{2} \leq \sqrt{2}\theta_t \cdot \delta\sigma L.$$

Combining with equation 72, it follows that

$$\int_{\theta_1}^\infty 4\theta \exp\left[-\frac{(\theta - \theta_t)^2}{2\sigma^2 L^2}\right] \, d\theta \leq \sqrt{2}\theta_t \cdot \delta\sigma L + 2\delta\sigma^2 L^2 \leq 4\delta\theta_t^2 + 4\delta\sigma^2 L^2, \tag{73}$$

where the last inequality is from Cauchy's inequality. Substituting inequalities equation 71 and equation 73 back into equation 70, we get

$$\mathbb{E}\left[\|Zf(x_t)\|_2^2 \mid \mathcal{F}_t\right] \leq \frac{\lambda^2}{4} + \delta\left[\frac{\lambda^2}{4} + 16\sigma^2 L^2 \log\left(\frac{2}{\delta}\right)\right] + 4\delta\theta_t^2 + 4\delta\sigma^2 L^2$$

$$\leq \frac{(1+\delta)\lambda^2}{4} + 16\sigma^2 L^2 \cdot \delta \log\left(\frac{2}{\delta}\right) + \delta\left[\frac{\lambda}{2} + \sqrt{2\sigma^2 L^2 \log\left(\frac{2}{1-\delta}\right)}\right]^2 + 4\delta\sigma^2 L^2$$

$$\leq \frac{(1+\delta)\lambda^2}{4} + 16\sigma^2 L^2 \cdot \delta \log\left(\frac{2}{\delta}\right) + \frac{\delta\lambda^2}{2} + 4\sigma^2 L^2 \cdot \delta \log\left(\frac{2}{\delta}\right) + 4\delta\sigma^2 L^2$$

$$\leq \frac{(1+3\delta)\lambda^2}{4} + 24\sigma^2 L^2 \cdot \delta \log\left(\frac{2}{\delta}\right).$$

where the second inequality is from equation 67 and the last inequality is from Cauchy's inequality and $\delta < 1/2$. On the other hand, Assumption 3 implies that

$$\mathbb{E}\left(\|Zf(x_t)\|_2^2 \mid \mathcal{F}_t\right) = \left\langle ZZ^\top, \mathbb{E}\left[f(x_t)f(x_t)^\top \mid \mathcal{F}_t\right]\right\rangle \geq \lambda^2 \|Z\|_F^2 = \lambda^2.$$

Combining the last two inequalities, we get

$$\lambda^2 \leq \frac{(1+3\delta)\lambda^2}{4} + 24\sigma^2 L^2 \cdot \delta \log\left(\frac{2}{\delta}\right),$$

which is equivalent to

$$\delta \log\left(\frac{2}{\delta}\right) \geq \frac{(3-3\delta)\lambda^2}{96\sigma^2 L^2} \geq \frac{\lambda^2}{23\sigma^2 L^2}.$$

For all $x \in (0,1)$, it holds that $x\log(2/x) < \sqrt{2x}$. Hence, we have

$$\sqrt{2\delta} > \frac{\lambda^2}{23\sigma^2 L^2},$$

which contradicts with our assumption equation 68.

$\square$

### D.9 PROOF OF THEOREM 7

*Proof of Theorem 7.* In this proof, we focus on the case when $m = n$ and the counterexample can be easily extended into more general cases. We construct the following system dynamics:

$$\bar{A} := \rho I_n, \quad f(x) := x, \quad \forall x \in \mathbb{R}^n,$$

where $\rho \geq 2 + \sqrt{6}$ is a constant. One can verify Assumption 4 holds with Lipschitz constant $L = 1$. Therefore, the stability condition (Assumption 5) is violated since $\rho > 1/L$. The system dynamics can be written as

$$x_t = \sum_{k \in \mathcal{K}, k < t} \rho^{t-k-1} d_k, \quad \forall t \in \{0, \dots, T\}. \tag{74}$$

Conditional on $\mathcal{F}_t$ and $t \in \mathcal{K}$, the attack vector is generated as

$$d_t \sim \text{Uniform}(\mathbb{S}^{n-1}),$$

where $\mathbb{S}^{n-1}$ is the unit ball $\{d \in \mathbb{R}^n \mid \|d\|_2 = 1\}$. The attack model satisfies Assumption 3 with $\lambda = 1/\sqrt{n}$ and Assumption 6 with $\sigma = 1/\sqrt{n}$. Define the event

$$\mathcal{E} := \{T - 1 \in \mathcal{K}, |\mathcal{K}| > 1\}.$$

By the definition of the probabilistic attack model, we can calculate that

$$\mathbb{P}(\mathcal{E}) = p\left[1 - (1-p)^{T-1}\right].$$

Our goal is to prove that

$$\mathbb{P}\left[\hat{d}_1(Z) - \hat{d}_2(Z) > 0 \mid \mathcal{E}\right] = 1,$$

where we define

$$\hat{d}_1(Z) := \sum_{t \in \mathcal{K}} \left\langle Z^\top, f(x_t)\hat{d}_t^\top \right\rangle, \quad \hat{d}_2(Z) := \sum_{t \in \mathcal{K}^c} \|Zf(x_t)\|_2.$$

Then, by Theorem 1, we know that $\bar{A}$ is not a global solution to problem equation 3 with probability at least

$$p\left[1 - (1-p)^{T-1}\right].$$

Let $t_1$ be the smallest element in $\mathcal{K}$, namely, the first time instance when there is an attack. Under event $\mathcal{E}$, it holds that $t_1 < T - 1$. We first prove that

$$x_t \neq 0_n, \quad \forall t \in \{t_1 + 1, \dots, T - 1\}.$$

By the system dynamics equation 74 and the triangle inequality, we have

$$\|x_t\|_2 \geq \rho^{t-t_1-1}\|d_{t_1}\|_2 - \sum_{k \in \mathcal{K}, t_1 < k < t} \rho^{t-k-1}\|d_k\|_2 = \rho^{t-t_1-1} - \sum_{k \in \mathcal{K}, t_1 < k < t} \rho^{t-k-1}$$

$$\geq \rho^{t-t_1-1} - \sum_{i=0}^{t-t_1-2} \rho^i = \frac{\rho^{t-t_1} - 2\rho^{t-t_1-1} + 1}{\rho - 1} > 0,$$

where the last inequality holds because $\rho \geq 2$. Then, we choose

$$Z := x_{T-1}\hat{d}_{T-1}^\top \neq 0.$$

It follows that

$$\hat{d}_1(Z) = \sum_{t \in \mathcal{K}} \left\langle Z^\top, f(x_t)\hat{d}_t^\top \right\rangle = \left\|x_{T-1}\hat{d}_{T-1}^\top\right\|_F^2 + \sum_{t \in \mathcal{K}, t < T-1} \left\langle x_{T-1}\hat{d}_{T-1}^\top, f(x_t)\hat{d}_t^\top \right\rangle$$

$$\geq \|x_{T-1}\|_2^2 - \sum_{t \in \mathcal{K}, t < T-1} \|x_{T-1}\|_2 \|x_t\|_2,$$

$$\hat{d}_2(Z) = \sum_{t \in \mathcal{K}^c} \|Zf(x_t)\|_2 = \sum_{t \in \mathcal{K}^c} \left\|x_{T-1}\hat{d}_{T-1}^\top x_t\right\|_2 \leq \sum_{t \in \mathcal{K}^c} \|x_{T-1}\|_2 \|x_t\|_2.$$

Combining the above two inequalities, we get

$$\hat{d}_1(Z) - \hat{d}_2(Z) \leq \|x_{T-1}\|_2 \left(\|x_{T-1}\|_2 - \sum_{t=0}^{T-2} \|x_t\|_2\right) = \|x_{T-1}\|_2 \left(\|x_{T-1}\|_2 - \sum_{t=t_1+1}^{T-2} \|x_t\|_2\right),$$

where the last equality holds because $x_t = 0_n$ for all $t \leq t_1$. Since $\|x_{T-1}\|_2 > 0$, it is sufficient to prove that

$$\|x_{T-1}\|_2 > \sum_{t=t_1+1}^{T-2} \|x_t\|_2. \tag{75}$$

Considering the system dynamics equation 74 and the fact that $\|d_k\|_2 = 1$ for all $k \in \mathcal{K}$ , we have the estimation

$$\rho^{t-t_1-1} - \sum_{k \in \mathcal{K}, t_1 < k < t} \rho^{t-k-1} \le \|x_t\|_2 \le \sum_{k \in \mathcal{K}, k < t} \rho^{t-k-1}.$$

The desired inequality equation 75 holds if we can show

$$\rho^{T-1-t_1-1} - \sum_{k \in \mathcal{K}, t_1 < k < T-1} \rho^{T-1-k-1} > \sum_{t=t_1+1}^{T-2} \sum_{k \in \mathcal{K}, k < t} \rho^{t-k-1},$$

which is further equivalent to

$$2\rho^{T-t_1-2} > \sum_{t=t_1+1}^{T-1} \sum_{k \in \mathcal{K}, k < t} \rho^{t-k-1}$$

$$\Longleftarrow 2\rho^{T-t_1-2} > \sum_{t=t_1+1}^{T-1} \sum_{k=t_1}^{t-1} \rho^{t-k-1} = \sum_{t=t_1+1}^{T-1} \frac{\rho^{t-t_1} - 1}{\rho - 1} = \frac{\rho^{T-t_1} - \rho - (T - t_1 - 1)(\rho - 1)}{(\rho - 1)^2}$$

$$\Longleftarrow 2\rho^{T-t_1-2} \ge \frac{\rho^{T-t_1}}{(\rho - 1)^2} \iff \rho^2 - 4\rho - 2 \ge 0 \Longleftarrow \rho \ge 2 + \sqrt{6}.$$

By our choice of $\rho$, we know condition equation 75 holds and this completes our proof.

$\square$

# E  NUMERICAL EXPERIMENTS FOR BOUNDED BASIS FUNCTION

In this section, we provide the descriptions of basis functions and analyze the performance of estimator equation 2 in the case of bounded basis function. We show that the estimator equation 2 is able to exactly recover the ground truth $\bar{A}$ with different attack probability $p$ and problem dimension $(n, m)$. We utilize the same evaluation metrics as in Section 7 and define the system dynamics as follows.

**Lipschitz basis function.**   Given the state space dimension $n$, we choose $m = n$ and define the basis function as

$$f(x) := \frac{1}{\sqrt{n}} \begin{bmatrix} \sqrt{\|x - x_1\|_2^2 + 1} - \sqrt{\|x_1\|_2^2 + 1} \\ \vdots \\ \sqrt{\|x - x_n\|_2^2 + 1} - \sqrt{\|x_n\|_2^2 + 1} \end{bmatrix}, \quad \forall x \in \mathbb{R}^n,$$

where $x_1, \ldots, x_n \in \mathbb{R}^n$ are instances of i.i.d. standard Gaussian random vectors. We can verify that the basis function is Lipschitz continuous with Lipschitz constant $L = 1$ and thus, it satisfies Assumption 4. For each time instance $t \in \mathcal{K}$, the noise $\bar{d}_t$ is generated by

$$\bar{d}_t := \ell_t \hat{d}_t, \quad \text{where } \ell_t \sim \mathcal{N}(0, \sigma_t^2), \ \hat{d}_t \sim \text{uniform}(\mathbb{S}^{n-1}), \ \ell_t \text{ and } \hat{d}_t \text{ are independent.}$$

Here, we define $\sigma_t^2 := \min\{\|x_t\|_2^2, 1/n\}$. We can verify that the random variable $\ell_t$ is zero-mean and sub-Gaussian with parameter $\sigma = 1$. In addition, the random vector $\hat{d}_t$ follows the uniform distribution and therefore, Assumption 6 is satisfied. Note that $\bar{d}_0, \ldots, \bar{d}_{T-1}$ are correlated and they violate the i.i.d. assumption in the literature. Our attack model implies that the intensity of an attack (namely, $\ell_t$) depends on the current state, which is a function of previous attacks. Since the points $x_1, \ldots, x_n$ are randomly generated, the multiquadric radial basis functions are linearly independent[1] with probability 1 and therefore, the non-degenerate assumption (Assumption 3) is satisfied. Finally, the ground truth matrix $\bar{A}$ is constructed as $U\Sigma V^\top$, where $U, V \in \mathbb{R}^{n \times n}$ are random orthogonal matrices and $\Sigma = \text{diag}(\sigma_1, \ldots, \sigma_n)$ is a diagonal matrix. The singular values are generated as follows:

$$\sigma_i \overset{\text{i.i.d.}}{\sim} \text{uniform}(0, \rho), \quad \forall i \in \{1, \ldots, n\},$$

where $\rho > 0$ is the upper bound on the spectral norm of $\bar{A}$.

---

[1]Functions $g_1(y), \ldots, g_k(y)$ are said to be linearly independent if there do not exist constants $c_1, \ldots, c_k$ such that $\sum_{i=1}^k c_i g_i(y) = 0$ for all $y$.

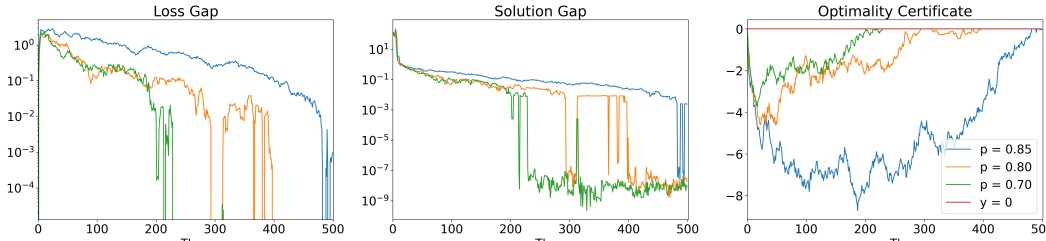

Figure 4: Loss gap, solution gap and optimality certificate of the bounded basis function case with attack probability $p = 0.7, 0.8$ and $0.85$.

**Bounded basis function.** Given the state space dimension $n$, we choose $m = 5n$ and define the basis function as

$$f(x) := \begin{bmatrix} \tilde{f}(x_1) \\ \vdots \\ \tilde{f}(x_n) \end{bmatrix}, \quad \text{where } \tilde{f}(y) := \begin{bmatrix} \sin(y) \\ \vdots \\ \sin(5y) \end{bmatrix}, \quad \forall x \in \mathbb{R}^n, \ y \in \mathbb{R}.$$

The basis function satisfies Assumption 2 with $B = 1$. For each time instance $t \in \mathcal{K}$ and for each $i \in \{1, \ldots, n\}$, the noise $\bar{d}_{t,i}$ is independently generated by

$$\bar{d}_{t,i} \sim \text{Uniform}\left(-c_{t,i}\pi, c_{t,i}\pi\right), \quad \text{where } c_{i,t} := \min\{\max\{|x_{t,i}|, 0.1\}, 0.5\}.$$

Note that $\bar{d}_{t,i}$ and $x_{t,i}$ is the $i$-th component of $\bar{d}_t$ and $x_t$, respectively. Since the attack is symmetric with respect to the origin, it satisfies Assumption 1. Since the sine functions $\sin(y), \ldots, \sin(5y)$ are linearly independent, the non-degenerate assumption (Assumption 3) is satisfied. Finally, the ground truth matrix $\bar{A}$ is constructed such that

$$\bar{A}f(x) = \begin{bmatrix} \sum_{k=1}^{5} \bar{a}_{1,k} \sin(kx_1) \\ \vdots \\ \sum_{k=1}^{5} \bar{a}_{n,k} \sin(kx_n) \end{bmatrix},$$

where

$$\bar{a}_{i,k} \overset{\text{i.i.d.}}{\sim} \text{Uniform}(-100, 100), \quad \forall i \in \{1, \ldots, n\}, \ k \in \{1, \ldots, 5\}.$$

We note that we choose the upper bound of coefficients $\bar{a}_{i,k}$ to be larger than 1 to show that the stability condition (Assumption 5) is not required in the bounded basis function case.

**Results.** We first compare the performance of estimator equation 2 with different attack probability $p$. We choose $T = 900$, $n = 1$ and $p \in \{0.7, 0.8, 0.85\}$. The results are plotted in Figure 4. We can observe behaviors similar to the Lipschitz basis function case. More specifically, the optimality certificate accurately measures the exact recovery of the estimator equation 2, and the required sample complexity increases with the probability of attack $p$.

Next, we show the performance of the estimator equation 2 with different dimensions $(n, m)$. We choose $T = 500$, $p = 0.7$ and $n \in \{1, 2, 4\}$. The results are plotted in Figure 5. We can see that the exact recovery occurs with more samples when $(n, m)$ is larger, which still verifies the results in Theorem 4.

## F    NUMERICAL EXPERIMENTS WITH LOW ATTACK FREQUENCY

In this section, we repeat the experiments in Figure 1 with $p \in \{0.001, 0.1, 0.3\}$ and $n = 5$. The results are plotted in Figure 6. We can see that the predictor fails to find the ground truth within 500 steps when $p = 0.01$, while it converges when $p = 0.1$ and $0.3$. Note that the loss gap and optimality certificate are both equal to 0 in the case when $p = 0.001$. This is because there exist multiple global solutions and the estimator fails to recover the ground truth solution within 500 iterations. Note that the algorithm will eventually converge to the ground truth solution when more samples are available.

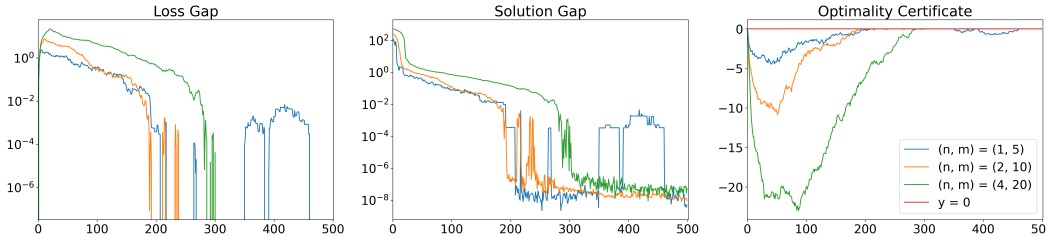

Figure 5: Loss gap, solution gap and optimality certificate of the bounded basis function case with dimension $(n, m) = (1, 5), (2, 10)$ and $(4, 20)$.

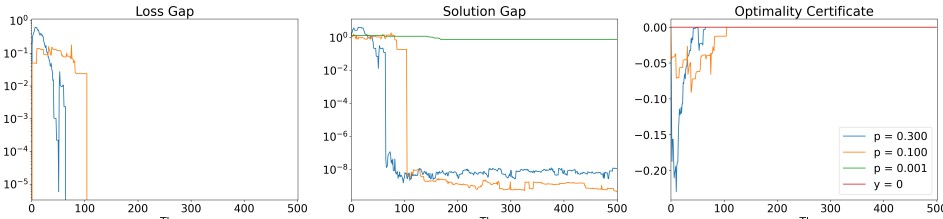

Figure 6: Loss gap, solution gap and optimality certificate of the Lipschitz basis function case with attack probability $p = 0.001, 0.1$ and $0.3$. Note that the loss gap and the optimality certificate for the case when $p = 0.001$ is always equal to 0.

With that said, the main focus of this paper is the regime when $p$ is larger than $0.5$. Note that when $p$ is very small or even zero, learning the system is a classic problem in control theory, where it is known that one should add an artificial noise to the system (named excitation signal) to be able to learn the system. There is a rich literature on why an excitation signal is necessary when the system is (almost) deterministic. As an example, assume that we have the system $x_{t+1} = Ax_t$, where our aim is to learn A from measuring $x_t$. If $x_0$ is zero, $x_t$ always remains zero and we cannot find $A$. To avoid this, we should excite the system as $x_{t+1} = Ax_t + w_t$ where $w_t$ is, for example, Gaussian noise. When $p$ is away from zero, the adversarial attack does us a favor and acts as an excitation signal.

# G NUMERICAL EXPERIMENTS WITH SPARSE $\bar{A}$

In this section, we repeat the experiments shown in Figure 1 using the sparse ground truth matrix $\bar{A}$. Specifically, we generate a sparse matrix $\bar{A}$ where $\bar{A}_{i,j}$ is set to 0 whenever $|i - j| > 1$. In other words, $\bar{A}$ is a tridiagonal matrix. We repeat the experiments for Lipschitz basis functions with $p \in 0.7, 0.8, 0.85$ and $n = 10$. Additionally, we extend the simulation period to $T = 1000$, compared to $T = 500$ in the previous experiments. To save computational time, we solve the problem in equation 2 every 10 time periods. Consequently, the plots exhibit discrete jumps corresponding to time periods that are multiples of ten. We excluded the loss gap from the figures because the estimator is computed only for a subset of the time periods. Figure 7 suggests that we achieve exact recovery despite the sparse structure of the ground truth matrix $\bar{A}$. This result is not surprising, as the theoretical results do not depend on the sparsity structure of $\bar{A}$. In addition to demonstrating robustness, the non-smooth objective function in equation 2 serves as a regularization term for the specific matrix structure.

# H NUMERICAL EXPERIMENTS WITH LARGER ORDER SYSTEMS

In this section, we repeat the experiments shown in Figure 2 with significantly higher-order dynamical systems and a larger number of basis functions, specifically $(n, m) \in (10, 20), (25, 50), (50, 100)$. We set the probability of an attack occurring to $p = 0.6$. Additionally, we extend the simulation period to $T = 1100$, compared to $T = 500$ in the previous experiments. To save computational time, we solve the problem in equation 2 every 100 time periods. Consequently, the plots exhibit discrete

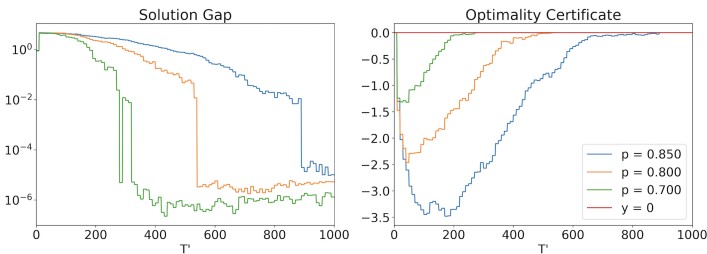

Figure 7: Solution gap and optimality certificate of the Lipschitz basis function case with $p \in \{0.7, 0.8, 0.85\}$ and $n = m = 10$.

jumps corresponding to time periods that are multiples of 100. We excluded the loss gap from the figures because the estimator is computed only for a subset of the time periods.

In Figure 8, we observe that exact recovery is achieved even when the system's dimension and the number of basis functions are significantly large within the context of system identification problems. Achieving exact recovery requires the system to run for a sufficiently long time, with the required time horizon specified in our theoretical results.

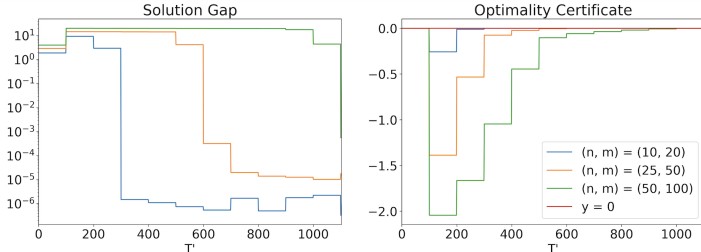

Figure 8: Loss gap, solution gap and optimality certificate of the Lipschitz basis function case with dimension $(n, m) = (10, 20), (25, 50)$ and $(50, 100)$.