# OpenReview forum: "Exact Recovery Guarantees for Parameterized Nonlinear System Identification Problem under Adversarial Attacks"
_ICLR.cc/2025/Conference — Submitted to ICLR 2025_

### Official Review · Reviewer_MSZa · 2024-10-29

**Soundness:** 3
**Presentation:** 3
**Contribution:** 3
**Rating:** 6
**Confidence:** 3

**Summary:**

This paper analyzes the exact recovery properties of a nonsmooth estimator for nonlinear dynamical system identification problem under adversarial attacks. The authors derive necessary and sufficient conditions for the optimality and uniqueness of the ground truth solution to the formulated problem. Additionally, they provide sample complexity bounds for the exact recovery of model parameters using bounded basis functions and Lipschitz basis functions, respectively.

**Strengths:**

**Originality**

Strengths:
- The paper provides necessary and sufficient conditions for the ground truth to be optimal and unique for the formulated problem.
- The paper derives sample complexity bounds that ensure these conditions are satisfied with high probability for bounded basis functions and Lipschitz basis functions under certain assumptions.
- The paper for the first time studies the sample complexity analysis of a nonsmooth estimator for the nonlinear system identification problem.

**Quality**

Strengths:
- The results of numerical experiments are consistent with the theoretical analysis.
- The formulated problem can be solved using the CVX solver.

**Clarity**

Strengths:
- The paper is generally well-written and easy to follow. Key technical concepts are explained clearly.

**Significance**

Strengths:
- Nonlinear dynamical system identification is an important problem for the areas of sequential decision-making, reinforcement
learning and control theory. Hence, deriving sample complexity bounds  for the exact recovery of bounded basis functions and Lipschitz basis functions are noteworthy.

**Weaknesses:**

1. The simulation problems are relatively small-scale.
2. Some of the mathematical notation and concepts may be difficult to follow.

**Questions:**

1. Some notations are confusing. For example, what is the definition of $\theta()$ in line 280 and $\theta[]$ in Equation (3)? Additionally, how is the matrix in equation 12 constructed?
2. I did not understand the proof of Corollary 3. Could the authors explain it more clearly?
3. Generally, the noise level can affect the accuracy of identification results. Therefore, I suggest that the authors show the identification results under different noise levels.
4. What are the basis functions used in the experiment?
5. In many cases, $A$ has a sparse topology. Empirically, solving problem 2 cannot accurately learn sparse $A$, particularly under high noise levels. However, the theoretical analysis does not depend on the topology of $A$. Hence, could the authors conduct the experiments with a sparse $A$ and explain the corresponding results?

---

> ### Author Response · Authors · 2024-11-16
> **Rebuttal #1**
>
> *The simulation problems are relatively small-scale.*
>
> **Response**: For each experiment, we solve two optimization problems using CVX, obtaining the estimate and the optimality certificate for every time period during a time horizon of $T = 500$ for $10$ random dynamical systems. This requires solving $10,000$ optimization problems for each experiment. To speed up the computations, we used smaller dynamical systems. In our updated paper, we will include larger-scale experiments, where we obtain the estimates every 5 or 10 time periods, rather than at every time period. This approach will allow us to conduct larger experiments at greater scales. On the other hand, it is easy to generate a single very-large-scale problem and solve it numerically. In this work, our plots (for illustration purposes) need solving the problem many times and this affected the size of our simulations.
>
> *Some of the mathematical notation and concepts may be difficult to follow.*
>
> **Response**: Due to space limitations, some notations and concepts may have been dense. We will provide additional explanations in the notation subsection of Section 1 and offering more details immediately after each new concept is introduced.
>
> *Some notations are confusing. For example, what is the definition of $\theta()$ in line 280 and $\theta[]$ in Equation (13)? Additionally, how is the matrix in equation 12 constructed?*
>
> **Response**: We defined the $\Theta$ notation in line 145 of our original manuscript as follows: 'Given two functions $f$ and $g$, the notation $f(x) = \Theta[g(x)]$ means that there exist universal positive constants $c_1$ and $c_2$ such that $c_1 g(x) \leq f(x) \leq c_2 g(x)$.' In the context of line 280, the number of attacked time periods, $|\mathcal{K}|$, can be bounded both below and above by $pT$ with positive constants $c_1$ and $c_2$, i.e., $c_1 pT \le |\mathcal{K}| \le c_2 pT$. This allows us to upper and lower bound $|\mathcal{K}|$ with high probability in finite time. Asymptotically, by the properties of the Bernoulli distribution and the law of large numbers, we expect $\mathcal{K}$ to converge to $pT$.
>
> For equation (13), the notation $\Theta[\cdot]$ follows a similar concept. It implies that we can find a positive constant $c$ such that the number of required samples $T$ for $\bar{A}$ to be the unique solution is lower-bounded by
>
> $$T \geq c\frac{m^2\kappa^4}{p(1-p)^2} \left[ mn\log\left(\frac{m\kappa}{p(1-p)}\right) + \log\left(\frac{1}{\delta}\right) \right].$$
>
> The matrix in equation (12) is constructed by appending $T - |\mathcal{K}|$ column vectors of $f(x_t) \in \mathbb{R}^m$ from the uncorrupted time periods. For example, if attacks occur at every odd time period, i.e., $\mathcal{K} = {1, 3, 5, \dots }$, then the matrix would be constructed by appending the column vectors of $f(x_0), f(x_2), f(x_4), \dots$, which are collected up to time period $T$:
>
> $$\begin{bmatrix}
>     f(x_0) & f(x_2) & f(x_4) & \dots
> \end{bmatrix}$$

---

> ### Author Response · Authors · 2024-11-16
> **Rebuttal #2**
>
> *I did not understand the proof of Corollary 3. Could the authors explain it more clearly?*
>
> **Response**: In Theorem 2, we proved that when equation (4) holds and equation (9) holds for every $Z \neq 0$, then $\bar{A}$ is the unique optimal solution to the proposed estimator. Whenever the condition in Corollary 3 holds, which is
>
> $$\sum_{t\in\mathcal{K}} \hat d_{t}^T Z f(x_t) < \sum_{t\in\mathcal{K}^c}\lVert Z f(x_t)\rVert_2,\quad\forall Z\in\mathbb{R}^{n\times m}\quad\mathrm{s.t.}~ Z\neq 0,$$
>
> we cannot have
>
> $$
> \sum_{t\in\mathcal{K}} \hat d_t^\top Z f(x_t) = \sum_{t\in\mathcal{K}^c}\lVert Z f(x_t) \rVert_2, \quad\forall Z\in\mathbb{R}^{n\times m}\quad\mathrm{s.t.}~ Z\neq 0,
> $$
>
> which is the condition on the left-hand side of equation (9). Therefore, equation (10) is a sufficient condition to guarantee that equation (9) holds. Since the other conditions in Theorem 2 are satisfied as a result of the statement in Corollary 3, this corollary is sufficient for the unique optimality of $\bar{A}$.
>
> *Generally, the noise level can affect the accuracy of identification results. Therefore, I suggest that the authors show the identification results under different noise levels.*
>
> **Response**: Thank you for this comment. Since our estimator treats noise as outliers, as a generalization of LASSO estimator, no matter how small or large those values are, it will be rejected without affecting the number of samples required to do so. This contrasts with the least-squares estimator for which the noise level directly affects the performance. For example, in the case of bounded basis functions, whenever we have uncorrupted data, $f(x_t)$ contracts $x_t$ within an $m$-dimensional cube of magnitude $B$. As a result, even if the noise vector $\bar{d}_t$ is arbitrarily large (but finite), the bounded basis functions ensure that the system remains within this cube. Therefore, we do not expect the magnitude of $\bar{d}_t$ to impact the sample complexity results. The sample complexity result in Theorem 3 supports this claim. However, the number of basis functions $m$ and the bound on the basis functions $B$ do affect the sample complexity because they determine the size of the cube in which the system operates without adversarial attacks.
>
> On the other hand, the required number of samples for learning with Lipschitz basis functions increases with the magnitude of the attack vectors. The magnitude of the attack vectors is influenced by the sub-Gaussian parameter $\sigma$ of $\ell_t$ in Assumption 6. A larger $\sigma$ implies higher variance for the attack vectors and thus, the subsequent states will have a higher variance. As a result, the sample complexity in Theorem 5 increases polynomially with $\sigma$ through the parameter $\kappa$, which is defined as $\kappa := \sigma L / \lambda$.
>
> We will incorporate additional simulation results in our revised manuscript to demonstrate the behavior of the estimator with different sub-Gaussian parameters $\sigma$ for the attack vectors.

---

> > ### Comment · Reviewer_MSZa · 2024-11-24
> >
> > Many thanks for the authors's response. However, my main concerns remain.
> >
> > 1. If equation (9) holds, we can deduce that $\overline{A}$ is the unique optimal solution to the proposed estimator. However, if equation (10) holds, equation (9) cannot be satisfied. Therefore, how can you still conclude that $\overline{A}$ is the unique optimal solution?
> > 2. What do you mean by the statement that the estimator treats noise as outliers, and that these values will be rejected. In my opinion, we do not know whether the observed data is perturbed by noise. Hence, how can the estimator treats noise as outliers and reject these values?
> > 3. Further experimental results are not provided.

---

> > > ### Author Response · Authors · 2024-11-24
> > > **Response to Reviewer**
> > >
> > > Thank you very much for reading our rebuttal.
> > >
> > > - Regarding equations (9) and (10), you may have misread equation (9). This equation says that uniqueness is equivalent to
> > >
> > > "Whenever some equality holds, another inequality must also hold."
> > >
> > > A sufficient condition to satisfy the above statement is the following
> > >
> > > "The inequality in (9) holds all the time (independent of whether the equality in (9) holds).”
> > >
> > > In other words, if the inequality in (10) holds all the time, it will also hold under the equality in (9). That’s why (10) guarantees (9).
> > >
> > >  The source of confusion might be the arrow we have used in (9) to describe the logical constraint. We will rewrite it to clarify what (9) means.
> > >
> > > - In our problem, we do not have noise and we do not assume that the data is subject to noise. To explain our argument, consider the linear system
> > >
> > > $$\bar x_{k+1}=A\bar x_{k} + \bar w_k$$
> > >
> > > where we measure \bar x_k for all values of k from which we aim to find the unknown matrix $A$. Here, $\bar w_k$ is the unknown disturbance/attack on the system and we do not assume it is data noise. Our estimator can be equivalently written as
> > >
> > > $$\min_{A,w_k} ||w_0||+||w_1||+…$$
> > > $$s.t. \bar x_{k+1}=A\bar x_{k} + w_k$$
> > >
> > > This is similar to Lasso but the issue is that the measurements $x_{k}$’s are correlated. If this was a regular Lasso, then $w_k$ were like outliers and Lasso would have found their values independent of their magnitudes. In the Lasso analysis, the sample complexity does not depend on the magnitude of $w_k$ and it is not the case that if an outlier has a larger value, we need more samples. We showed in our paper that this is indeed true even in our correlated case and the estimator finds the correct vectors $w_k$ no matter how small or large the values of $w_k$ are and that the complexity does not directly depend on the values of $w_k$. We tried to draw a parallel between outliers and large disturbances, but we never treated disturbances as outliers in the paper.
> > >
> > > - We will share the numerical results shortly.

---

> > > > ### Author Response · Authors · 2024-11-25
> > > > **Additional Experimental Results**
> > > >
> > > > We added the additional experimental results at the end of the supplementary material highlighted with blue text. We explained the setup for the experiments in the manuscript. We did two types of experiments. First, we set the matrix A to be sparse. Specifically, A is a tridiagonal matrix. All the non-diagonal, non-subdiagonal, and non-superdiagonal are set to zero. The experiments showed that we achieve exact recovery.
> > > >
> > > > Secondly, we did the experiment for the systems with (n,m) = (10,20), (25, 50), (50, 100). We achieve exact recovery in T = 1100 for the case (n,m) =  (50, 100). We observed that the sample complexity increases as expected with theoretical sample complexities obtained in the paper.

---

> > > > > ### Comment · Reviewer_MSZa · 2024-12-02
> > > > >
> > > > > Thank you for answering my questions. I feel confirmed in my decision and I keep my rating.

---

> ### Author Response · Authors · 2024-11-16
> **Rebuttal #3**
>
> *What are the basis functions used in the experiment?*
>
> **Response**: Due to space limitations in our submission, we provide a more detailed description of the experimental setup in Section D of the Appendix. This section outlines the basis functions used, the distribution of the attack vectors, and the process by which the unknown parameters are generated.
>
> For our experiments with the \textbf{Lipschitz basis functions}, Given the state space dimension $n$, we choose $m = n$. Then, we define the basis function as
>
> $$ f(x) := \frac{1}{\sqrt{n}}\begin{bmatrix}
>     \sqrt{\|x - x_1\|_2^2 + 1} - \sqrt{\|x_1\|_2^2 + 1}\\
>     \vdots\\
>     \sqrt{\|x - x_n\|_2^2 + 1} - \sqrt{\|x_n\|_2^2 + 1}\\
> \end{bmatrix},\quad \forall x\in\mathbb{R}^n, $$
>
> where $x_1,\dots,x_n\in\mathbb{R}^n$ are instances of i.i.d. standard Gaussian random vectors. We can verify that the basis function is Lipschitz continuous with Lipschitz constant $L = 1$ and thus, it satisfies Assumption 4 required for Thoerem 5. For our experiments with the \textbf{bounded basis functions}, given the state space dimension $n$, we choose $m = 5n$. Then, we define the basis function as
>
> $$
>     f(x) := \begin{bmatrix}
>         \tilde{f}(x_1)\\
>         \vdots\\
>         \tilde{f}(x_n)\\
>     \end{bmatrix},\quad \text{where }
>     \tilde{f}(y) := \begin{bmatrix}
>         \sin(y)\\
>         \vdots\\
>         \sin(5y)\\
>     \end{bmatrix},\quad \forall x\in\mathbb{R}^n,~y\in\mathbb{R}.
> $$
>
> The basis function satisfies Assumption 1 with $B = 1$.
>
> *In many cases, $A$ has a sparse topology. Empirically, solving problem 2 cannot accurately learn sparse $A$, particularly under high noise levels. However, the theoretical analysis does not depend on the topology of $A$. Hence, could the authors conduct the experiments with a sparse $A$ and explain the corresponding results?*
>
> **Response**: The results in Sections 4 and 5 will hold as long as the assumptions presented in those sections are satisfied. The only assumption required for matrix $\bar{A}$ is the system stability assumption in the case of Lipschitz basis functions. Therefore, our method will correctly learn sparse matrices under high noise levels.
> % Although it may seem counter-intuitive,
> On the other hand, our theory does not establish the sample complexity advantage of our estimator in the case of a sparse $\bar{A}$. In the case when $\bar{A}$ is sparse, we expect the $\ell_1$-estimator to perform better. When the estimator in Problem (2) is non-smooth, we can learn the matrix $\bar{A}$ even when it has a sparse topology. In the literature, when the least-squares estimator is used, non-smooth regularizers are typically included to enforce sparsity in the estimator [1].
>  % Thanks to the non-smoothness, our results in Sections 2 and 3 regarding necessary and sufficient conditions for optimality and unique optimality hold for any matrix $\bar{A}$.
>
> We will conduct numerical experiments with various sparsity structures of the matrix $\bar{A}$, using a setup similar to the one in our experiments. We expect the required number of samples to exhibit similar behavior to the non-sparse case for matrix $\bar{A}$, when the sparse topology is unknown to the learner. However, if the sparse topology is known a priori, specific parameters can be set to $0$ to reduce the number of decision variables in optimization Problem (2), thereby improving the learning rate of the estimator.
>
> *[1] S. Fattahi and S. Sojoudi, "Sample Complexity of Block-Sparse System Identification Problem," in IEEE Transactions on Control of Network Systems, vol. 8, no. 4, pp. 1905-1917, Dec. 2021, doi: 10.1109/TCNS.2021.3089141.*

---

### Official Review · Reviewer_hKKD · 2024-10-31

**Soundness:** 4
**Presentation:** 3
**Contribution:** 2
**Rating:** 5
**Confidence:** 3

**Summary:**

The paper investigates the system identification problem for nonlinear, parameterized systems under adversarial attacks, where for time steps $t=1,...,T$ one observes $x_{t+1} = \bar{A} f(x_t) + {\bar{d_t}}$, with $f$ a known, possibly nonlinear function, $\bar{A}$ the unknown "system dynamics" matrix, and $\bar{d_t}$ the unknown "disturbance" vector, which is adversarially chosen. The class of estimators considered are of the form $\hat{A} = \{argmin} \sum_{t=1}^T \|x_{t+1} - A f(x_t)\|_2$. The nonlinear functions studied in specificity are the class of bounded functions and Lipschitz functions. The paper establishes rigorous theoretical results for exact recovery in nonlinear system identification under adversarial attacks, providing necessary and sufficient conditions for both global optimality and uniqueness of solutions and gives sample complexity upper- and lower-bounds. The authors provide numerical experiments that support the theoretical results.

**Strengths:**

The presentation is overall strong. The paper is well-written and the results are easy to follow, barring the overly wooly introduction/literature treatment (see weaknesses). I think the setting is interesting. The paper is well-structured and the results are clearly presented. I especially appreciate the clarity of the proofs, which are well-structured and easy to follow.

**Weaknesses:**

*Unclear and ambiguous writing in the introduction and lack of coherent literature overview*: The abstract and introduction lack clarity in conveying the actual problem studied, making it difficult for readers to grasp the work's focus early on. The initial (long) paragraphs and literature overview are somewhat vague, imprecise and seem to contribute little to the understanding of the work. Additionally, the literature review extends into the appendix, but it would be more cohesive if integrated into the introduction. This would allow the authors to eliminate ambiguous sections while simultaneously introducing the problem and discussing prior research on different versions of it, providing readers with a clearer understanding the paper’s contributions. Overall, I would like to see more comparison of the estimator studied by the authors with the LSE estimator. The author's contribution becomes difficult to judge because of the lack of context provided.

*In the stochastic adversarial noise case, the authors seem to make some seemingly very strong assumptions on $\hat{d}_t$*. The authors say about Assumption 2 (Stealthy condition) that "If an attack is not stealth, the operator can quickly detect and nullify it". Can the authors formally explain how this would work, and what "quickly" means in this context? Similarly, I am not convinced that the zero-mean, sub-Gaussianity assumption on the disturbance makes sense in this context. In what sense is the disturbance still adversarial in this case? The authors should provide a comparison with the case where $\hat{d}_t$ is just sub-Gaussian observational noise.

**Questions:**

Could the authors comment more on the gaps between Theorem 3 and 4? Particularly in terms of its implications for practical applications? For instance, are there cases where the current bounds could be tightened, or would further assumptions (e.g., regarding the nature of the adversarial noise) be required to bridge this gap? Additionally, are there insights on whether this gap reflects fundamental limitations or merely theoretical conservatism, based on the simulation findings?

---

> ### Author Response · Authors · 2024-11-16
> **Rebuttal #1**
>
> *Unclear and ambiguous writing in the introduction and lack of coherent literature overview: The abstract and introduction lack clarity in conveying the actual problem studied, making it difficult for readers to grasp the work's focus early on. The initial (long) paragraphs and literature overview are somewhat vague, imprecise and seem to contribute little to the understanding of the work. Additionally, the literature review extends into the appendix, but it would be more cohesive if integrated into the introduction. This would allow the authors to eliminate ambiguous sections while simultaneously introducing the problem and discussing prior research on different versions of it, providing readers with a clearer understanding the paper’s contributions. Overall, I would like to see more comparison of the estimator studied by the authors with the LSE estimator. The author's contribution becomes difficult to judge because of the lack of context provided.*
>
> **Response**: We will revise the literature review section in the main part of the manuscript and move the relevant parts from the Appendix into the main manuscript for a better description of our paper. We can briefly summarize the main contributions as follows: The system identification literature predominantly uses the LSE for learning, and there are extensive results regarding the asymptotic properties of this estimator. It was only after 2018 that researchers began to analyze the LSE non-asymptotically, to obtain sample complexity results in the context of learning dynamical systems. The estimation error for the LSE in parameterized non-linear systems scales as $T^{-1/2}$ when the disturbance vectors are independent and identically distributed sub-Gaussian vectors. In the presence of adversarial large noise injections, however, the LSE performs poorly. We cannot extend the existing results from the robust regression literature because they typically assume independence and identical distribution of the data. As a result, the non-asymptotic analysis of system identification in the presence of adversarial vectors has not been studied in the literature until recently.
>
> We found two related works on problem (2) in our paper within the literature, both of which study the sample complexity of system identification for linear dynamical systems only. We generalize their results to parameterized nonlinear systems using bounded and Lipschitz basis functions.
>
> Specifically, the work by Yalcin et al. (2023) is the closest to our work in the literature. Yalcin et al. (2023) only provide sufficient optimality conditions for linear systems, whereas the necessary and sufficient conditions presented in Sections 2 and 3 of our paper are novel and stronger than the conditions in Yalcin et al. (2023). In Sections 4 and 5, we analyze the probabilistic attack structure inspired by Yalcin et al. (2023), where the data are corrupted with probability $p$. Our paper generalizes these results for linear dynamical systems using Lipschitz basis functions. Both Yalcin et al. (2023) and our work make similar assumptions regarding the stability of the system and the distribution of the attack vectors. Their sample complexity result for autonomous dynamical systems grows with $n^2 \log(n)$ with respect to the dimension, $\max\{p^{-1}(1-p)^{-1}, (1-p)^{-2}\}$ with respect to the fraction of corrupted data, and $(1-\rho L)^{-3}$ with respect to the system's stability. Similarly, our result in Theorem 5 for Lipschitz basis functions grows with $n^2$ with respect to the dimension, $\max\{p^{-1}(1-p)^{-1}, (1-p)^{-2}\}$ with respect to the fraction of corrupted data, and $(1-\rho L)^{-3}$ with respect to stability. Thus, we argue that the sample complexity results for Theorem 5 and Yalcin et al. are similar up to a logarithmic factor, with our result being a generalization for any Lipschitz basis function. Furthermore, we do not require the stability assumption (Assumption 5) in the case of bounded basis functions. Note that the stability assumption was critical in the linear case, as it was directly related to the eigenvalues of $A$ and the behavior of $A^t$ as $t$ approaches infinity. As a result, the proof for the bounded case is novel and differs from that in Yalcin et al. (2023).

---

> > ### Author Response · Authors · 2024-11-16
> > **Rebuttal #1 (continued)**
> >
> > Lastly, we would like to emphasize that the LSE cannot achieve exact recovery in our problem setup. Let us define the matrices $f(X) := [f(x_0), \ldots, f(x_{T-1})]$ and $\bar{D} := [ \bar{d}_{0},\dots, \bar{d}_T-1 ]$. The solution to the least-squares problem is $\hat{A} = (\bar{A} f(X) + \bar{D})^T f(X) (f(X)^T f(X))^{-1}$. Consequently, the norm of the estimation error is $\|\bar{D}^T f(X) (f(X)^T f(X))^{-1}\|$, which is non-zero and can become arbitrarily large when the disturbance vectors are arbitrarily large. Since $f(X)$ depends on $\bar{D}$, the expected value of the term inside the norm is not zero. This means that the estimation error is the norm of a vector with a nonzero mean and large variance. As a result, the LSE cannot achieve zero estimation error, leading to a plateau in its estimation error. We conducted additional experiments and observed that the estimation error of the LSE plateaus, while our proposed estimator achieves exact recovery. We will provide new simulations in the revised paper to demonstrate this fact.

---

> ### Author Response · Authors · 2024-11-16
> **Rebuttal #2**
>
> *In the stochastic adversarial noise case, the authors seem to make some seemingly very strong assumptions on $\hat d_t$. The authors say about Assumption 2 (Stealthy condition) that "If an attack is not stealth, the operator can quickly detect and nullify it". Can the authors formally explain how this would work, and what "quickly" means in this context? Similarly, I am not convinced that the zero-mean, sub-Gaussianity assumption on the disturbance makes sense in this context. In what sense is the disturbance still adversarial in this case? The authors should provide a comparison with the case where $\hat d_t$ is just sub-Gaussian observational noise.*
>
> **Response**: Stealthy attacks are considered in the system identification and control theory literature to model the behavior of attack models [1, 2] (these new references are added at the end of this response). In [1], the authors design an anomaly detection system that measures the distance between the observed state of the system and the expected state based on the current best estimate. They state that the controller should stop the system if the residual value exceeds a preset threshold $\tau$. The controller can choose this $\tau$ based on the desired false alarm rate. Therefore, the speed at which anomalies or attacks are detected depends on the hypothesis test constructed by the controller. Similarly, the authors of [2] argue that there is a trade-off between staying undetected and the utility impact of the adversarial region in the case of dynamical adversarial injections into the system. Therefore, a rational adversarial agent should inject zero-mean attack vectors over time to cause the maximum possible damage to the system.
>
> We provide the following simple example to demonstrate how zero-mean attacks could cause significant impact. Consider an energy system with two nodes: node 1 represents a neighborhood of homes, and node 2 is a supplier owned by a utility company. Every five minutes, node 1 reports to node 2 the amount of electrical power the neighborhood needs for the next five minutes. Assume the neighborhood requires a constant amount of $1$ unit of power for the next five hours. However, an attacker has compromised the communication channel between node 1 and node 2, altering the requested amount from $1$ unit to either $1 + e$ or $1 - e$ every half hour, where $e$ is a large number compared to $1$ unit, and its value depends on $\|x_i\|$ at the current or previous time. Since the average of $1 + e$ and $1 - e$ is $1$, this could serve as a stealth attack. When node 1 actually needs one unit of power, but node 2 generates either $1 + e$ or $1 - e$, the mismatch violates the laws of physics, potentially triggering grid instability and leading to a blackout. In this scenario, the attacker infrequently injects an adversarial input into the system that can take arbitrary values. For the attack to remain stealthy, the mean of the attack should be zero (in this case, the mean of $+e$ and $-e$ is zero). Cyberattacks on power systems fit perfectly into our mathematical models. Note that power system operators use hypothesis testing to detect anomalies in data, and if there is a nonzero mean, they would flag it. In the literature, it is commonly assumed that an attacker has a budget. If the attacker has absolute freedom in influencing the system, then it can never be defeated. For that reason, existing papers all make different assumptions about what an attacker is capable to do. If you disagree that zero-directional-mean non-i.i.d. attack is important, we would like to provide a different angle. The existing literature has mainly focused on i.i.d. Gaussian. Our work generalizes Gaussian to correlated disturbance and this is a major generalization. The Gaussian distribution also satisfies all assumptions of our paper and we make a major contribution to the study of non-Gaussian correlated disturbances (which may or may not be an attack).

---

> > ### Author Response · Authors · 2024-11-16
> > **Rebuttal #2 (continued)**
> >
> > Assumption 6 decomposes the attack into angle and magnitude, where its angle should be uniform to remain undetectable, while its magnitude can be arbitrary to maximize the damage to the system. Moreover, the magnitude of the attack vector could be correlated over time. To highlight this, we separate the attack vector into its direction and magnitude. The main difference between the sub-Gaussian attack vectors in Assumption 6 and simple observational sub-Gaussian noise is the correlation over time. In the literature, small observational and/or measurement noise is often assumed to be independent and identically distributed (i.i.d.). In contrast, Assumption 6 imposes no restriction on the independence of the magnitude of the attack vectors over time. Additionally, the attack vectors are not necessarily identically distributed over time. For example, in our experiments with Lipschitz basis functions, we create attack vectors that depend on time. Specifically, for each time instance $t \in \mathcal{K}$, the noise $\bar{d}_t$ is generated by
> >
> > $$ \bar{d}_t := \ell_t \hat{d}_t,\quad \text{where } \ell_t \sim \mathcal{N}(0, \sigma_t^2), ~ \hat{d}_t \sim \mathrm{uniform}(\mathbb{S}^{n-1}), ~ \ell_t \text{ and }\hat{d}_t \text{ are independent}.$$
> >
> > Here, we define $\sigma_t^2 := \min{\|x_t\|_2^2, 1 / n}$. We can verify that the random variable $\ell_t$ is zero-mean and sub-Gaussian with parameter $\sigma = 1$. Additionally, the random vector $\hat{d}_t$ follows the uniform distribution, which satisfies Assumption 6. Note that $\bar{d}_0, \dots, \bar{d}_{T-1}$ are correlated and violate the i.i.d. assumption commonly made in the literature.
> >
> > Regarding you comment on observational noise, we should mention that this is done in the literature as
> >
> > $$ x_{k+1}=f(x_k,w_k), y_{k}=x_k+v_k $$
> >
> > where $x_k$ is the true state while the measurement $y_k$ adds an observational  noise to the state. If we treat $w_k$ as sensor noise (as you suggested), it would violate the fundamentals of control theory (this is also discussed in control theory textbooks). Let us assume that
> >
> > $$ x_1=Ax_0+w_0 $$
> >
> > and $w_0$ is the sensor noise and not disturbance. Then
> >
> > $$ x_2=Ax_1+w_1 $$
> >
> > would be problematic. The reason is that state $x_2$ is generated based on state $x_1$ and that means the observational noise at time 1 affects the internal state of the system in the future (time 2). This is not the case in most dynamical systems. The internal state works on its own and sensor noise happens when we collect data. In other words, no matter if we collect data (for which noise is zero) or do not collect data (for which noise is nonzero), the states of the system should remain the same. For this reason, control theorists always put the noise in the output equation and the disturbance in the state equation.
> >
> > *[1] A. Teixeira, I. Shames, H. Sandberg, and K. H. Johansson, “Revealing stealthy attacks in control systems,”
> > in 2012 50th Annual Allerton Conference on Communication, Control, and Computing (Allerton), 2012, pp. 1806–1813.*
> >
> > *[2] P. Pradhan and P. Venkitasubramaniam, “Stealthy attacks in dynamical systems: Tradeoffs between
> > utility and detectability with application in anonymous systems,” IEEE Transactions on Information
> > Forensics and Security, vol. 12, no. 4, pp. 779–792, 2017.*

---

> ### Author Response · Authors · 2024-11-16
> **Rebuttal #3**
>
> *Could the authors comment more on the gaps between Theorem 3 and 4? Particularly in terms of its implications for practical applications? For instance, are there cases where the current bounds could be tightened, or would further assumptions (e.g., regarding the nature of the adversarial noise) be required to bridge this gap? Additionally, are there insights on whether this gap reflects fundamental limitations or merely theoretical conservatism, based on the simulation findings?*
>
> **Response**: The main purpose of Theorem 4 is to demonstrate the dependence of sample complexity on $\frac{m}{2p(1-p)}$. Therefore, the lower bound in Theorem 4 is not tight, and we expect improvements to it in the future. Moreover, the numerical experiments support the sample complexity bound (upper bound) in Theorem 3, leading us to conjecture that the bound in Theorem 3 is closer to the optimal sample complexity (i.e., the minimal number of samples that guarantees exact recovery with high probability). We leave the study of optimal sample complexity to future work.
>
> Theoretically, the term $\frac{1}{p(1-p)}$ is an artifact of the Bernoulli model (Definition 1). Intuitively, the uniqueness conditions in Section 3 require that $\{f(x_t), t \notin \mathcal{K}\}$ spans the full space $\mathbb{R}^m$. Therefore, it is necessary to have at least $m$ attacks in the first $T$ steps with high probability, and the probability of this is proportional to $\exp[-C p(1-p) / m]$ for some constant $C > 0$. More rigorously, in Theorem 4, we provide a lower bound proportional to $\frac{m}{p(1-p)}$, which partially justifies the dependence on $\frac{m}{p(1-p)}$.
>
> Note that the main question addressed in the paper is whether it is possible to learn a dynamical system with zero error in finite time when more than half of the data are arbitrarily corrupted. We show that this is possible by providing an upper bound on the required time horizon. This result has major implications for real-world systems. On the other hand, the lower bound is mostly theoretical. Unlike machine learning problems, where the size of a learning problem (such as the number of parameters in a deep neural network) may reach billions, the number of states in many real-world systems is much smaller, often fewer than several thousand. For this reason, our upper bound is already practical, and improving the lower bound may have marginal practical value, though it remains an important and interesting theoretical problem.

---

> ### Comment · Reviewer_hKKD · 2024-11-23
>
> I thank the authors for the effort put forward in their rebuttal.
>
> Rebuttal #1: These sound like reasonable improvements.
>
> Rebuttal #2: I think the fact that there is other literature out there considering such settings is not necessarily an argument for the applicability of the scenario to real life problems. Similarly, the example provided above seems a bit contrived. Are these attacks necessarily "smarter" attacks than other types of attacks? Why would I worry about these attacks over other types of attacks? I am still not convinced that these attacks are generally interesting beyond the fact that they suit the proofs of the authors. The choice for Gaussian noise is still poorly defended in my opinion.
>
> Rebuttal #3: This is now clear to me, thanks.
>
> Overall, I wish to maintain my score.

---

> > ### Author Response · Authors · 2024-11-23
> > **Response to Reviewer**
> >
> > Thank you very much for reading our responses.
> >
> > - The example we provided is not contrived and this is how the Russian attack on the Ukrainian power grid happened in 2015. This type of attack is the main source of cybersecurity issue with power grids. There have been several smaller-scale attacks in different parts of the world on power systems and they are all based on changing the data so that the dynamical system would violate laws of physics and move away from its main equilibrium. We explained the concept in a 2-node system for illustration purposes but the general concept is exactly the same.
> >
> > - We are not sure what you mean by the “choice for Gaussian noise”. An attack is unlikely to be Gaussian and our method is designed to be applicable to non-Gaussian disturbances in the bounded basis function case. In the Lipschitz basis case, the sub-Gaussian assumption applies to the magnitude of the attack and this ensures that the system state does not deviate from "normal" trajectories; otherwise, the system operator may be able to detect the attack.
> >
> > - The literature has mainly focused on i.i.d. Gaussian-type disturbances. We are not aware of any work that provides a non-asymptotic/finite-time learning for correlated disturbance for nonlinear systems. Our paper is the first one offering this result. I think there is some disagreement here about where $w(t)$ serves as an attack, but if we look at it as correlated disturbance, our work significantly generalizes the existing literature. Our model of disturbance includes the existing ones in the literature. Aside from this, we called $w(t)$ an attack since its value is arbitrary. In some literature, they call it semi-oblivious since we have some restrictions on its direction but its magnitude is designed by an attacker and is completely arbitrary. We would be happy to revise our paper and mention that our work focuses on correlated disturbances and semi-oblivious attacks. As we mentioned, attacks in power systems fall into this category, and regarding your question on whether there are smarter attacks, the issue is that such smarter attacks would be detectable by power operators. An attacker would theoretically aim for the worst attack but if it is easily detectable, it should scale down the attack to avoid nullification. That is why there are always assumptions on the attack types in different papers.
> >
> > Thanks again for the follow-up response! If our response is able to address the concerns #2, we hope that the reviewer could kindly raise the score.

---

> > ### Author Response · Authors · 2024-11-24
> > **Response to the reviewer (continued)**
> >
> > We would like to provide more details on the response we submitted yesterday. The fact is that exact recovery is not possible in general if the attacks are arbitrary. For example, consider the system
> >
> > $$x_{k+1}=Ax_k+w_k$$
> >
> > where the matrix $A$ is unknown. If the attacker chooses the attack to be $-Ax_k$, then $x_{k+1}$ is zero. The attacker can repeat this strategy to make all the states to be zero. Then, it is impossible to learn $A$ from a zero trajectory. This means that recovery is impossible without making any assumption on the attack $w_k$. On the other hand, an attacker cannot implement such extreme attack since it would be easily detectable. In most literature about adversarial attacks, there are assumptions made about the attacks even if the attacks are considered to be adversarial. Common assumptions include the bound on the magnitude and/or the frequency of attacks. In our work, Assumption 6 is the assumption on attacks and our theory guarantees the exact recovery under this condition. Although our assumption may seem strong, it is a reasonable choice of the attack assumption for many reasons. In our model, we assume that the direction of the attack is uniform abut the magnitude is arbitrary. To make this assumption milder, we can assume that the undesirable directions that an attacker will not choose have a tiny magnitude (say epsilon). Then, technically those directions that an attacker will not choose act as a small noise with magnitude epsilon. We hope this addresses your concern to some extent and we would be happy to discuss this idea in the revised paper.
> >
> > The requirement on the attack frequency is necessary for the trajectory to explore the whole state space (otherwise, we can establish similar exact recovery results on a subspace of the state space, where the trajectory has explored). On the other hand, the attack frequency condition can be removed if the system operator injects small noise to excite the exploration of the state space, which is a commonly used technique in control literature. Moreover, our Theorem 4 also shows the necessity of attack frequency when there is no injected noise.
> >
> > In summary, our assumption on the attacks is not artificially constructed to suit our proofs. Instead, it is a reasonable model for many real-world attacks and our theory is broadly applicable to these applications. Also, we would like to mention that our work provides the first exact recovery guarantee for the case when the data does not satisfy the i.i.d. assumption. It may be possible to further improve the results and relax the assumptions in this work, but this work serves a meaningful first step in the exact recovery theory for autocorrelated systems. We would like to request that you kindly re-evaluate our work given the fact that none of the existing works have been able to get results on correlated disturbance for nonlinear systems.

---

> > > ### Comment · Reviewer_hKKD · 2024-11-27
> > >
> > > Thanks again for the detailed rebuttal. While you have provided some justification for your assumptions in the rebuttal, I find it still difficult to judge the framework of the article in full as such explanations are for a large part lacking from the original paper. The fundamental motivation for your assumptions should be clearly presented in the paper itself, ideally in the section where you introduce your attack model. The connection between Assumptions 2, 6, the probabilistic attack model and practical attack detection/prevention remains unclear in the manuscript. Your rebuttal suggests that extreme attacks would be "easily detectable," but the paper should explicitly discuss why your probabilistic attack model reasonably captures realistic scenarios where attackers try to avoid detection. While your rebuttal explains that the attack frequency requirement enables state space exploration, such insights are not well-developed in the paper. If these assumptions where needed to make the proofs work, a discussion of the important technical steps that use these assumptions would have been appropriate.
> > >
> > > The paper needs to better justify its technical choices and setting to its readers -- in addition to explaining its position in regards to existing literature. While I appreciate the effort put into the rebuttal, I am not sufficiently assured that these issues can be salvaged to raise my current score.

---

> > > > ### Author Response · Authors · 2024-11-27
> > > > **New Section Added to Paper Explaining Attack Models**
> > > >
> > > > Dear Reviewer hKKD: We greatly appreciate the time and effort you have put to review our responses and provide constructive feedback. Based on your comment, we uploaded a revision that has a new Section 4 named “Attack Model.” In this section, we explain the notions of attack problem and defense problem, and provide examples on why an attacker should not try to implement “extreme” attacks. We also added a new theorem to explain why the assumptions made in the paper are necessary. We believe that this new section has enhanced the quality of our work and clarified major issues about the reasonings behind our attack model. We will continue to improve our paper (for example by adding more references) but since the submission deadline is today, we decided to focus on this new section for now and re-upload the latest version before the system closes. We hope that you can kindly re-evaluate our work. Many thanks.

---

> > > > > ### Author Response · Authors · 2024-11-30
> > > > > **Regarding Revised Paper**
> > > > >
> > > > > Dear  Reviewer hKKD: Since the discussion period will end in two days, could you please read the revised paper and let us know if the added materials have sufficiently addressed your concerns? Many thanks.

---

### Official Review · Reviewer_vTY1 · 2024-11-02

**Soundness:** 2
**Presentation:** 3
**Contribution:** 1
**Rating:** 3
**Confidence:** 3

**Summary:**

This paper studies the problem of learning the dynamical system x_{t+1} = Af(x_t) + d_t where A is unknown, f is known, and d_t is adversarial but often 0 (from a single trajectory). The main results study the estimator argmin_{A'} sum_t ||x_{t+1} - A'f(x_t)|| in two settings:
1) f is bounded; the times when d_t is nonzero are i.i.d. with some probability p; there is some non-degeneracy in the directions of the adversarial noise; and the direction of the adversarial noise is mean-0. Under these assumptions, it's shown that with a poly(dimension, condition number, 1 / p(1-p)) length trajectory, the above estimator has a unique optimum at A. A matching lower bound is also shown.
2) f is Lipschitz; same non-degeneracy condition as above; the adversarial noise has mean-0 direction and is sub-Gaussian; and the system is stable. Again under these assumptions a similar upper bound on the length of the needed trajectory is shown.

**Strengths:**

Learning non-linear dynamical systems is an interesting problem. The paper is written clearly.

**Weaknesses:**

For a paper that purports to study an adversarial-noise setting, the assumptions seem absurdly strong.
- Why should the direction of the adversarial noise be mean-0? The paper gives some justification that otherwise it wouldn't be "stealth" and that the operator could detect and nullify such noise, but this is not at all clear to me. How would d_t be detected if it were not mean 0? We only see one trajectory. How are the mentioned references related? I did not see a comparable assumption in those works. Also, why mean-zero direction instead of unnormalized mean-zero?
- Why is Assumption 3 reasonable (that the attacks have non-degenerate directions)?

In some ways it seems that the goal of unique recovery of A is far too strong. Presumably it would be more reasonable to ask to learn any equivalent dynamics?
- This might get rid of the non-degeneracy assumption
- Also, the main results **get worse as the fraction of adversarially-affected time-steps decreases**. This is completely absurd and suggests that the problem should be set up differently.

Finally, all of the results and lower bounds are specific to the proposed autoregressive least squares estimator. Could there be a better estimator? This paper does not seem to address that question.

**Questions:**

See above

---

> ### Author Response · Authors · 2024-11-16
> **Rebuttal #1**
>
> *For a paper that purports to study an adversarial-noise setting, the assumptions seem absurdly strong. Why should the direction of the adversarial noise be mean $0$? The paper gives some justification that otherwise it wouldn't be "stealth" and that the operator could detect and nullify such noise, but this is not at all clear to me. How would $d_t$ be detected if it were not mean $0$? We only see one trajectory. How are the mentioned references related? I did not see a comparable assumption in those works. Also, why mean-zero direction instead of unnormalized mean-zero?*
>
> **Response**: Stealthy attacks are commonly considered in the system identification and control theory literature to model attack behaviors [1, 2](these new references are given at the end of this response). In [1], the authors design an anomaly detection system that measures the distance between the observed state of the system and the expected state based on the current best available estimate. They state that the controller should stop the system if the residue value exceeds a preset threshold $\tau$. The controller can choose $\tau$ based on the desired false alarm rate. Therefore, the speed at which anomalies or attacks are detected depends on the hypothesis test constructed by the controller. Similarly, the authors of [2] suggest that there is a trade-off between staying undetected and the utility impact of the adversarial region when dealing with dynamical adversarial injections into the system. A rational adversarial agent, in this case, should inject zero-mean attack vectors over time to cause the maximum possible damage to the system. You argued that since it is one trajectory, nonzero mean may be stealth too. As a real-world example, we should mention that power systems have been around for many years and system operators perform hypothesis testing every 5 minutes to detect anomalies in data. Since it is a single trajectory, they select a small window and take the average over the window as a rough surrogate for the mean at a given time. Power system state estimation has worked for years under the same assumption that an attacker would design a zero mean attack to pass the hypothesis testing module. You may argue that this is not the definition of stealth you have in mind, but it has been quite successful for the US Power Grid with over 80,000 nodes.
>
> We provide a simple example to demonstrate how zero-mean attacks could have a significant impact. Consider an energy system with two nodes: node 1 represents a neighborhood of homes, and node 2 is a supplier owned by a utility company. Every five minutes, node 1 reports to node 2 the amount of electrical power the neighborhood needs for the next five minutes. Assume the neighborhood requires a constant amount of $1$ unit of power for the next five hours. However, an attacker has compromised the communication channel between node 1 and node 2, altering the requested amount from $1$ unit to either $1 + e$ or $1 - e$ every half hour, where $e$ is a large number compared to $1$ unit, and its value depends on $|x_i|$ at the current or previous time. Since the average of $1 + e$ and $1 - e$ is $1$, this could serve as a stealthy attack. When node 1 actually needs one unit of power, but node 2 generates either $1 + e$ or $1 - e$, the mismatch violates the laws of physics, potentially triggering grid instability and leading to a blackout. In this scenario, the attacker infrequently injects an adversarial input into the system that can take arbitrary values. For the attack to remain stealthy, the mean of the attack should be zero (in this example, the mean of $+e$ and $-e$ is zero). Cyberattacks on power systems fit perfectly into our mathematical models. Note that power system operators use hypothesis testing to detect anomalies in data, and if there is a nonzero mean, they would flag it.

---

> > ### Author Response · Authors · 2024-11-16
> > **Rebuttal #1 (continued)**
> >
> > Assumption 6 decomposes the attack into angle and magnitude, where the angle should be uniform to remain undetectable, while the magnitude can be arbitrary to maximize damage to the system. Moreover, the magnitude of the attack vector could be correlated over time. To highlight this, we separate the attack vector into its direction and magnitude. The main difference between the sub-Gaussian attack vectors in Assumption 6 and simple observational sub-Gaussian noise is the correlation over time. In the literature, small observational and/or measurement noise is typically assumed to be independent and identically distributed (i.i.d.). On the other hand, Assumption 6 places no restrictions on the independence of the attack vector's magnitude over time, and the magnitudes are not necessarily identically distributed. For example, in our experiments with Lipschitz basis functions, we create attack vectors that are dependent over time. Specifically, for each time instance $t \in \mathcal{K}$, the noise $\bar{d}_t$ is generated by
> >
> > $$ \bar{d}_t := \ell_t \hat{d}_t,\quad \text{where } \ell_t \sim \mathcal{N}(0, \sigma_t^2), ~ \hat{d}_t \sim \mathrm{uniform}(\mathbb{S}^{n-1}), ~ \ell_t \text{ and }\hat{d}_t \text{ are independent}.$$
> >
> > Here, we define $\sigma_t^2 := \min{\lVert x_t\rVert_2^2, 1 / n}$. We can verify that the random variable $\ell_{t}$ is zero-mean and sub-Gaussian with parameter $\sigma=1$. Additionally, the random vector $\hat{d}_t$ follows a uniform distribution, thus satisfying Assumption 6. Note that the vectors $\bar{d}_0,... ,\bar{d}_{T-1}$ are correlated, violating the i.i.d. assumption commonly found in the literature.
> >
> > In the literature, it is commonly assumed that an attacker has a budget. If the attacker has absolute freedom in influencing the system, then it can never be defeated.  For that reason, existing papers all make different assumptions about what an attacker is capable to do. If you disagree that zero-directional-mean non-i.i.d. attack is important, we would like to provide a different angle. The existing literature has mainly focused on i.i.d. Gaussian. Our work generalizes Gaussian to correlated disturbance and this is a major generalization. The Gaussian distribution also satisfies all assumptions of our paper and we make a major contribution to the study of non-Gaussian correlated disturbances (which may or may not be an attack).We will add  more exposition to the paper based on your questions.
> >
> > *[1] A. Teixeira, I. Shames, H. Sandberg, and K. H. Johansson, “Revealing stealthy attacks in control systems,” in 2012 50th Annual Allerton Conference on Communication, Control, and Computing (Allerton), 2012, pp. 1806–1813.*
> >
> > *[2] P. Pradhan and P. Venkitasubramaniam, “Stealthy attacks in dynamical systems: Tradeoffs between utility and detectability with application in anonymous systems,” IEEE Transactions on Information Forensics and Security, vol. 12, no. 4, pp. 779–792, 2017.*

---

> ### Author Response · Authors · 2024-11-16
> **Rebuttal #2**
>
> *Why is Assumption 3 reasonable (that the attacks have non-degenerate directions)?*
>
> **Response**: In order to learn the system dynamics over the entire state space, we require the system to explore all subspaces of the state space. If the states of the system, in the presence of attacks, are degenerate or restricted to a strict subspace, we can only learn the dynamics within that subspace, while the behavior of the system outside of that subspace will remain unknown.
>
> For example, consider a two-dimensional linear dynamical system of the form $x_{t+1} = \bar{A} x_t + \bar{d}_t$. Suppose the system dynamics are given by
>
> $$\begin{bmatrix}
>         x^1_{t+1} \\ x^2_{t+1}
>     \end{bmatrix} = \begin{bmatrix}
>         \bar a_1 & 0 \\ 0 & \bar a_2
>     \end{bmatrix}\begin{bmatrix}
>         x^1_{t} \\ x^2_{t}
>     \end{bmatrix} + \begin{bmatrix}
>         \bar d^1_{t} \\ \bar d^2_{t}
>     \end{bmatrix}$$
>
> If $x_0 = 0$ and the attack vectors are restricted to the first dimension only, i.e., $\bar{d}^2_t = 0$ for all $t$, then $x^2_t = 0$ for all $t$. Since there is no exploration in the second dimension, it becomes impossible to recover $\bar{a}_2$. As a result, we need to ensure that every subspace of the state space is explored, which leads to enforcing a non-degenerate assumption to guarantee exact recovery. System identification in control theory has always relied on the notion of excitation and it is known that system identification is not possible without that in general. Assumption 3 aims to make sure that the system is excited via disturbances.

---

> ### Author Response · Authors · 2024-11-16
> **Rebuttal #3**
>
> *In some ways it seems that the goal of unique recovery of A is far too strong. Presumably it would be more reasonable to ask to learn any equivalent dynamics? This might get rid of the non-degeneracy assumption.*
>
> **Response**: Thank you for this comment. When we have a system in the form of $x_{k+1}=Ax_k+Bu_{k}$, $y_k=Cx_k$, then the input-output behavior remains the same if we change the system dynamics $(A,B,C)$ to $(TAT^{-1},TB,CT^{-1})$ for any matrix $T$. This is the commonly used notion of equivalent dynamics. In that case, there is no unique solution. However, in our setting, we measure $x_k$ directly and in that case given an input-state sequence, the linear dynamics generating the data sequence is unique with probability 1. So, there is no notion of equivalence to leverage here. The same story holds for nonlinear systems parametrized by well-behaved basis functions. Aside from this, we believe that the unique recovery assumption is not too strong. In the extreme case, if the system is initialized at the origin and there are no adversarial injections into the system, the system will remain at the origin at every time step. In this case, any matrix would be a valid solution to the problem (2), leading to infinitely many solutions. This is not a desirable outcome for safety-critical systems, where accurate learning is of utmost importance.
>
> If the non-degeneracy assumption is not satisfied, it implies that the system dynamics are not explored in some subspaces of the state space. In such a scenario, we could focus on learning within that subspace alone. In this case, Assumption 3 could be relaxed to focus on the smallest nonzero eigenvalue of the matrix to learn the system dynamics within that subspace. However, this is not a desired outcome for safety-critical systems. Therefore, the controller should inject small exploration noise into the system whenever sufficient exploration is not achieved through adversarial injections, to ensure the system dynamics can be uniquely learned.

---

> ### Author Response · Authors · 2024-11-16
> **Rebuttal #4**
>
> *Also, the main results get worse as the fraction of adversarially-affected time-steps decreases. This is completely absurd and suggests that the problem should be set up differently.*
>
> **Response**: Thank you for this important observation. This counter-intuitive phenomenon is long known in control theory through the notion of excitation which appears even in learning deterministic systems without any disturbances or attacks. To explain this with more details, when $p$, the fraction of adversarially-affected time steps, is very small or even zero, learning the system becomes a classic problem in control theory. It is well known that artificial noise (referred to as an excitation signal) must be added to the system to enable learning. There is a rich literature explaining why an excitation signal is necessary when a system is (nearly) deterministic. For instance, consider the system $x_{i+1} = \bar{A} x_i$. If $x_0 = 0$, then $x_i$ will always remain zero, preventing us from identifying $\bar{A}$. To overcome this, we must excite the system, i.e., modify the dynamics to $x_{i+1} = \bar{A} x_i + w_i$, where $w_i$ is, for example, a Gaussian noise. When $p$ is not close to zero, the adversarial attack serves as an excitation signal, assisting in this regard.
>
> Moreover, we proved theoretically that the dependence $p^{-1}(1-p)^{-1}$ cannot be improved, using a carefully designed counterexample in Theorem 4. The term $\frac{1}{p(1-p)}$ is an artifact of the Bernoulli model (Definition 1). Intuitively, the uniqueness conditions in Section 3 require that the set $\{f(x_t), t \notin \mathcal{K}\}$ spans the full space $\mathbb{R}^m$. Therefore, at least $m$ attacks are necessary in the first $T$ steps with high probability. The probability of this event is proportional to $\exp[-C p(1-p) / m]$ for some constant $C > 0$. More rigorously, in Theorem 4, we provide a lower bound that is proportional to $\frac{m}{p(1-p)}$, which supports the necessity of the dependence on $\frac{m}{p(1-p)}$. Since our focus is on adversarial cases, specifically for $p > 1/2$, and the classical system identification literature addresses how to solve the uncorrupted problem, we did not delve into the case where the fraction of corrupted data is small. If $p$ is small, as commonly done in system identification textbooks, we should add an i.i.d. Gaussian noise to the system to excite it to make it away from a deterministic system. This has been known since 80s and we avoided this classic case in the paper. We will add more discussions to the paper in response to your comment.

---

> ### Author Response · Authors · 2024-11-16
> **Rebuttal #5**
>
> *Finally, all of the results and lower bounds are specific to the proposed auto-regressive least squares estimator. Could there be a better estimator? This paper does not seem to address that question.*
>
> **Response**:  The set of estimators studied in the system identification literature are very limited. The most of the literature focus on the least-squares estimator due to the availability of the closed-form analytical solution. A simple analysis of the least-squares estimator shows that the exact recovery in finite time in our problem setup with the least-squares estimator is impossible. The application of other non-smooth estimators in the literature are limited to Wasserstein distance minimization and our estimator in problem (2) with different norm choices. Moreover, the non-asymptotic analysis of any non-smooth estimator for the system identification problem is done for the estimator in problem (2) with $\ell_2$ and $\ell_1$ norm choices. Since the convex estimator presented in problem (2) is more fitted for theoretical analysis and practical application in real-life system, we analyzed this estimator.
>
> The correlation of the data over time make the analysis complicated for any arbitrary estimator. The estimators involving loss functions that are used in robust learning literature, such as Huber loss, minimax loss and non-smooth regularizers, could be used in this context as well. We focused on the non-smooth estimators in this work as an initial step in the exact recovery theory. Nevertheless, the theoretical analyses of these estimators are important future works.
>
> There is no paper in the existing literature that shows a dynamical system under attack can be learned precisely in finite time. This is the main message of the paper, which is against all existing results that provide asymptotic convergence.

---

> > ### Comment · Reviewer_vTY1 · 2024-11-23
> >
> > Thanks for the detailed response. It does not alleviate my main concern, which is that the assumptions on the noise are so strong (uniform direction, induces exploration, occurs frequently) that this cannot be fairly called an "adversarial noise" model. It would be more interesting if the model had an operator-induced stochastic noise component and an arbitrary adversarial noise component, and recovery were possible under some mild norm bound or sparsity bound on the adversarial component. Is this a corollary of the paper's results?

---

> > > ### Author Response · Authors · 2024-11-23
> > > **Response to the reviewer**
> > >
> > > Thank you very much for reading our responses.
> > >
> > > - The example we provided in the response showcased an important and common attack scheme that appeared in real-world attacks. For example, this is how the Russian attack on the Ukrainian power grid happened in 2015. This type of attack is the main source of cybersecurity issue with power grids. There have been several smaller scale attacks in different parts of the world on power systems and they are all based on changing the data so that the dynamical system would violate laws of physics and move away from its main equilibrium. We explained the concept in a 2-node system for illustration purposes but the general concept is exactly the same.
> > >
> > > - Our Assumption 6 is a mathematical model for this type of attack and our theory guarantees the exact recovery under this case.  The literature has mainly focused on i.i.d. Gaussian-type disturbances. We are not aware of any work that provides a non-asymptotic/finite-time learning for correlated disturbance for nonlinear systems. Our paper is the first one offering this result. As we mentioned, attacks in power systems fall into this category, and regarding your question on whether there are smarter attacks, the issue is that such smarter attacks would be detectable by power operators. An attacker would theoretically aim for the worst attack but if it is easily detectable, it should scale down the attack to avoid nullification. That is why there are always assumptions on the attack types in different papers.
> > >
> > > - On the other hand, if there is no stealthy assumption on the attack and the attacks can take arbitrary values, the attacker can always make $x_t=0$ for all $t$ and it makes the exact recovery impossible (although in practice, it would be easily detectable by power operators). Therefore, our assumption on the attack distribution is a reasonable condition, under which the system operators may not be able to easily detect the attacks and the attackers can avoid attack nullification.
> > >
> > > Thanks again for the follow-up response! If our response is able to address the concerns, we hope that the reviewer could kindly raise the score.

---

> > > ### Author Response · Authors · 2024-11-24
> > > **Response to the reviewer (continued)**
> > >
> > > We would like to provide more details on the response we submitted yesterday. The fact is that exact recovery is not possible in general if the attacks are arbitrary. For example, consider the system
> > >
> > > $$x_{k+1}=Ax_k+w_k$$
> > >
> > > where the matrix $A$ is unknown. If the attacker chooses the attack to be $-Ax_k$, then $x_{k+1}$ is zero. The attacker can repeat this strategy to make all the states to be zero. Then, it is impossible to learn $A$ from a zero trajectory. This means that recovery is impossible without making any assumption on the attack $w_k$. On the other hand, an attacker cannot implement such extreme attack since it would be easily detectable. In most literature about adversarial attacks, there are assumptions made about the attacks even if the attacks are considered to be adversarial. Common assumptions include the bound on the magnitude and/or the frequency of attacks. In our work, Assumption 6 is the assumption on attacks and our theory guarantees the exact recovery under this condition. Although our assumption may seem strong, it is a reasonable choice of the attack assumption for many reasons. In our model, we assume that the direction of the attack is uniform abut the magnitude is arbitrary. To make this assumption milder, we can assume that the undesirable directions that an attacker will not choose have a tiny magnitude (say epsilon). Then, technically those directions that an attacker will not choose act as a small noise with magnitude epsilon. We hope this addresses your concern to some extent and we would be happy to discuss this idea in the revised paper.
> > >
> > > The requirement on the attack frequency is necessary for the trajectory to explore the whole state space (otherwise, we can establish similar exact recovery results on a subspace of the state space, where the trajectory has explored). On the other hand, the attack frequency condition can be removed if the system operator injects small noise to excite the exploration of the state space, which is a commonly used technique in control literature. Moreover, our Theorem 4 also shows the necessity of attack frequency when there is no injected noise.
> > >
> > > In summary, our assumption on the attacks is not artificially constructed to suit our proofs. Instead, it is a reasonable model for many real-world attacks and our theory is broadly applicable to these applications. Also, we would like to mention that our work provides the first exact recovery guarantee for the case when the data does not satisfy the i.i.d. assumption. It may be possible to further improve the results and relax the assumptions in this work, but this work serves a meaningful first step in the exact recovery theory for autocorrelated systems. We would like to request that you kindly re-evaluate our work given the fact that none of the existing works have been able to get results on correlated disturbance for nonlinear systems.

---

> ### Author Response · Authors · 2024-11-26
> **New Theorem**
>
> We are not sure if you got a chance to read our rebuttal but to further respond to your comment about the assumption in our paper, we would like to add the following theorem:
>
> Theorem: Consider the linear system $x_{k+1}=Ax_{k}+w_k$ where $A$ is unknown. Assume that at each time period $w_k$ is zero with probability 1-p and is selected randomly from a subspace M with probability p. If the dimension of M is less than n, then there does not exist any estimator that can correctly identify A from the trajectory $x_{k}$’s all the time.
>
> The above theorem says that if $w_k$ is restricted to a subspace that does not cover the entire space, then it is impossible to learn the system correctly. This means that the assumption in out paper about $w_k$ covering all directions is essential and in this regard our assumption is not strong but rather necessary. As we explained in our previous response, in areas like power systems, the operator has a hypothesis testing scheme in place to detect suspicious attacks and if an attack is always biased in some direction, the operator would easily detect it. So, our assumption is not only necessary but also practical.We would be happy to add the above theorem to the paper if this addresses your concern.
>
> To prove the above theorem, consider a symmetric matrix A with a repeated eigenvalue lambda whose eigenspace is M. Then, if $x_0$ is zero, it can be shown that $x_k$ always belongs to M. Now, if another eigenvalue of A changes without changing lambda and its corresponding eigenspace, the measured states $x_k$ would remain the same. This means that there are multiple systems corresponding to the same measurements and this is an observability issue and no estimator can find the correct system all the time.

---

> > ### Author Response · Authors · 2024-11-30
> > **Regarding Responses and Updated Paper**
> >
> > Dear Reviewer vTY1: We have provided several responses to your previous comment that has unfortunately called our assumptions "absurdly strong” and are still waiting to hear back from you. We would greatly appreciate it if you could check our responses and the updated paper. As we discussed in both our responses and the revised paper, the assumptions are not an artifact of our method and indeed the learning problem is meaningless without these assumptions (if an attacker can arbitrarily attack a system, then there is no learning mechanism to detect the attacks from the observed state trajectory). We added a new theorem to the paper to explain this fact. We hope that you would re-consider your evaluation. Many thanks.

---

### Official Review · Reviewer_cHm8 · 2024-11-10

**Soundness:** 3
**Presentation:** 3
**Contribution:** 3
**Rating:** 6
**Confidence:** 3

**Summary:**

The paper studies a (nonlinear) system identification problem. The authors prove necessary and sufficient conditions for optimality in the general case (when the data is corrupted by an adaptive adversary, assuming that not all the samples are corrupted), and study a random version of the problem, when the corruptions are zero or nonzero independently from each other.

**Strengths:**

The problem seems to be much more general than the version studied before in literature (e.g. [Yalcin et al. (2023)] or [Feng & Lavaei (2021)]). This generalization looks interesting and non-trivial to me, and the results are strong.

**Weaknesses:**

I didn't find major weaknesses, but since the results are on a generalization of the (liner version of the) problem that was studied before, I would like to see a more detailed comparison with the analysis from prior work. While the authors provided some insights on pages 2 and 4, it is not very clear to me what is the technical contribution of the paper.

**Questions:**

**Typos:** line 696, Farks’s -> Farkas'

**Questions**:

In line 229 you compare your result with [Yalcin et al. (2023)], Figure 1 from their paper. Since your result is theoretical, and you simplify it for large $\Delta$ for this comparison, could you compare it not with the table of values, but with their theoretical guarantees?

Also, your randomized version of the problem looks very similar to linear regression with oblivious adversary (see, for example [1] and [2]), and your Assumption 3 reminds the strong convexity assumption from [2]. Could you please check if the models or their analysis of those papers have any connection to your result? If it is the case, it might be nice since their results were generalized to sparse regression and other problems (see [3], they used the symmetry assumption that is somewhat similar to your stealthy condition), so it might be interesting if there exists a general framework that captures also your problem.

One other question (somewhat related to the previous one): you have a $\log m$ factor in Theorem 3. Do you think it is an artefact of your proof, or do you expect it to be inherent?

[1] Arun Sai Suggala, Kush Bhatia, Pradeep Ravikumar, Prateek Jain. Adaptive Hard Thresholding for Near-optimal Consistent Robust Regression.

[2] Tommaso d'Orsi, Gleb Novikov, David Steurer. Consistent regression when oblivious outliers overwhelm.

[3] Tommaso d'Orsi, Rajai Nasser, Gleb Novikov, David Steurer. Higher degree sum-of-squares relaxations robust against oblivious outliers.

---

> ### Author Response · Authors · 2024-11-16
> **Rebuttal #1**
>
> *I didn't find major weaknesses, but since the results are on a generalization of the (linear version of the) problem that was studied before, I would like to see a more detailed comparison with the analysis from prior work. While the authors provided some insights on pages 2 and 4, it is not very clear to me what is the technical contribution of the paper.*
>
> **Response**: As stated in the literature review of our manuscript, non-asymptotic analyses for the system identification problem with non-smooth estimators are very limited. The only relevant paper we found was Yalcin et al. (2023), which examines the linear system identification problem using the same estimator. Linear dynamical systems could be written as $x_{t+1} = \bar A x_t + \bar B u_t + \bar d_t$, where $u_t$ are the given exogenous input vector. The sample complexity results for Theorem 5 and Yalcin et al. (2023) are similar up to a logarithmic factor, yet our result is the generalization to any Lipschitz basis function. Furthermore, we do not need the stability assumption (Assumption 5) in the case of a bounded basis function (note that the stability assumption was the key in the linear case since it was directly related to the eigenvalues of A and the behavior of $A^t$ when t goes to infinity). As a result, the proof for the bounded case is novel and different with those in Yalcin et al. (2023).
>
> Compared with Yalcin et al. (2023), the non-linear basis function in our work makes it infeasible to directly analyze the optimization problem by writing out the explicit expression of $x_t$, as shown in the proof of Theorem 2 in Yalcin et al. (2023). Note that when the system is in the form of $x_{t+1}=Ax_t$, then $x_t$ can be directly written as $A^t x_0$ and we only need to analyze the eigenvalues of A. For a nonlinear system in the form of $x_{t+1}=f(x_t)$, writing $x_t$ in terms of $x_0$ needs the composition of t functions and this cannot be done analytically. There is no counterpart of linear-system eigenvalue analysis for nonlinear systems. This challenge is well-documented in many nonlinear systems textbooks in control theory, and, consequently, several results known for linear systems do not have counterparts in the nonlinear setting. Therefore, we took a different approach to estimate the terms that appear in the uniqueness condition (10) in Section 3.
>
> As a result, we provide a more general necessary and sufficient condition for unique recovery and stronger sample complexity results for learning autonomous linear systems, among which the most important part is the extension to nonlinear systems. Due to the lack of a closed form expression for $x_t$, we cannot utilize the techniques in Yalcin (2023) to analyze nonlinear systems.

---

> ### Author Response · Authors · 2024-11-16
> **Rebuttal #2**
>
> *In line 229 you compare your result with [Yalcin et al. (2023)], Figure 1 from their paper. Since your result is theoretical, and you simplify it for large $\Delta$ for this comparison, could you compare it not with the table of values, but with their theoretical guarantees?*
>
> **Response**: We can compare the sample complexity results for autonomous linear systems of the form $x_{t+1} = \bar{A} x_t + \bar{d}_t$, since $f(x_t) = x_t$ is Lipschitz with constant $L = 1$. The sample complexity result from Yalcin et al. (2023) for autonomous dynamical systems grows with $n^2 \log(n)$ with respect to the dimension, $\max\{p^{-1}(1-p)^{-1}, (1-p)^{-2}\}$ with respect to the fraction of corrupted data, and $(1-\rho L)^{-3}$ with respect to the stability of the system. Similarly, our result in Theorem 5 for Lipschitz basis functions grows with $n^2$ with respect to the dimension, $\max\{p^{-1}(1-p)^{-1}, (1-p)^{-2}\}$ with respect to the fraction of corrupted data, and $(1-\rho L)^{-3}$ with respect to the stability of the system. Therefore, we argue that the sample complexity results for Theorem 5 and Yalcin et al. are similar up to a logarithmic factor, with our result being a generalization for any Lipschitz basis function, can be applied to almost all dynamical systems by selecting suitable basis functions.

---

> ### Author Response · Authors · 2024-11-16
> **Rebuttal #3**
>
> *Also, your randomized version of the problem looks very similar to linear regression with oblivious adversary (see, for example [1] and [2]), and your Assumption 3 reminds the strong convexity assumption from [2]. Could you please check if the models or their analysis of those papers have any connection to your result? If it is the case, it might be nice since their results were generalized to sparse regression and other problems (see [3], they used the symmetry assumption that is somewhat similar to your stealthy condition), so it might be interesting if there exists a general framework that captures also your problem.*
>
> *[1] Arun Sai Suggala, Kush Bhatia, Pradeep Ravikumar, Prateek Jain. Adaptive Hard Thresholding for Near-optimal Consistent Robust Regression.*
>
> *[2] Tommaso d'Orsi, Gleb Novikov, David Steurer. Consistent regression when oblivious outliers overwhelm.*
>
> *[3] Tommaso d'Orsi, Rajai Nasser, Gleb Novikov, David Steurer. Higher degree sum-of-squares relaxations robust against oblivious outliers.*
>
> **Response**: Thank you for pointing out these interesting papers on robust regression. There are similarities between the references and our work, and we will include them in the literature review section of our manuscript in the next revision. However, there are some foundational differences between the problem structures in our paper and those in the references.
>
> When we examined the problem setup in reference [1], we noticed that the corruption vector $\bar{d}_t$ (denoted as $b_i^*$ in the reference) is independent of the system states. However, we do not impose this assumption on our attack vectors. For example, in our experiments with Lipschitz basis functions, we create attack vectors that depend on the current state of the system. Specifically, for each time instance $t \in \mathcal{K}$, the noise $\bar{d}_t$ is generated by
>
> $$ \bar{d}_t := \ell_t \hat{d}_t,\quad \text{where } \ell_t \sim \mathcal{N}(0, \sigma_t^2), ~ \hat{d}_t \sim \mathrm{uniform}(\mathbb{S}^{n-1}), ~ \ell_t \text{ and }\hat{d}_t \text{ are independent}.$$
>
> Here, we define $\sigma_t^2 := \min\{\|x_t\|_2^2, 1 / n\}$. We can verify that the random variable $\ell_t$ is zero-mean and sub-Gaussian with parameter $\sigma = 1$. Additionally, the random vector $\hat{d}_t$ follows a uniform distribution, thus satisfying Assumption 6. Note that the variance of $\bar{d}_t$ depends on the current state of the system, which violates the oblivious adversary assumption in reference [1].
>
> Similarly, reference [2] assumes an oblivious adversary and restricts each entry of the adversary vector to take values between $-1$ and $1$. Our adversarial vector assumption does not make any restriction on the magnitude of the vector. As you mentioned, their strong convexity result is similar to Assumption 3, in that Assumption 3 is necessary to guarantee the uniqueness of the solution. Our assumption ensures that we explore the entire state space to observe the behavior of the dynamical system. When the system is stuck in a subspace, we cannot learn the dynamics in unexplored regions.
>
> Lastly, reference [3] also assumes that the injection vectors are oblivious to the current state of the system and that the entries of the injection vector are independent of each other. We do not make such an independence assumption for our vectors $\bar{d}_t$.
>
> In summary, the existing literature including the above interesting references work has primarily focused on the case with independent disturbances. Studying the case with correlated disturbances is challenging, which is the core of our work.

---

> ### Author Response · Authors · 2024-11-16
> **Rebuttal #4**
>
> *One other question (somewhat related to the previous one): you have a $\log(m)$ factor in Theorem 3. Do you think it is an artifact of your proof, or do you expect it to be inherent?*
>
> **Response**: We think the $\log(m)$ factor in Theorem 3 is an artifact of the proof and the definition of the bounded basis functions. In our proof, we need to show that the sufficient condition in equation (10) is satisfied for every non-zero $Z$ with $\lVert Z\rVert_F \le 1$. We use a metric entropy and covering argument for the unit ball in terms of the Frobenius norm, utilizing small $\epsilon$-radius balls in terms of the Frobenius norm to cover the unit ball. Since the objective function is the sum of $\ell_2$ norms and we use the Frobenius norm for matrices, the bounds also involve the $\ell_2$ norm of the vectors $f(x_t)$. In the case of bounded basis functions, we assume that the bound on the basis functions is in terms of the $\ell_\infty$ norm. The conversion between the $\ell_2$ norm and the $\ell_\infty$ norm requires a $\sqrt{m}$ scaling factor. Because the covering number $N$, i.e., the number of small $\epsilon$-radius balls required to cover the unit ball, appears inside the logarithmic term, an additional $\log(m)$ factor arises in Theorem 3. We can remove the $\log(m)$ factor if we change Assumption 1 to $\lVert f(x_t) \rVert_2 \le B, \forall x \in \mathbb{R}^n$. In the case of Lipschitz basis functions, everything scales well for Theorem 5 because the Lipschitz condition in Assumption 4 is defined in terms of $\ell_2$ norms. As a result, we do not have the $\log(m)$ factor in that case.

---

### Author Response · Authors · 2024-11-20
**Checking In Regarding Rebuttals**

Since the discussion period would end in a week, could you please read our rebuttal and let us know if you have any further concerns or comments? We provided extensive answers to your previous comments and hope to discuss them with you during this discussion period. We appreciate your time and effort.

---

### Meta-Review · Area_Chair_yRuS · 2024-12-11

**Metareview:**

The reviewers on this paper are rather mixed, with a number of positive and negative points being raised.  However, two reviewers still have non-minor concerns about the justification of the problem formulation and its assumptions.  I tend to agree that these are non-minor.  For example, even in the newly added (blue) text, some important unsubstantiated claims are made like “has been commonly used in real-world systems” and “have appeared in different parts of the world”.  I can also appreciate, for example, the significant concern that "the main results get worse as the fraction of adversarially-affected time-steps decreases", which raises significant doubts on whether this should really be viewed as "adversarial".  The authors gave some explanation and argued that some of these assumptions are needed to obtain the results.  But that is not necessarily a convincing justification, as it can raise broader doubts about the theoretical model itself.

**Additional Comments On Reviewer Discussion:**

The author-reviewer discussion was summarized in the meta-review.  The private discussion confirmed the remaining concerns.

---

### Decision · Program_Chairs · 2025-01-22

Reject